# CROSS-DOMAIN FEW-SHOT CLASSIFICATION VIA MAXIMIZING OPTIMIZED KERNEL DEPENDENCE

## ABSTRACT

In cross-domain few-shot classification, *nearest centroid classifier* (NCC) aims to learn representations to construct a metric space where few-shot classification can be performed by measuring the similarities between samples and the prototype of each class. An intuition behind NCC is that each sample is pulled closer to the class centroid it belongs to while pushed away from other classes. However, in this paper, we find that there exist high similarities between NCC-learned representations of two samples from different classes. These undesirable high similarities may induce uncertainty and further result in incorrect classification of samples. In order to solve this problem, we propose a bi-level optimization framework, *maximizing optimized kernel dependence* (MOKD), to learn a set of class-specific representations via simultaneously maximizing dependence between representations and labels and minimizing dependence among all samples based on the optimized kernel dependence measures. Specifically, MOKD first optimizes the kernel used in *Hilbert-Schmidt Independence Criterion* (HSIC) to obtain the optimized kernel HSIC (opt-HSIC) by maximizing its test power to increase its capability in dependence detection. Then, an optimization problem regarding the opt-HSIC is addressed to simultaneously maximize the dependence between representations and labels and minimize the dependence among all samples. The learning process can be intuitively treated as exploring a set of class-specific representations that match the cluster structures of labeled data in the given set. Extensive experiments on representative Meta-Dataset benchmark demonstrate that MOKD can not only achieve better generalization performance on unseen domains in most cases but also learn better data clusters for each class in the given data set.

## 1 INTRODUCTION

Cross-domain few-shot classification (Dvornik et al., 2020; Li et al., 2021a; Liu et al., 2021a; Triantafillou et al., 2020), also known as CFC, is a learning paradigm which aims at learning to perform classification on tasks sampled from previously unseen data or domains where only a few labeled training data are available. Compared with conventional few-shot classification problem (Finn et al., 2017; Ravi & Larochelle, 2017; Snell et al., 2017; Vinyals et al., 2016) which learns to adapt to new tasks sampled from unseen data with the same distribution as seen data, cross-domain few-shot classification is a much more challenging learning task since there exist discrepancies between the distributions of source and target domains (Chi et al., 2021; Kuzborskij & Orabona, 2013).

Due to its simplicity and scalability, *nearest centroid classifier* (NCC) (Snell et al., 2017) has been widely applied in recent works (Doersch et al., 2020; Li et al., 2021a; Liu et al., 2021a; Triantafillou et al., 2020) regarding cross-domain few-shot classification. The goal of NCC is to learn representations to construct a metric space where few-shot classification can be performed by measuring the similarities between samples and the prototype of each class. Intuitively, the learning process via NCC is pulling each sample closer to the class centroid it belongs to while pushing it away from other class centroids. Thus, the learned representations are expected to be specific enough to be distinguished from other classes while identified by the class it belongs to.

However, in this paper, we find that there exist high similarities between NCC-learned representations of two samples from different classes. For example, as shown in Fig. 1(a), the heatmap of similarity matrix, which depicts the similarities among data representations, reveals that the NCC-learned

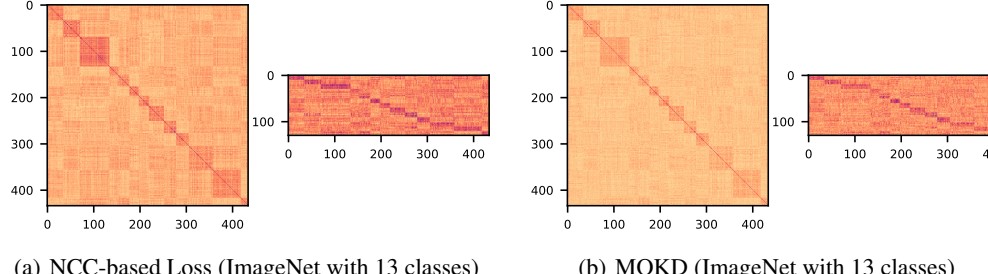

(a) NCC-based Loss (ImageNet with 13 classes)      (b) MOKD (ImageNet with 13 classes)

Figure 1: **Heatmaps of similarity matrices of representations respectively learned with NCC-based loss and MOKD.** The left of each figure describes the similarities among all support data representations and the right side describes the similarities between query data and support data representations. As shown in (a), NCC-learned representations of samples are not only similar to samples belonging to their own class, but also similar to samples from other classes. (b) shows that the undesirable high similarities existing between samples from different classes are significantly alleviated and the cluster structures of the given set of data are well explored by applying MOKD.

representations of each sample not only resemble the samples within its own class, but also have high similarities with samples from other classes. Such undesirable high similarities among samples may induce uncertainty and further result in misclassification of samples. Thus, learning a set of *class-specific* representations, where similarities among samples within the same class are maximized while similarities between samples from different classes are minimized, is crucial for CFC tasks.

To this end, we propose an efficient and effective approach, *maximizing optimized kernel dependence* (MOKD), to maximize the similarities among samples belonging to the same class and minimize the similarities between samples from different classes simultaneously. Generally, MOKD is formulated as a bi-level optimization problem for dependence optimization based on optimized kernel measures. To be concrete, MOKD first optimizes the kernel used in Hilbert-Schmidt Independence Criterion (HSIC) via maximizing its test power to obtain a powerful kernel dependence measure, the optimized kernel HSIC (opt-HSIC). The goal of this step is to make the kernel HSIC more sensitive to dependence detection so that the dependence among representations of samples can be precisely measured. Then, an optimization problem with respect to the opt-HSIC is addressed to simultaneously maximize the dependence between representations and labels and minimize the dependence among all samples. In this way, a set of class-specific representations are learned. As shown in Fig 1(b), MOKD does help alleviate the aforementioned undesirable high similarities and learn better cluster structures.

Extensive experiments on representative Meta-Dataset (Triantafillou et al., 2020) benchmark under several mainstream settings and further experimental analyses demonstrate that MOKD is an efficient and effective algorithm that can achieve better generalization performance than previous baselines on unseen domains. Moreover, visualization results also show that MOKD can learn better class clusters and alleviate undesirable high similarities between representations of samples from different classes.

## 2 PRELIMINARY

**Dataset Structure.** Let $\mathcal{S}$ denote a meta dataset that contains $n$ sub-datasets with different distributions, i.e. $\mathcal{S} = \{\mathcal{S}_1, \mathcal{S}_2, ..., \mathcal{S}_n\}$. For each sub-dataset $\mathcal{S}_i$, three *disjoint* subsets, which are training set $\mathcal{D}_{\mathcal{S}_i}^{\text{tr}}$, validation set $\mathcal{D}_{\mathcal{S}_i}^{\text{val}}$ and test set $\mathcal{D}_{\mathcal{S}_i}^{\text{test}}$, are included, i.e. $\mathcal{S}_i = \{\mathcal{D}_{\mathcal{S}_i}^{\text{tr}}, \mathcal{D}_{\mathcal{S}_i}^{\text{val}}, \mathcal{D}_{\mathcal{S}_i}^{\text{test}}\}$. In the context of cross-domain few-shot classification, a feature encoder is trained on the training sets of a portion of sub-datasets of $\mathcal{S}$. Thus, the sub-datasets whose training sets are observed by feature encoder during pre-training phase is called *seen domains* and denoted as $\mathcal{S}^{\text{seen}}$ while the remaining datasets are called *unseen domains* and denoted as $\mathcal{S}^{\text{unseen}}$. Note that $\mathcal{S}^{\text{seen}}$ and $\mathcal{S}^{\text{unseen}}$ are also disjoint.

**Task Generation.** A task $\mathcal{T} = \{\mathcal{D}_{\mathcal{T}}, \mathcal{Q}_{\mathcal{T}}\}$, where $\mathcal{D}_{\mathcal{T}} = \{\boldsymbol{X}^{\text{s}}, Y^{\text{s}}\} = \{(\boldsymbol{x}_i^{\text{s}}, y_i^{\text{s}})\}_{i=1}^{|\mathcal{D}_{\mathcal{T}}|}$ denotes the support data pairs and $\mathcal{Q}_{\mathcal{T}} = \{\boldsymbol{X}^{\text{q}}, Y^{\text{q}}\} = \{(\boldsymbol{x}_j^{\text{q}}, y_j^{\text{q}})\}_{j=1}^{|\mathcal{Q}_{\mathcal{T}}|}$ denotes query data pairs, is randomly sampled from the specific dataset at the beginning of each episode. The sampling of *vary-way vary-shot* cross-domain few-shot classification tasks follows the rules proposed in Triantafillou et al. (2020). The sampling process can be roughly divided into two steps. Firstly, the number of classes $N$

is randomly sampled from the interval $[5, N_{\max}]$, where $N_{\max}$ denotes the maximum of the number of classes and is either 50 or as many classes as available. Then, the number of shots of each class, $K_n$, for $n = \{1, 2, ..., N\}$, in support set is determined with specific rules (refer to Triantafillou et al. (2020) or Appendix E.4 for details). Thus, in the support set of a given task, each datapoint belonging to class $n$ can be treated as independently sampling from the given dataset with the probability $\frac{1}{NK_n}$.

**Pre-training Few-shot Classification Pipeline.** Consider a pre-trained backbone $f_{\phi^*}$ parameterized with the optimal pre-trained parameters $\phi^*$ and a linear transformation head $h_\theta$ parameterized with $\theta$. Given a support set $\mathcal{D}_\mathcal{T}$, the corresponding representations can be obtained by applying the pre-trained backbone $f_{\phi^*}$ and the linear transformation head $h_\theta$: $\mathcal{Z} = \{z_i^s\}_{i=1}^{|\mathcal{D}_\mathcal{T}|} = \{h_\theta \circ f_{\phi^*}(x_i^s)\}_{i=1}^{|\mathcal{D}_\mathcal{T}|}$. Then, according to URL (Li et al., 2021a), the learning problem can be solved by minimizing the following NCC-based loss (Snell et al., 2017) on the given support set $\mathcal{D}_\mathcal{T}$:

$$\mathcal{L}_{\text{NCC}}(\theta, \mathcal{D}_\mathcal{T}) = -\frac{1}{|\mathcal{D}_\mathcal{T}|} \sum_{i=1}^{|\mathcal{D}_\mathcal{T}|} \log\left(p(y = y_i^s | z_i^s, \theta)\right), \tag{1}$$

where $p(y = c | z, \theta) = \frac{e^{k(z, c_c)}}{\sum_{i=1}^{N_C} e^{k(z, c_i)}}$ denotes the likelihood of a given sample $z$ belonging to class $c$, $k(\cdot, \cdot)$ denotes a kernel function which is formulated as a cosine similarity function in URL, $c_c$ denotes the prototype of class $c$ and is calculated as $c_c = \frac{1}{|\mathcal{C}_c|} \sum_{z \in \mathcal{C}_c} z, \mathcal{C}_c = \{z_j | y_j = c\}$.

**Hilbert-Schmidt Independence Criterion.** Given separable *reproducing kernel Hilbert spaces* (RKHSs) $\mathcal{F}$, $\mathcal{G}$ and two feature spaces $\mathcal{X}$, $\mathcal{Y}$, HSIC (Gretton et al., 2005a) measures the dependence between two random variables $X \in \mathcal{X}$ and $Y \in \mathcal{Y}$ by evaluating the norm of the cross-variance between the features that are respectively transformed by non-linear kernels: $\varphi : \mathcal{X} \to \mathcal{F}$ and $\psi : \mathcal{Y} \to \mathcal{G}$:

$$\text{HSIC}(X, Y) = ||\mathbb{E}[\varphi(X)\psi(Y)^\top] - \mathbb{E}[\varphi(X)]\mathbb{E}[\psi(Y)]^\top||_{HS}^2, \tag{2}$$

where $|| \cdot ||_{HS}$ is the Hilbert-Schmidt norm which is the Frobenius norm in finite dimensions. Further, let $(X', Y')$ and $(X'', Y'')$ be the independent copies of $(X, Y)$, the HSIC can be formulated as:

$$\text{HSIC}(X, Y) = \mathbb{E}[k(X, X')l(Y, Y')] - 2\mathbb{E}[k(X, X')l(Y, Y'')] + \mathbb{E}[k(X, X')]\mathbb{E}[l(Y, Y')], \tag{3}$$

where $k(x, x') = \langle \varphi(x), \varphi(x') \rangle_\mathcal{F}$ and $l(y, y') = \langle \psi(y), \psi(y') \rangle_\mathcal{G}$ are kernel functions which are defined as inner product operations in reproducing kernel Hilbert space. Note that HSIC takes zero if and only if $X$ and $Y$ are mutually independent (Gretton et al., 2005a; Song et al., 2012).

**Test power of HSIC.** In this paper, test power is used to measure the probability that, for particular two dependent distributions and the number of samples $m$, the null hypothesis that the two distributions are independent is correctly rejected. Consider a $\widehat{\text{HSIC}}_u$ as an unbiased HSIC estimator (e.g., U-statistic estimator), under the hypothesis that the two distributions are dependent, the central limit theorem (Serfling, 2009) holds:

$$\sqrt{m}(\widehat{\text{HSIC}}_u^2 - \text{HSIC}^2) \xrightarrow{d} \mathcal{N}(0, v^2),$$

where $v^2$ denotes the variance, $\xrightarrow{d}$ denotes convergence in distribution. The CLT implies that test power can be formulated as:

$$\Pr\left(m\widehat{\text{HSIC}}_u^2 > r\right) \to \Phi\left(\frac{\sqrt{m}\text{HSIC}^2}{v} - \frac{r}{\sqrt{m}v}\right),$$

where $r$ denotes a rejection threshold and $\Phi$ denotes the standard normal CDF. Since the rejection threshold $r$ will converge to a constant, and HSIC, $v$ are constants, for reasonably large $m$, the test power is dominated by the first term. Thus, a feasible way to maximize the test power is to find a kernel function to maximize $\text{HSIC}^2/v$. The intuition of test power maximization is increasing the sensitivity of the estimated kernel to the dependence among data samples.

## 3   AN UNDERSTANDING OF NCC FROM PERSPECTIVE OF HSIC

In this section, we provide an understanding of NCC-based loss from perspective of HSIC. Specifically, we first reveals two insights behind NCC-based loss. Then, inspired by Li et al. (2021b), we bridge a connection between NCC-based loss and HSIC, and find that the upper bound of NCC-based loss can be treated as the surrogate loss under mild assumptions. All proofs are available in Appendix C.

### 3.1 TWO INSIGHTS BEHIND NCC-BASED LOSS

**Theorem 1 (An upper bound of NCC-based loss)** *Given a kernel function $k(\cdot, \cdot)$, a support set $\mathcal{D} = \{(\boldsymbol{x}_i, y_i)\}_{i=1}^{|\mathcal{D}|}$ that includes $N_C$ classes. Let $(\boldsymbol{X}, Y) \sim \mathbb{U}_{\mathcal{D}}$ represent a random variable $(\boldsymbol{X}, Y)$ whose distribution is a uniform distribution defined on a dataset $\mathcal{D}$. Then, with the corresponding support representation set $\mathcal{Z} = \{\boldsymbol{z}_i\}_{i=1}^{|\mathcal{D}|} = \{h_\theta \circ f_{\phi^*}(\boldsymbol{x}_i)\}_{i=1}^{|\mathcal{D}|}$, the NCC-based loss $-\mathbb{E}_{(\boldsymbol{X}, Y) \sim \mathbb{U}_{\mathcal{D}}} \log(p(Y = \hat{y} | \boldsymbol{X}, \theta))$ is upper bounded by:*

$$-\mathbb{E}_{\boldsymbol{Z} \sim \mathbb{U}_{\mathcal{Z}}} \mathbb{E}_{\boldsymbol{Z}'' \sim \mathbb{U}_{\mathcal{C}}} k(\boldsymbol{Z}, \boldsymbol{Z}'') + \mathbb{E}_{\boldsymbol{Z} \sim \mathbb{U}_{\mathcal{Z}}} \log \mathbb{E}_{\boldsymbol{Z}' \sim \mathbb{U}_{\mathcal{Z}}} e^{k(\boldsymbol{Z}, \boldsymbol{Z}')} + C,$$

*where all elements of $\mathcal{C} \subset \mathcal{Z}$ belong to the same class, $\boldsymbol{Z}'$ is an independent copy of $\boldsymbol{Z}$, $\hat{y}$ denotes the prediction and $C = \log N_C$ is a constant. The equation is satisfied if and only if the similarity $k(\boldsymbol{Z}, \boldsymbol{Z}')$ for arbitrary two samples $\boldsymbol{Z}$ and $\boldsymbol{Z}'$ in a class is same.*

The upper bound in Theorem 1 reveals two insights behind NCC-based loss. On the one hand, it maximizes the similarities among samples within the same class via the first term. On the other hand, the similarities between arbitrary two samples in support set are minimized through the second term. This potentially helps reduce the similarities derived from trivial information (e.g. similar background information) and further contributes to guiding models to focus on the correct feature areas.

### 3.2 CONNECTION BETWEEN NCC-BASED LOSS AND HSIC MEASURE

In this section, we follow Li et al. (2021b) to bridge a connection between NCC-based loss and HSIC.

**Definition 2 (Label kernel)** *Given a support set that includes $N_C$ classes and $K_n$ shots for each class $n = \{1, 2, ..., N_C\}$, we assume that the label of each data point is encoded as a one-hot vector, and each datapoint belonging to class $n$ is sampled with the probability $\frac{1}{N_C K_n}$. Then, any kernel that is a function of $y_i^\top y_j$ or $||y_i - y_j||$ have the form*

$$l(y_i, y_j) = \begin{cases} l_1, & y_i = y_j, \\ l_0, & otherwise \end{cases} \equiv \Delta l \mathbb{I}(y_i = y_j) + l_0, \quad where \quad \Delta l = l_1 - l_0. \tag{4}$$

**Theorem 3** *Given a support representation set $\mathcal{Z} = \{\boldsymbol{z}_i\}_{i=1}^{|\mathcal{Z}|} = \{h_\theta \circ f_{\phi^*}(\boldsymbol{x}_i)\}_{i=1}^{|\mathcal{Z}|}$ where $N_C$ classes are included, a kernel function $k(\cdot, \cdot)$ defined in Eq. (3) and a label kernel $l(\cdot, \cdot)$ defined in Eq. (4), let $\mathbb{U}_{\mathcal{Z}}$ denotes a uniform distribution defined on $\mathcal{Z}$, $\mathrm{HSIC}(\boldsymbol{Z}, Y)$ takes the form*

$$\mathrm{HSIC}(\boldsymbol{Z}, Y) = \frac{\Delta l}{N_C} \left( \mathbb{E}_{\boldsymbol{Z} \sim \mathbb{U}_{\mathcal{Z}}} \mathbb{E}_{\boldsymbol{Z}'' \sim \mathbb{U}_{\mathcal{C}}} \left[ k(\boldsymbol{Z}, \boldsymbol{Z}'') \right] - \mathbb{E}_{\boldsymbol{Z} \sim \mathbb{U}_{\mathcal{Z}}, \boldsymbol{Z}' \sim \mathbb{U}_{\mathcal{Z}}} \left[ k(\boldsymbol{Z}, \boldsymbol{Z}') \right] \right),$$

*where all elements of $\mathcal{C} \subset \mathcal{Z}$ belong to the same class, $\boldsymbol{Z}'$ is an independent copy of $\boldsymbol{Z}$, $\mathbb{U}_{\mathcal{Z}}$ and $\mathbb{U}_{\mathcal{C}}$ denote the uniform distributions respectively defined on $\mathcal{Z}$ and $\mathcal{C}$.*

**Remark 1** *Under the settings of few-shot classification task, Theorem 3 shows that $\mathrm{HSIC}(\boldsymbol{Z}, Y)$ shares the similar structure with the upper bound of NCC-based loss obtained in Theorem 1. Such phenomenon results from the label kernel defined in Definition 2 which indicates the cluster structure related to the given support set. Intuitively, maximizing $\mathrm{HSIC}(\boldsymbol{Z}, Y)$ is equivalent driving models to explore a set of class-specific representations that matches the cluster structure in the given task.*

**Theorem 4** *Assume that $\Delta l = N_C$, $k(\boldsymbol{Z}, \boldsymbol{Z}')$ does not deviate much from $\mathbb{E}_{\boldsymbol{Z}' \sim \mathbb{U}_{\mathcal{Z}}} \left[ k(\boldsymbol{Z}, \boldsymbol{Z}') \right]$. Given the same mild assumptions in Theorem 1 and 3, by applying Taylor-expanding to the term $\mathbb{E}_{\boldsymbol{Z} \sim \mathbb{U}_{\mathcal{Z}}} \log \mathbb{E}_{\boldsymbol{Z}' \sim \mathbb{U}_{\mathcal{Z}}} e^{k(\boldsymbol{Z}, \boldsymbol{Z}')}$ around $\mathbb{E}_{\boldsymbol{Z}' \sim \mathbb{U}_{\mathcal{Z}}} \left[ k(\boldsymbol{Z}, \boldsymbol{Z}') \right]$, we have:*

$$-\mathbb{E}_{(\boldsymbol{X}, Y) \sim \mathbb{U}_{\mathcal{D}}} \log(p(Y = \hat{y} | \boldsymbol{X}, \theta)) \leq -\mathrm{HSIC}(\boldsymbol{Z}, Y) + \frac{1}{2} \mathbb{E}_{\boldsymbol{Z} \sim \mathbb{U}_{\mathcal{Z}}} \left[ \mathbb{V}\mathrm{ar}_{\boldsymbol{Z}' \sim \mathbb{U}_{\mathcal{Z}}} \left[ k(\boldsymbol{Z}, \boldsymbol{Z}') \right] \right] + C.$$

In Theorem 4, the upper bound of NCC-based loss is further expressed as a combination of $\mathrm{HSIC}(\boldsymbol{Z}, Y)$ and a high-variance moments term. As revealed in Theorem 3, $\mathrm{HSIC}(\boldsymbol{Z}, Y)$ owns the similar structure and plays the same role as the upper bound obtained in Theorem 1 which aims to

simultaneously maximizes the similarities among samples belonging to the same class and minimizes the similarities between samples from different classes. The high-variance moments term functions as a regularization term that penalizes the high-variance kernelized representations. Moreover, an important insight derived from Theorem 4 is that the obtained upper bound can be treated as a surrogate loss of NCC-based loss if the mild assumptions that $k(\boldsymbol{Z}, \boldsymbol{Z}^{'})$ does not deviate much from $\mathbb{E}_{\boldsymbol{Z}^{'} \sim \mathbb{U}_{\mathcal{Z}}} \left[ k(\boldsymbol{Z}, \boldsymbol{Z}^{'}) \right]$ and the similarity of arbitrary two samples in a class is same are satisfied.

### 3.3 PERFORM CFC TASKS WITH HSIC

**Comparison to NCC.** Compared with NCC-based loss, the surrogate loss mentioned above owns the following two desirable merits. First of all, different from NCC-based loss which implicitly and simply leverage label information via class centroids to learn a data cluster for each class, the first term of the surrogate loss explicitly measures the dependence between support data representations and labels that contain cluster structure information of the given set to explore the class-specific representations for each class. Moreover, due to the scarce training data included in few-shot classification tasks, the model usually overfits the data and in turn obtains bad generalization performance. However, the high-variance moments term in the surrogate loss is able to alleviate such undesirable phenomenon.

**Challenges When Using HSIC.** A challenge of applying HSIC to performing cross-domain few-shot classification task is that kernel HSIC may fail to accurately measure the dependence between two data samples. For example, as shown in Fig. 1(a), although the similarities among samples within the same class are maximized, there still exist high similarities between two samples from different classes. This phenomenon implies that NCC-based loss fails to accurately measure the dependence between representations and labels, and in turn fails to learn a set of class-specific representations that match the cluster structures of the support set. Such phenomenon may induce undesirable uncertainty and further result in misclassification of samples. To address this problem, a feasible way is improving the capability of HSIC in detecting the dependence to learn a set of class-specific representations.

## 4 MAXIMIZING OPTIMIZED KERNEL DEPENDENCE

In order to solve the above problem, we propose *maximizing optimized kernel dependence* (MOKD) to perform cross-domain few-shot classification with HSIC where kernels are optimized to accurately measure the dependence. Specifically, we first maximize the test power of the kernels used in HSIC to improve its capability in dependence detection, and then optimizing the dependence respectively between representations and labels, and among representations based on the optimized kernels. Intuitively, test power maximization increases sensitivity of kernels to dependence. In this way, the dependence among samples can be more accurately measured and effectively optimized so that we can learn a set of class-specific representations where the similarities among samples within the same class are maximized while the similarities between two samples from different classes are minimized.

### 4.1 PROBLEM FORMULATION FOR MOKD

Consider a set of $m$ support representations pairs $\mathcal{D}_{\mathcal{Z}} = \{(\boldsymbol{z}_i, y_i)\}_{i=1}^m$, the ultimate goal of MOKD is to learn a set of optimal task-specific parameters $\theta^*$ from given data by performing optimization on the optimized HSIC where the test power is maximized to increase the sensitivity to the dependence. Thus, MOKD is formulated as a bi-level optimization problem:

$$\min_{\theta} -\mathrm{HSIC}(\boldsymbol{Z}, Y; \sigma_{\boldsymbol{Z}Y}^*, \theta) + \gamma \mathrm{HSIC}(\boldsymbol{Z}, \boldsymbol{Z}; \sigma_{\boldsymbol{Z}\boldsymbol{Z}}^*, \theta)$$

$$s.t. \max_{\sigma_{\boldsymbol{Z}Y}} \frac{\mathrm{HSIC}(\boldsymbol{Z}, Y; \sigma_{\boldsymbol{Z}Y}, \theta)}{\sqrt{v_{\boldsymbol{Z}Y} + \epsilon}}, \max_{\sigma_{\boldsymbol{Z}\boldsymbol{Z}}} \frac{\mathrm{HSIC}(\boldsymbol{Z}, \boldsymbol{Z}; \sigma_{\boldsymbol{Z}\boldsymbol{Z}}, \theta)}{\sqrt{v_{\boldsymbol{Z}\boldsymbol{Z}} + \epsilon}}, \tag{5}$$

where $\sigma_{\boldsymbol{Z}Y}$ and $\sigma_{\boldsymbol{Z}\boldsymbol{Z}}$ are the bandwidths of Gaussian kernels respectively calculated in $\mathrm{HSIC}(\boldsymbol{Z}, Y)$ and $\mathrm{HSIC}(\boldsymbol{Z}, \boldsymbol{Z})$, $v_{\boldsymbol{Z}Y}$ and $v_{\boldsymbol{Z}\boldsymbol{Z}}$ are the variances of estimated $\mathrm{HSIC}(\boldsymbol{Z}, Y)$ and $\mathrm{HSIC}(\boldsymbol{Z}, \boldsymbol{Z})$, $\gamma$ is the scalar coefficient of $\mathrm{HSIC}(\boldsymbol{Z}, \boldsymbol{Z})$ term and $\epsilon$ is a scalar that is used to avoid the case $v \leq 0$.

Since the true distribution of support data features is unknown, in this paper, we follow Song et al. (2012) and estimate the kernel HSIC with a set of finite data samples in the unbiased way:

$$\widehat{\mathrm{HSIC}}(\boldsymbol{Z}, Y) = \frac{1}{m(m-3)} \left[ \mathrm{tr}\left( \tilde{\boldsymbol{K}} \tilde{\boldsymbol{L}} \right) + \frac{\mathbf{1}^\top \tilde{\boldsymbol{K}} \mathbf{1}\mathbf{1}^\top \tilde{\boldsymbol{L}} \mathbf{1}}{(m-1)(m-2)} - \frac{2}{m-2} \mathbf{1}^\top \tilde{\boldsymbol{K}} \tilde{\boldsymbol{L}} \mathbf{1} \right], \tag{6}$$

where $\tilde{\boldsymbol{K}}$ and $\tilde{\boldsymbol{L}}$ are kernel matrices where $\tilde{\boldsymbol{K}}_{i,j} = (1 - \delta_{i,j})k(\boldsymbol{z}_i, \boldsymbol{z}_j)$ and $\tilde{\boldsymbol{L}}_{i,j} = (1 - \delta_{i,j})l(y_i, y_j)$, $m$ denotes the number of samples in support set. The calculation of $\widehat{\mathrm{HSIC}}(\boldsymbol{Z}, Y)$ takes $\mathcal{O}(m^2)$ time.

The bi-level optimization problem proposed in Eq. (5) mainly contains two aspects: inner optimization for test power maximization and outer optimization for dependence optimization. To be specific, during inner optimization phase, MOKD optimizes the kernel HSIC to maximize its test power for better dependence detection via maximizing $\frac{\mathrm{HSIC}(\cdot,\cdot;\sigma,\theta)}{\sqrt{v+\epsilon}}$. In this way, the kernel HSIC is more sensitive to dependence and thus the dependence among data samples can be precisely measured. During outer optimization, with the optimized kernel HSIC, MOKD maximizes the dependence between representations and labels to explore a set of class-specific representations, where the similarities among samples within the same class are maximized while the similarities between samples from different classes are minimized, to match the cluster structures of the given support set. Meanwhile, the dependence among all representations is minimized as a regularization to penalize the high-variance representations to alleviate the overfitting phenomenon derived from scarce data.

In problem formulation, we adopt a scaled $\mathrm{HSIC}(\boldsymbol{Z}, \boldsymbol{Z})$ instead of $\mathbb{E}_{\boldsymbol{Z}\sim\mathbb{U}_\mathcal{Z}}\left[\mathbb{Var}_{\boldsymbol{Z}'\sim\mathbb{U}_\mathcal{Z}}\left[k(\boldsymbol{Z}, \boldsymbol{Z}')\right]\right]$ as the regularization term for penalizing high-variance kernelized representations. On the one hand, $\mathrm{HSIC}(\boldsymbol{Z}, \boldsymbol{Z})$ plays the same role as the high-order moments term by minimizing the dependence among all samples and it is easy to prove $\mathrm{HSIC}(\boldsymbol{Z}, \boldsymbol{Z}) \leq \mathbb{E}_{\boldsymbol{Z}\sim\mathbb{U}_\mathcal{Z}}\left[\mathbb{Var}_{\boldsymbol{Z}'\sim\mathbb{U}_\mathcal{Z}}\left[k(\boldsymbol{Z}, \boldsymbol{Z}')\right]\right]$ with the definition of HSIC proposed in Eq. (3). On the other hand, compared with high-order moments term, HSIC family has been well studied in abundant previous works (Blaschko & Gretton, 2008; Ma et al., 2020; Nøkland & Eidnes, 2019; Pogodin & Latham, 2020; Song et al., 2007; 2012), and these works have demonstrated that HSIC owns many desirable properties in calculation and theoretical analysis. Since an important component in MOKD is kernel optimization for more precise dependence detection, HSIC, which is a reliable and simple statistical measure, is more preferable.

**Differences between SSL-HSIC and MOKD.** We notice that the outer optimization objective in Eq. (5) shares the similar format as that in SSL-HSIC (Li et al., 2021b). From our perspective, in fact, there are two major differences between them. Firstly, the most obvious difference between SSL-HSIC and MOKD is that MOKD takes the test power into consideration. This facilitates to increase the sensitivity of kernel HSIC to dependence and further contributes to dependence optimization. In addition, SSL-HSIC derives from unsupervised contrastive learning and focuses on learning robust and discriminative features by contrasting two different views of a sample. However, MOKD derives from supervised few-shot classification and aims at learning a set of class-specific features where similarities among samples within the same class are maximized while similarities between samples from different classes are minimized. More details and empirical results are available in Appendix D.

## 4.2 Adaptive Bandwidth Selection for Maximizing Test Power

Since accurately measuring dependence is crucial for dependence optimization, in this paper, we propose to first optimize the kernel HSIC to maximize its test power to improve its capability in detecting dependence. According to preliminary, the test power maximization problem is formulated as finding a kernel function to maximize $\frac{\mathrm{HSIC}(\cdot,\cdot;\sigma,\theta)}{\sqrt{v+\epsilon}}$. In practice, we adopt Gaussian kernel which contains a parameter $\sigma$ as the kernel function. Thus, test power maximization can be further formulated as finding an optimal bandwidth $\sigma^*$ for the kernel function to maximize $\frac{\mathrm{HSIC}(\cdot,\cdot;\sigma,\theta)}{\sqrt{v+\epsilon}}$.

As we can observed, a key step of performing test power maximization is estimating the variance of $\mathrm{HSIC}(\cdot, \cdot, \sigma, \theta)$ for each given $\sigma$. According to Theorem 5 in Song et al. (2012), $\widehat{\mathrm{HSIC}}$, which is estimated in the unbiased way of Eq. (6), converges in distribution to a Gaussian random variable with the mean HSIC and the estimated variance:

$$v = \frac{16}{m}\left(R - \widehat{\mathrm{HSIC}}^2\right), R = (4m)^{-1}(m-1)_3^{-2}\boldsymbol{h}^\top\boldsymbol{h}, \tag{7}$$

where $m$ denotes the number of samples, $(m-1)_3$ denotes the Pochhammer symbols $\frac{(m-1)!}{(m-4)!}$, and $\boldsymbol{h}$ is a basic vector for the calculation of $R$, and can be written as:

$$\begin{aligned}\boldsymbol{h} =&(m-2)^2\left(\tilde{\boldsymbol{K}} \circ \tilde{\boldsymbol{L}}\right)\mathbf{1} + (m-2)\left((\mathrm{tr}\tilde{\boldsymbol{K}}\tilde{\boldsymbol{L}})\mathbf{1} - \tilde{\boldsymbol{K}}\tilde{\boldsymbol{L}}\mathbf{1} - \tilde{\boldsymbol{L}}\tilde{\boldsymbol{K}}\mathbf{1}\right) - m(\tilde{\boldsymbol{K}}\mathbf{1}) \circ (\tilde{\boldsymbol{L}}\mathbf{1}) \\ &+ (\mathbf{1}^\top\tilde{\boldsymbol{L}}\mathbf{1})\tilde{\boldsymbol{K}}\mathbf{1} + (\mathbf{1}^\top\tilde{\boldsymbol{K}}\mathbf{1})\tilde{\boldsymbol{L}}\mathbf{1} - (\mathbf{1}^\top\tilde{\boldsymbol{K}}\tilde{\boldsymbol{L}}\mathbf{1})\mathbf{1},\end{aligned} \tag{8}$$

Table 1: **Results on Meta-Dataset (Trained on ImageNet Only).** Mean accuracy and 95% confidence interval are reported.

| Datasets | Finetune | ProtoNets | ProtoNets(large) | BOHB | FP-MAML | ALFA+FP-MAML | FLUTE | SSL-HSIC | URL | MOKD(Ours) |
|---|---|---|---|---|---|---|---|---|---|---|
| ImageNet | 45.8±1.1 | 50.5±1.1 | 53.7±1.1 | 51.9±1.1 | 49.5±1.1 | 52.8±1.1 | 46.9±1.1 | 55.5±1.1 | **57.3±1.1** | **57.3±1.1** |
| Omniglot | 60.9±1.6 | 60.0±1.4 | 68.5±1.3 | 67.6±1.2 | 63.4±1.3 | 61.9±1.5 | 61.6±1.4 | 66.4±1.2 | 69.4±1.2 | **70.9±1.3** |
| Aircraft | 68.7±1.3 | 53.1±1.0 | 58.0±1.0 | 54.1±0.9 | 56.0±1.0 | 63.4±1.1 | 48.5±1.0 | 49.5±0.9 | 57.6±1.0 | **59.8±1.0** |
| Birds | 57.3±1.3 | 68.8±1.0 | **74.1±0.9** | 70.7±0.9 | 68.7±1.0 | 69.8±1.1 | 47.9±1.0 | 71.6±0.9 | 72.9±0.9 | **73.6±0.9** |
| Textures | 69.0±0.9 | 66.6±0.8 | 68.8±0.8 | 68.3±0.8 | 66.5±0.8 | 70.8±0.9 | 63.8±0.8 | 72.2±0.7 | 75.2±0.7 | **76.1±0.7** |
| Quick Draw | 42.6±1.2 | 49.0±1.1 | 53.3±1.0 | 50.3±1.0 | 51.5±1.0 | 59.2±1.2 | 57.5±1.0 | 54.2±1.0 | 57.9±1.0 | **61.2±1.0** |
| Fungi | 38.2±1.0 | 39.7±1.1 | 40.7±1.2 | 41.4±1.1 | 40.0±1.1 | 41.5±1.2 | 31.8±1.0 | 43.4±1.1 | 46.2±1.0 | **47.0±1.1** |
| VGG Flower | 85.5±0.7 | 85.3±0.8 | 87.0±0.7 | 87.3±0.6 | 87.2±0.7 | 86.0±0.8 | 80.1±0.9 | 85.5±0.7 | 86.9±0.6 | **88.5±0.6** |
| Traffic Sign | **66.8±1.3** | 47.1±1.1 | 58.1±1.1 | 51.8±1.0 | 48.8±1.1 | 60.8±1.3 | 46.5±1.1 | 50.5±1.1 | 61.2±1.2 | 61.6±1.1 |
| MSCOCO | 34.9±1.0 | 41.0±1.1 | 41.7±1.1 | 48.0±1.0 | 43.7±1.1 | 48.1±1.1 | 41.4±1.0 | 51.4±1.0 | 53.0±1.0 | **55.3±1.0** |
| MNIST | - | - | - | - | - | - | 80.8±0.8 | 77.0±0.7 | 86.2±0.7 | **88.3±0.7** |
| CIFAR-10 | - | - | - | - | - | - | 65.4±0.8 | 71.0±0.8 | 69.5±0.8 | **72.2±0.8** |
| CIFAR-100 | - | - | - | - | - | - | 52.7±1.1 | 59.0±1.0 | 62.0±1.0 | **63.1±1.0** |
| Average Seen | 45.8 | 50.5 | 53.7 | 51.9 | 49.5 | 52.8 | 46.9 | 55.5 | **57.3** | **57.3** |
| Average Unseen | - | - | - | - | - | - | 56.5 | 62.5 | 66.6 | **68.1** |
| Average All | - | - | - | - | - | - | 55.8 | 62.0 | 65.9 | **67.3** |
| Average Rank | 7.1 | 8.4 | 4.6 | 5.5 | 6.8 | 4.4 | 8.9 | 4.9 | 2.8 | **1.4** |

[1] The results on URL and MOKD are the average of 5 random seed. The ranks only consider the first 10 datasets and are calculated only with the methods in the table.

where $\circ$ denotes elementwise multiplication, $\mathbf{1} \in \mathbb{R}^{m \times 1}$ denotes the vector where all elements are 1.

The maximization of test power can be performed by any optimizer, such as gradient-based optimizer. However, in practice, we perform test power maximization via selecting the optimal bandwidth from a list of candidates in the way of grid search (Jitkrittum et al., 2016) since optimizing the bandwidth with optimizers requires extra hyperparameter selection (e.g., learning rate and weight decay) and gradient steps which are time-consuming and may exacerbate the efficiency of the algorithm.

Generally, the bandwidth selection of kernel HSIC mainly includes two steps. Firstly, the bandwidth $\sigma$ is initialized as the median of the non-zero elements of a kernel matrix. Meanwhile, a list of coefficients, which covers as many potential values as possible, is manually set to scale the median. Then, the bandwidth selection is respectively performed for both $\mathrm{HSIC}(Z, Y)$ and $\mathrm{HSIC}(Z, Z)$ by selecting the optimal scale coefficient to generate a scaled bandwidth that is able to maximize $\frac{\mathrm{HSIC}(\cdot,\cdot;\sigma,\theta)}{\sqrt{v+\epsilon}}$. A complete learning process of MOKD is summarized in Algorithm 1 in details.

## 5 EXPERIMENTS

In this section, we evaluate our proposed MOKD method on the representative mainstream benchmark Meta-Dataset (Triantafillou et al., 2020) under several task settings in order to answer the following 4 questions: (1). Does MOKD achieve better performance on Meta-Dataset? (2). What roles do test power and $\mathrm{HSIC}(\boldsymbol{Z}, \boldsymbol{Z})$ play in our proposed MOKD? (3). Is MOKD efficient? (4). Does MOKD facilitate to alleviate the high similarity problem and further learn a better data cluster for each class?

In this paper, we follow most settings in URL (Li et al., 2021a) to train a simple linear head on top of a pre-trained ResNet-18 backbone by initializing it as an identity matrix for each adaptation episode and optimizing it with Adadelta (Zeiler, 2012). In order to validate the performance of MOKD, we compare MOKD with existing state-of-the-art approaches, including Proto-MAML (Triantafillou et al., 2020), fo-Proto-MAML (Triantafillou et al., 2020), ALFA+fo-Proto-MAML (Baik et al., 2020) CNAPS (Requeima et al., 2019), SimpleCNAPS (S-CNAPS) (Bateni et al., 2020), SUR (Dvornik et al., 2020), URT (Liu et al., 2021a), FLUTE (Triantafillou et al., 2021), Tri-M (Liu et al., 2021b), 2LM (Qin et al., 2023) and URL (Li et al., 2021a). More details are available in Appendix E and F.

### 5.1 MAIN RESULTS

In this section, we evaluate MOKD on vary-way vary-shot tasks under both 'Train on All Datasets' and 'Train on ImageNet Only' settings. To be clear, we mark seen domains with green while unseen domains with red. More details about task settings are available in Appendix E.2 and E.4.

**Train on ImageNet Only.** The results under 'Train on ImageNet Only' settings are reported in Table 1. As shown in the table, MOKD outperforms other baselines on 10 out of 13 datasets and ranks 1.4 in average. Compared with URL, which MOKD is based on, MOKD outperforms on almost all domains with the average improvement of $1.4\%$. In particular, we find that MOKD performs better on unseen domains (all datasets except ImageNet) compared with the performance on seen domains.

Table 2: **Results on Meta-Dataset (Trained on All Datasets).** Mean accuracy and 95% confidence interval are reported.

| Datasets | ProtoMAML | CNAPS | S-CNAPS | SUR | URT | Tri-M | FLUTE | 2LM | SSL-HSIC | URL | MOKD |
|---|---|---|---|---|---|---|---|---|---|---|---|
| ImageNet | 46.5± 1.1 | 50.8±1.1 | 58.4 ±1.1 | 56.2 ± 1.0 | 56.8 ± 1.1 | **58.6 ± 1.0** | 51.8 ± 1.1 | 58.0 ± 3.6 | 56.5 ± 1.2 | 57.3 ± 1.1 | 57.3 ± 1.1 |
| Omniglot | 82.7± 1.0 | 91.7±0.5 | 91.6 ± 0.6 | 94.1 ± 0.4 | 94.2 ± 0.4 | 92.0 ± 0.6 | 93.2 ± 0.5 | **95.3 ± 1.0** | 92.0 ± 0.9 | 94.1 ± 0.4 | **94.2 ± 0.5** |
| Aircraft | 75.2± 0.8 | 83.7±0.6 | 82.0 ± 0.7 | 85.5 ± 0.5 | 85.8 ± 0.5 | 82.8 ± 0.7 | 87.2 ± 0.5 | 88.2 ± 0.5 | 87.3 ± 0.7 | 88.2 ± 0.5 | **88.4 ± 0.5** |
| Birds | 69.9± 1.0 | 73.6±0.9 | 74.8 ± 0.9 | 71.0 ± 1.0 | 76.2 ± 0.8 | 75.3 ± 0.8 | 79.2 ± 0.8 | **81.8 ± 0.6** | 78.1 ± 1.1 | 80.2 ± 0.7 | 80.4 ± 0.8 |
| Textures | 68.2± 1.0 | 59.5±0.7 | 68.8 ± 0.9 | 71.0 ± 0.8 | 71.6 ± 0.7 | 71.2 ± 0.8 | 68.8 ± 0.8 | 76.3 ± 2.4 | 75.2 ± 0.8 | 76.2 ± 0.7 | **76.5 ± 0.7** |
| Quick Draw | 66.8± 0.9 | 74.7±0.8 | 76.5 ±0.8 | 81.8 ± 0.6 | **82.4 ± 0.6** | 77.3 ± 0.7 | 79.5 ± 0.7 | 78.3 ± 0.7 | 81.4 ± 0.7 | 82.2 ± 0.6 | 82.2 ± 0.6 |
| Fungi | 42.0±1.2 | 50.2±1.1 | 46.6 ± 1.0 | 64.3 ± 0.9 | 64.0 ± 1.0 | 48.5 ± 1.0 | 58.1 ± 1.1 | **69.6 ± 1.5** | 63.5 ± 1.2 | 68.7 ± 1.0 | 68.6 ± 1.0 |
| VGG Flower | 88.7± 0.7 | 88.9±0.5 | 90.5 ± 0.5 | 82.9 ± 0.8 | 87.9 ± 0.6 | 90.5 ± 0.5 | 91.6 ± 0.6 | 90.3 ± 0.8 | 90.9 ± 0.8 | 91.9 ± 0.5 | **92.5 ± 0.5** |
| Traffic Sign | 52.4 ± 1.1 | 56.5±1.1 | 57.2 ± 1.0 | 51.0 ± 1.1 | 48.2 ± 1.1 | 63.0 ± 1.0 | 58.4 ± 1.1 | 63.6 ± 1.5 | 59.7 ± 1.3 | 63.3 ± 1.2 | **64.5 ± 1.1** |
| MSCOCO | 41.7 ± 1.1 | 39.4 ±1.0 | 48.9 ± 1.1 | 52.0 ± 1.1 | 51.5 ± 1.1 | 52.8 ± 1.1 | 50.0 ± 1.0 | **57.0 ± 1.1** | 51.4 ± 1.1 | 54.2 ± 1.0 | 55.5 ± 1.0 |
| MNIST | - | - | 94.6 ± 0.4 | 94.3 ± 0.4 | 90.6 ± 0.5 | **96.2 ± 0.3** | 95.6 ± 0.5 | 94.7 ± 0.5 | 93.4 ± 0.6 | 94.7 ± 0.4 | 95.1 ± 0.4 |
| CIFAR-10 | - | - | 74.9 ± 0.7 | 66.5 ± 0.9 | 67.0 ± 0.8 | 75.4 ± 0.8 | **78.6 ± 0.7** | 71.5 ± 0.9 | 70.0 ± 1.1 | 71.9 ± 0.8 | 72.8 ± 0.8 |
| CIFAR-100 | - | - | 61.3 ± 1.1 | 56.9 ± 1.1 | 57.3 ± 1.0 | 62.0 ± 1.0 | **67.1 ± 1.0** | 60.0 ± 1.1 | 61.8 ± 1.1 | 62.9 ± 1.0 | **63.9 ±1.0** |
| Average Seen | 67.5 | 71.6 | 73.7 | 75.9 | 77.4 | 76.2 | 76.2 | 79.7 | 76.5 | 79.9 | **80.0** |
| Average Unseen | - | - | 67.4 | 64.1 | 62.9 | 69.9 | 69.9 | 69.4 | 68.2 | 69.4 | **70.3** |
| Average All | - | - | 71.2 | 71.3 | 71.8 | 73.8 | 73.8 | 75.7 | 74.6 | 75.8 | **76.3** |
| Average Rank | - | - | 7.2 | 7.3 | 6.4 | 5.2 | 5.2 | 3.4 | 5.5 | 3.1 | **2.2** |

[1] Results of URL are the average of 5 random seeds. The reproductions are consistent with the results reported on their website. The results of our method are the average of 5 random reproduction experiments. The ranks considers all 13 datasets and are calculated only with the methods in the table.

Table 3: **Comparisons of MOKD with different characteristic kernels.**

| Datasets | ImageNet | Omniglot | Aircraft | Birds | DTD | QuickDraw | Fungi | VGG_Flower | Traffic Sign | MSCOCO | MNIST | CIFAR10 | CIFAR100 |
|---|---|---|---|---|---|---|---|---|---|---|---|---|---|
| Gaussian | 57.3±1.1 | 94.2±0.5 | **88.4±0.5** | 80.4±0.8 | **76.5±0.7** | 82.2±0.6 | **68.6±1.0** | **92.5±0.5** | **64.5±1.1** | **55.5±1.0** | 95.1±0.4 | 72.8±0.8 | **63.9±1.0** |
| IMQ | 57.3±1.1 | 94.3±0.5 | 88.0±0.5 | **80.5±0.8** | 76.2±0.7 | 82.3±0.6 | 67.7±1.0 | 92.1±0.5 | 63.8±1.1 | 54.8±1.0 | **95.4±0.4** | 72.7±0.8 | 63.7±1.0 |

MOKD roughly achieves about $1.5\%$ improvements in average on unseen domains. Due to large gaps between seen and unseen domains, performing classification on unseen domains is more challenging. Such phenomenon further reveals that MOKD can generalize well on prevously unseen domains.

**Train on All Datasets.** The results under 'Train on All Datasets' settings are reported in Table 2. As shown in the table, MOKD achieves the best performance in average and ranks 2.2 among all methods. Compared with URL, which MOKD is based on, MOKD achieves better performance on 10 out of 13 datasets. Moreover, MOKD also outperforms 2LM on 8 out of 13 datasets. In addition, consistent with 'Train on ImageNet Only' settings, MOKD performs better on unseen domains, and achieves nearly $1\%$ improvements in average compared with URL. We notice that the improvements margins drop since the backbone used under 'Train on All Datasets' contains more information compared with the domain-specific backbones used under 'Train on ImageNet Only' settings.

We also compare MOKD with SSL-HSIC. To be fair, we estimate HSIC measures in SSL-HSIC with the same way as MOKD and use the same $\gamma$. According to the results reported in Table 1 and 2, MOKD outperforms SSL-HSIC under both settings. Complete analysis is available in Appendix D.

### 5.2 EXPERIMENTAL ANALYSIS

**Kernel Type.** HSIC is a valid statistical measure with characteristic kernels (e.g., Gaussian kernel). To study the effect of kernels, we further run MOKD with inverse multiquadric kernel (IMQ) under 'Train on All Datasets' settings with the same random seeds. Since linear kernels, such as cosine similarity, are not characteristic kernels, we do not consider them in this study. The results are reported in Table 3. According to the table, it is easy to observe that MOKD with Gaussian kernel totally achieves better generalization performance than MOKD with IMQ.

**Running Time.** To discuss the efficiency of MOKD, we assume that the number of data is $m$, the length of bandwidth list is $k$ and the number of adaptation steps is $s$. Then, for each task, the time complexity can be roughly expressed as $(4k + 2s)\mathcal{O}(m^2)$ according to Algorithm 1. The experimental results are reported in Fig. 2 and Table 8. We run the experiment on the same RTX 3090 GPU with the same seeds for fairness. According to the results, we find that the time which MOKD consumes for each adaptation step is acceptable. In some cases, such as datasets like Omniglot and Fungi,

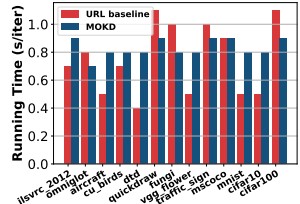

Figure 2: Comparison of Running time between URL and MOKD on Meta-Dataset.

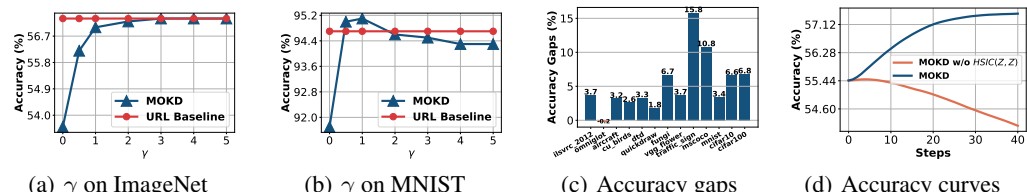

(a) $\gamma$ on ImageNet    (b) $\gamma$ on MNIST    (c) Accuracy gaps    (d) Accuracy curves

Figure 3: **Quantitative analysis of $\gamma$. (a).** Effect of $\gamma$ on accuracy of ImageNet dataset; **(b).** Effect of $\gamma$ on accuracy of MNIST dataset; **(c).** Performance gaps between MOKD with and without HSIC($\boldsymbol{Z}, \boldsymbol{Z}$); **(d).** Test accuracy curves of MOKD with and without HSIC($\boldsymbol{Z}, \boldsymbol{Z}$) on ImageNet.

MOKD is even more efficient compared with URL. Thus, although MOKD is a bi-level optimization algorithm which includes HSIC and variance estimations, the algorithm is still efficient in total.

**Analysis of $\gamma$.** In our work, $\gamma$ functions as a coefficient of regularization term HSIC($\boldsymbol{Z}, \boldsymbol{Z}$). Since HSIC($\boldsymbol{Z}, \boldsymbol{Z}$) mainly facilitates to penalize high-variance representations and remove common information shared among samples, $\gamma$ intuitively determines the power of penalization and suppression imposed to high-variance representations and common information shared across samples. In order to figure out how datasets in Meta-Dataset react to $\gamma$, we run MOKD with different $\gamma$ values under 'Train on all datasets' settings. Complete discussions and results are reported in Appendix G.3.

According to reported results, a general conclusion we can summarize is that different datasets achieve their best performance with different $\gamma$ values. Sepcifically, complicated datasets, such as ImageNet (3(a)), prefer large $\gamma$ while simple datasets, such as MNIST (Fig. 3(b)), prefer small $\gamma$. For complicated datasets, due to the existence of abundant objects and semantic information, large $\gamma$ is required to simultaneously drive model to learn discriminative features for each sample and penalize high-variance representations to avoid overfitting. In this way, samples tend to be 'independent' to each other and trivial common information, which is shared across samples and may further results in high similarities among samples, is removed. In contrast, since simple datasets, such as Omniglot and MNIST, owns evident and definite semantic areas, small $\gamma$ is enough to achieve good performance.

In addition, when $\gamma$ is set to 0, it is equivalent to perform an ablation study on HSIC($\boldsymbol{Z}, \boldsymbol{Z}$). According to Fig. 3(c), HSIC($\boldsymbol{Z}, \boldsymbol{Z}$) plays an important role in achieving good generalization performance. When HSIC($\boldsymbol{Z}, \boldsymbol{Z}$) is removed, the performance drops significantly. According to Fig. 3(d) and 6, it is easy to find that the reason of performance drop is that overfitting happens. Thus, these results demonstrate that HSIC($\boldsymbol{Z}, \boldsymbol{Z}$) helps alleviate overfitting and improve the generalization performance.

**Ablation Study: Test Power.** Fig. 4 shows the performance gaps between MOKD with and without test power maximization (TPM) on Meta-dataset benchmark. As shown in the figure, the gaps are evident and MOKD with test power maximization performs better. Besides, we also notice that MOKD with TPM performs better on unseen domains and on those complicated datasets, such as Fungi.

**Visualization Results.** Fig. 8 and 9 show the heatmaps of the similarity matrices of support data features respectively learned with NCC-based loss and our porposed MOKD. As shown in Fig. 9(a), MOKD learns more definite and clear data clusters compared with those learned with NCC-based loss (URL), which demonstrates that

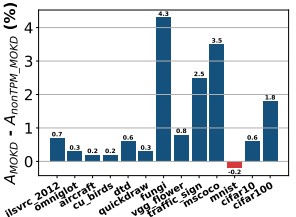

Figure 4: Acc. gaps between MOKD w. and w/o. TPM.

MOKD facilitates to capture the cluster structures of the given support set and learn better class-specific features. Meanwhile, both Fig. 1(b) and Fig. 9(c) further reveal that MOKD is able to alleviate the undesirable high similarities between two samples belonging to different classes.

## 6 CONCLUSION

In this paper, we propose an efficient bi-level framework, maximizing optimized kernel dependence, to perform classification on cross-domain few-shot tasks. Specifically, MOKD first maximizes the test power of kernel HSIC to maximize its sensitivity to dependence, and then optimizes the dependence directly with the optimized kernel HSIC. Extensive experimental results on Meta-Dataset benchmark demonstrate that MOKD can not only achieve better performance but also learn better clusters for each class and alleviate the undesirable high-similarities between samples from different classes.

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

## A  FUTURE WORK

In this paper, we mainly focus on learning better class-specific representations, where similarities among samples within the same class are maximized while similarities between samples from different classes are minimized, for cross-domain few-shot classification tasks via optimizing optimized kernel dependence, where test power is maximized, between two sets of data (representation and label, representation and representation). Specifically, we try to maximize the dependence between representations and labels in order to learn a set of representations that matches the cluster structure in the given task while minimizing dependence among samples in order to remove trivial common information shared among samples and penalize high-variance representations.

Following previous works (Triantafillou et al., 2020; Dvornik et al., 2020; Liu et al., 2021a; Li et al., 2021a), we only discuss about cross-domain few-shot classification problem in close-set scenarios. Since one goal of our work is to learn to classify data from previous unseen domains, it will be a good direction to extend our work to open set recognition (Dhamija et al., 2018; Yang et al., 2020; Miller et al., 2021; Cevikalp et al., 2023), where NCC-based loss is also used as loss (Yang et al., 2020). In this topic, since cosine similarity kernel adopted in previous works (Dvornik et al., 2020; Liu et al., 2021a; Li et al., 2021a) may confine the features on the outer shell of a hypersphere, it is not believed as a suitable kernel in this area. Thus, it will be interesting to further discuss about extending our method to open set recognition area from the perspective of kernels.

## B  MORE RELATED WORK

**Cross-domain Few-shot Classification.** Cross-domain few-shot classification aims to perform classification on tasks that are sampled from not only the previously unseen data with the same distribution of seen domains but also previously unseen domains with different distributions. Compared with conventional few-shot classification (Vinyals et al., 2016; Ravi & Larochelle, 2017; Finn et al., 2017; Snell et al., 2017; Nichol et al., 2018; Tian et al., 2020), CFC is much more challenging mainly for two aspects. First of all, the distribution gaps between source and target domains sometimes are quite large. For example, the data distribution on ImageNet is quite different from that of Omniglot. Thus, the model will fail to achieve good generalization performance if it was pre-trained on Omniglot dataset (Lake et al., 2015) but evaluated on ImageNet (Russakovsky et al., 2015) since the feature patterns of ImageNet has never been observed by the model. On the other side, the task settings of CFC are much more difficult. For example, vary-way vary-shot setting is proposed in Triantafillou et al. (2020) as the benchmark for later works. For a given task, the number of classes and the number of shots for each class are randomly sampled from given intervals at the beginning of each episode.

Generally, although all existing cross-domain few-shot classification methods are performed in the way of typical meta-learning Finn et al. (2017); Snell et al. (2017), they can be mainly divided into two genres according to the ways of training feature encoders. One of the genres trains the feature encoder and the classifier from scratch. For example, Triantafillou et al. (2020) proposes Proto-MAML which combines Prototypical Nets (Snell et al., 2017) and MAML (Finn et al., 2017) by treating the prototypes obtained from the feature encoder as the parameters of the linear classification layer to perform cross-domain few-shot classification. Based on Proto-MAML, ALFA+fo-Proto-MAML (Baik et al., 2020) further proposes to adaptively generate task-specific hyperparameters, such as learning rate and weight decay, from a small model for each task. Besides, CNAPS (Requeima et al., 2019) proposes to leverage the FiLM (Perez et al., 2018) module to adapt the parameters of both feature encoder and classifier to new tasks. Further, SimpleCNAPS proposes to substitute the parametric classifier used in CNAPS with a class-covariance-based distance metric to improve the efficiency and generalization performance with less parameters. Currently, as far as we know, the state-of-the-art method in this genre is CrossTransformer (Doersch et al., 2020) which proposes to learn spatial correspondence with an self-attention module from tasks sampled from ImageNet and generalize the prior knowledge to other unseen domains.

The other genre in cross-domain few-shot classification aims to transfer the prior knowledge of a pre-trained backbone to tasks sampled from previously unseen domains by further training or fine-tuning a module on top of the frozen backbone. An intuition behind these methods is that a pre-trained backbone is able to extract good representations from the given new tasks sampled from previously unseen domains since it has been well trained on some large datasets, such as

ImageNet (Russakovsky et al., 2015)). Thus, the semantic feature space embedded in the pre-trained model can partially or fully cover the feature space of target domains and recognize as many feature patterns as possible. To be specific, SUR (Dvornik et al., 2020) proposes to learn a weight vector to select features from several pre-trained backbones and combine these features in a linear way with corresponding weights. Later, URT (Liu et al., 2021a) proposes to train a Universal Representation Transformer (Vaswani et al., 2017) model to select features from the embeddings extracted from 8 pre-trained domain-specific backbones. Besides, FLUTE (Triantafillou et al., 2021) proposes to treat the convolutional layers of a model as universal templates which will be frozen during test phase while train specific batch normalization layers for each of the 8 seen domain datasets. Then, a 'Blender' model is trained to learn to combine the statistical information from all specific BN layers. During meta-test, a new specific BN is generated by feeding the support data in the Blender model. However, since the forward passes of several backbones consumes too much time during test phase, Li et al. (2021a) proposes URL to learn a universal multi-domain backbone from the 8 pre-trained backbones via knowledge distillation. During test phase, URL fine-tunes a simple linear transformation head on top of the distilled backbone for adaptation of the new tasks. As a further version of URL, TSA (Li et al., 2022), which is the state-of-the-art method in this genre, plugs additional learning modules into the backbone so that more task-specific features can be learned.

**Feature Selection via Dependence Measures** Dependence measure has been well studied in statistics, and a series of measures has been proposed in previous works, such as constrained covariance (COCO) (Gretton et al., 2005b), kernel canonical correlation (Bach & Jordan, 2002) and Hilbert-Schmidt Independence Criterion (HSIC) (Gretton et al., 2005a). Recently, these tools, especially HSIC, have been widely applied in deep learning tasks in order to learning desirable features. Song et al. (2012) firstly introduces the framework for feature selection based on dependence maximization between the selected features and the labels of an estimation problem. Kumagai et al. (2022) follows previous works (Yamada et al., 2014; Freidling et al., 2021; Koyama et al., 2022) and propose to select relevant features and remove the redundant features by solving an $\ell_1$-regularized regression problem. Besides, Li et al. (2021b) proposes to replace InfoNCE (Oord et al., 2018) with SSL-HSIC to directly optimize statistical dependence without restrictive data assumptions or indirect mutual information estimators.

## C  Proof Results

### C.1  Proof of Theorem 1

**Proof 1** *We first reformulate the NCC-based loss in Eq. (1) as $-\mathbb{E}_{(\boldsymbol{X},Y)\sim\mathbb{U}_{\mathcal{D}}}\log(p(Y=\hat{y}|\boldsymbol{X},\theta))$ where $(\boldsymbol{X},Y)\sim\mathbb{U}_{\mathcal{D}}$ represent a random variable $(\boldsymbol{X},Y)$ whose distribution is a uniform distribution defined on a dataset $\mathcal{D}$. Let $\boldsymbol{Z}'$ be the independent copies of $\boldsymbol{Z}$, with the corresponding support representations $\mathcal{Z}=\{\boldsymbol{z}_i\}_{i=1}^{|\mathcal{D}|}=\{h_\theta\circ f_{\phi^*}(\boldsymbol{x}_i)\}_{i=1}^{|\mathcal{D}|}$ and uniform distributions $\mathbb{U}_{\mathcal{Z}}$ and $\mathbb{U}_{\mathcal{D}_{\mathcal{Z}}}$ respectively defined on $\mathcal{Z}$ and support representation label pairs, we begin with:*

$$-\mathbb{E}_{(\boldsymbol{X},Y)\sim\mathbb{U}_{\mathcal{D}}}\log(p(Y=\hat{y}|\boldsymbol{X},\theta)) = -\mathbb{E}_{(\boldsymbol{Z},Y)\sim\mathbb{U}_{\mathcal{D}_{\mathcal{Z}}}}\log(p(Y=\hat{y}|\boldsymbol{Z},\theta))$$

$$= \mathbb{E}_{\boldsymbol{Z}\sim\mathbb{U}_{\mathcal{Z}}}\left[\log\sum_{i=1}^{N_C}e^{k(\boldsymbol{Z},\boldsymbol{c}_i)}-\left(k(\boldsymbol{Z},\boldsymbol{c})\right)\right]$$

$$= \mathbb{E}_{\boldsymbol{Z}\sim\mathbb{U}_{\mathcal{Z}}}\left[\log\sum_{i=1}^{N_C}e^{\frac{1}{|\mathcal{C}_i|}\sum_{\boldsymbol{z}'\in\mathcal{C}_i}k(\boldsymbol{Z},\boldsymbol{z}')}-\left(\frac{1}{|\mathcal{C}|}\sum_{\boldsymbol{z}''\in\mathcal{C}}k(\boldsymbol{Z},\boldsymbol{z}'')\right)\right]$$

$$\leq \mathbb{E}_{\boldsymbol{Z}\sim\mathbb{U}_{\mathcal{Z}}}\left[\log\sum_{i=1}^{N_C}\frac{1}{|\mathcal{C}_i|}\sum_{\boldsymbol{z}'\in\mathcal{C}_i}e^{k(\boldsymbol{Z},\boldsymbol{z}')}-\mathbb{E}_{\boldsymbol{Z}''\sim\mathbb{U}_{\mathcal{C}}}\left[k(\boldsymbol{Z},\boldsymbol{Z}'')\right]\right]$$

$$= \mathbb{E}_{\boldsymbol{Z}\sim\mathbb{U}_{\mathcal{Z}}}\log\sum_{i=1}^{N_C}\frac{1}{|\mathcal{C}_i|}\sum_{\boldsymbol{z}'\in\mathcal{C}_i}e^{k(\boldsymbol{Z},\boldsymbol{z}')}-\mathbb{E}_{\boldsymbol{Z}\sim\mathbb{U}_{\mathcal{Z}}}\mathbb{E}_{\boldsymbol{Z}''\sim\mathbb{U}_{\mathcal{C}}}k(\boldsymbol{Z},\boldsymbol{Z}''),$$

$$(9)$$

where $\boldsymbol{c}$ denote the class centroid representations, $\boldsymbol{c}_i$ denotes the $i$-th class centroid, $\mathcal{C} \subset \mathcal{Z}$ denotes a support representation subset where all samples belong to the same class. The fourth line is obtained with Jensen Inequality.

Then, since each data point belonging to class set $\mathcal{C}_i$ in given support representation set $\mathcal{Z}$ is randomly sampled with the probability $\frac{1}{N_C \cdot |\mathcal{C}_i|}$, for the first term of Eq. (9), we have:

$$
\begin{aligned}
\mathbb{E}_{\boldsymbol{Z} \sim \mathbb{U}_{\mathcal{Z}}} \log \sum_{i=1}^{N_C} \frac{1}{|\mathcal{C}_i|} \sum_{\boldsymbol{z}' \in \mathcal{C}_i} e^{k(\boldsymbol{Z}, \boldsymbol{z}')} &= \mathbb{E}_{\boldsymbol{Z} \sim \mathbb{U}_{\mathcal{Z}}} \log (N_C \cdot \frac{1}{N_C}) \cdot \sum_{i=1}^{N_C} \frac{1}{|\mathcal{C}_i|} \sum_{\boldsymbol{z}' \in \mathcal{C}_i} e^{k(\boldsymbol{Z}, \boldsymbol{z}')} \\
&= \mathbb{E}_{\boldsymbol{Z} \sim \mathbb{U}_{\mathcal{Z}}} \log \sum_{i=1}^{N_C} \sum_{\boldsymbol{z}' \in \mathcal{C}_i} \frac{1}{N_C \cdot |\mathcal{C}_i|} e^{k(\boldsymbol{Z}, \boldsymbol{z}')} + \log N_C \\
&= \mathbb{E}_{\boldsymbol{Z} \sim \mathbb{U}_{\mathcal{Z}}} \log \mathbb{E}_{\boldsymbol{Z}' \sim \mathbb{U}_{\mathcal{Z}}} e^{k(\boldsymbol{Z}, \boldsymbol{z}')} + \log N_C.
\end{aligned}
\tag{10}
$$

Thus, by combining Eq. (9) and Eq. (10), we obtain an upper bound for the NCC-based loss:

$$
-\mathbb{E}_{(\boldsymbol{X}, Y) \sim \mathcal{D}} \log(p(Y = \hat{y} | \boldsymbol{X}, \theta)) \leq -\mathbb{E}_{\boldsymbol{Z} \sim \mathbb{U}_{\mathcal{Z}}} \mathbb{E}_{\boldsymbol{Z}'' \sim \mathbb{U}_{\mathcal{C}}} k(\boldsymbol{Z}, \boldsymbol{Z}'') + \mathbb{E}_{\boldsymbol{Z} \sim \mathbb{U}_{\mathcal{Z}}} \log \mathbb{E}_{\boldsymbol{Z}' \sim \mathbb{U}_{\mathcal{Z}}} e^{k(\boldsymbol{Z}, \boldsymbol{Z}')} + \log N_C.
$$

## C.2 PROOF OF THEOREM 3

**Proof 2** *Following Li et al. (2021b), we compute HSIC by directly calculating the three terms in Eq. (3) respectively. Given a set of support representations $\mathcal{Z} = \{\boldsymbol{z}_i\}_{i=1}^{|\mathcal{Z}|} = \{h_\theta \circ f_{\phi^*}(\boldsymbol{x}_i)\}_{i=1}^{|\mathcal{Z}|}$, a uniform distribution $\mathbb{U}_{\mathcal{Z}}$ defined on $\mathcal{Z}$, and let $\boldsymbol{Z}'$ be the independent copy of $\boldsymbol{Z}$ and $Y', Y''$ be the independent copies of $Y$, for the first term, we can obtain*

$$
\begin{aligned}
\mathbb{E}\left[k(\boldsymbol{Z}, \boldsymbol{Z}') l(Y, Y')\right] &= \mathbb{E}_{\boldsymbol{Z}, \boldsymbol{Z}', Y, Y'}\left[k(\boldsymbol{Z}, \boldsymbol{Z}') l(Y, Y')\right] \\
&= \Delta l \mathbb{E}_{\boldsymbol{Z}, \boldsymbol{Z}', Y, Y'}\left[k(\boldsymbol{Z}, \boldsymbol{Z}') \mathbb{I}(Y, Y')\right] + l_0 \mathbb{E}_{\boldsymbol{Z}, \boldsymbol{Z}'}\left[k(\boldsymbol{Z}, \boldsymbol{Z}')\right] \\
&= \Delta l \sum_{i=1}^{|\mathcal{Z}|} \sum_{j=1}^{|\mathcal{Z}|} \mathbb{E}_{\boldsymbol{Z}|y_i, \boldsymbol{Z}'|y_j}\left[\frac{1}{N_C |\mathcal{C}_{y_i}|} \cdot \frac{1}{N_C |\mathcal{C}_{y_j}|} \cdot k(\boldsymbol{Z}, \boldsymbol{Z}') \mathbb{I}(y_i = y_j)\right] + l_0 \mathbb{E}_{\boldsymbol{Z} \sim \mathbb{U}_{\mathcal{Z}}, \boldsymbol{Z}' \sim \mathbb{U}_{\mathcal{Z}}}\left[k(\boldsymbol{Z}, \boldsymbol{Z}')\right] \\
&= \Delta l \sum_{i=1}^{|\mathcal{Z}|} \mathbb{E}_{\boldsymbol{Z}|y_i}\left[\frac{1}{N_C |\mathcal{C}_{y_i}|} \cdot \frac{1}{N_C} \cdot \frac{1}{|\mathcal{C}_{y_i}|} \sum_{\boldsymbol{Z}'' \in \{\boldsymbol{Z}''|y_j = y_i\}} k(\boldsymbol{Z}, \boldsymbol{Z}'')\right] + l_0 \mathbb{E}_{\boldsymbol{Z} \sim \mathbb{U}_{\mathcal{Z}}, \boldsymbol{Z}' \sim \mathbb{U}_{\mathcal{Z}}}\left[k(\boldsymbol{Z}, \boldsymbol{Z}')\right] \\
&= \frac{\Delta l}{N_C} \sum_{i=1}^{|\mathcal{Z}|} \frac{1}{N_C |\mathcal{C}_{y_i}|} \mathbb{E}_{\boldsymbol{Z}|y_i}\left[\frac{1}{|\mathcal{C}_{y_i}|} \sum_{\boldsymbol{Z}'' \in \{\boldsymbol{Z}''|y_j = y_i\}} k(\boldsymbol{Z}, \boldsymbol{Z}'')\right] + l_0 \mathbb{E}_{\boldsymbol{Z} \sim \mathbb{U}_{\mathcal{Z}}, \boldsymbol{Z}' \sim \mathbb{U}_{\mathcal{Z}}}\left[k(\boldsymbol{Z}, \boldsymbol{Z}')\right] \\
&= \frac{\Delta l}{N_C} \mathbb{E}_{\boldsymbol{Z} \sim \mathbb{U}_{\mathcal{Z}}} \mathbb{E}_{\boldsymbol{Z}'' \sim \mathbb{U}_{\mathcal{C}}}\left[k(\boldsymbol{Z}, \boldsymbol{Z}'')\right] + l_0 \mathbb{E}_{\boldsymbol{Z} \sim \mathbb{U}_{\mathcal{Z}}, \boldsymbol{Z}' \sim \mathbb{U}_{\mathcal{Z}}}\left[k(\boldsymbol{Z}, \boldsymbol{Z}')\right],
\end{aligned}
$$

where $\mathcal{C}_y$ denotes the class set that contains the data $\{\boldsymbol{Z} | Y = y\}$.

Then, due to the independence between $\boldsymbol{Z}'$ and $Y''$, the second term can be calculated as:

$$
\begin{aligned}
\mathbb{E}\left[k(\boldsymbol{Z}, \boldsymbol{Z}') l(Y, Y'')\right] &= \mathbb{E}_{\boldsymbol{Z} Y}\left[\mathbb{E}_{\boldsymbol{Z}'}[k(\boldsymbol{Z}, \boldsymbol{Z}')] \mathbb{E}_{Y''}[l(Y, Y'')]\right] \\
&= \mathbb{E}_{\boldsymbol{Z} \sim \mathbb{U}_{\mathcal{Z}}, \boldsymbol{Z}' \sim \mathbb{U}_{\mathcal{Z}}}\left[k(\boldsymbol{Z}, \boldsymbol{Z}')\right] \sum_{i=1}^{|\mathcal{Z}|} \frac{1}{N_C |\mathcal{C}_{y_i}|} \left(\sum_{j=1}^{|\mathcal{Z}|} \frac{1}{N_C |\mathcal{C}_{y_j}|} \cdot \Delta l \mathbb{I}(y_i = y_j) + l_0\right) \\
&= \left(\sum_{i=1}^{|\mathcal{Z}|} \frac{1}{N_C |\mathcal{C}_{y_i}|} \cdot \frac{\Delta l}{N_C} + l_0\right) \mathbb{E}_{\boldsymbol{Z} \sim \mathbb{U}_{\mathcal{Z}}, \boldsymbol{Z}' \sim \mathbb{U}_{\mathcal{Z}}}\left[k(\boldsymbol{Z}, \boldsymbol{Z}')\right] \\
&= \left(\frac{\Delta l}{N_C} + l_0\right) \mathbb{E}_{\boldsymbol{Z} \sim \mathbb{U}_{\mathcal{Z}}, \boldsymbol{Z}' \sim \mathbb{U}_{\mathcal{Z}}}\left[k(\boldsymbol{Z}, \boldsymbol{Z}')\right].
\end{aligned}
$$

*For the third part, we can obtain:*

$$
\mathbb{E}\left[k(\boldsymbol{Z}, \boldsymbol{Z}^{'})\right] \mathbb{E}\left[l(Y, Y^{'})\right] = \mathbb{E}_{\boldsymbol{Z}, \boldsymbol{Z}^{'}}\left[k(\boldsymbol{Z}, \boldsymbol{Z}^{'})\right] \mathbb{E}_{Y, Y^{'}}\left[l(Y, Y^{'})\right]
$$

$$
= \mathbb{E}_{\boldsymbol{Z} \sim \mathbb{U}_{\mathcal{Z}}, \boldsymbol{Z}' \sim \mathbb{U}_{\mathcal{Z}}}\left[k(\boldsymbol{Z}, \boldsymbol{Z}^{'})\right] \left(\sum_{i=1}^{|\mathcal{Z}|} \sum_{j=1}^{|\mathcal{Z}|} \frac{1}{N_C |\mathcal{C}_{y_i}|} \cdot \frac{1}{N_C |\mathcal{C}_{y_j}|} \cdot \Delta l \mathbb{I}(y_i = y_j) + l_0\right)
$$

$$
= \mathbb{E}_{\boldsymbol{Z} \sim \mathbb{U}_{\mathcal{Z}}, \boldsymbol{Z}' \sim \mathbb{U}_{\mathcal{Z}}}\left[k(\boldsymbol{Z}, \boldsymbol{Z}^{'})\right] \left(\sum_{i=1}^{|\mathcal{Z}|} \frac{1}{N_C |\mathcal{C}_{y_i}|} \cdot \frac{\Delta l}{N_C} + l_0\right)
$$

$$
= \left(\frac{\Delta l}{N_C} + l_0\right) \mathbb{E}_{\boldsymbol{Z} \sim \mathbb{U}_{\mathcal{Z}}, \boldsymbol{Z}' \sim \mathbb{U}_{\mathcal{Z}}}\left[k(\boldsymbol{Z}, \boldsymbol{Z}^{'})\right]
$$

*Thus,* $\mathrm{HSIC}(\boldsymbol{Z}, Y)$ *can be specifically reformulated as:*

$$
\mathrm{HSIC}(\boldsymbol{Z}, Y) = \mathbb{E}[k(\boldsymbol{Z}, \boldsymbol{Z}^{'})l(Y, Y^{'})] - 2\mathbb{E}[k(\boldsymbol{Z}, \boldsymbol{Z}^{'})l(Y, Y^{''})] + \mathbb{E}[k(\boldsymbol{Z}, \boldsymbol{Z}^{'})]\mathbb{E}[l(Y, Y^{'})]
$$

$$
= \frac{\Delta l}{N_C} \left(\mathbb{E}_{\boldsymbol{Z} \sim \mathbb{U}_{\mathcal{Z}}} \mathbb{E}_{\boldsymbol{Z}^{''} \sim \mathbb{U}_{\mathcal{C}}}\left[k(\boldsymbol{Z}, \boldsymbol{Z}^{''})\right] - \mathbb{E}_{\boldsymbol{Z} \sim \mathbb{U}_{\mathcal{Z}}, \boldsymbol{Z}' \sim \mathbb{U}_{\mathcal{Z}}}\left[k(\boldsymbol{Z}, \boldsymbol{Z}^{'})\right]\right)
$$

### C.3 PROOF OF THEOREM 4

**Proof 3** *We first perform Taylor expansion on* $\mathbb{E}_{\boldsymbol{Z} \sim \mathbb{U}_{\mathcal{Z}}} \log \mathbb{E}_{\boldsymbol{Z}' \sim \mathbb{U}_{\mathcal{Z}}} e^{k(\boldsymbol{Z}, \boldsymbol{Z}^{'})}$ *around* $\mu \equiv \mathbb{E}_{\boldsymbol{Z}' \sim \mathbb{U}_{\mathcal{Z}}}\left[k(\boldsymbol{Z}, \boldsymbol{Z}^{'})\right]$, *for* $k(\boldsymbol{Z}, \boldsymbol{Z}^{'}) \approx \mu$, *we have:*

$$
\mathbb{E}_{\boldsymbol{Z} \sim \mathbb{U}_{\mathcal{Z}}} \log \mathbb{E}_{\boldsymbol{Z}' \sim \mathbb{U}_{\mathcal{Z}}}\left[e^{k(\boldsymbol{Z}, \boldsymbol{Z}^{'})}\right] = \mathbb{E}_{\boldsymbol{Z} \sim \mathbb{U}_{\mathcal{Z}}} \log \mathbb{E}_{\boldsymbol{Z}' \sim \mathbb{U}_{\mathcal{Z}}}\left[e^{\mu}(1 + k(\boldsymbol{Z}, \boldsymbol{Z}^{'}) - \mu + \frac{(k(\boldsymbol{Z}, \boldsymbol{Z}^{'}) - \mu)^2}{2})\right]
$$

$$
= \mathbb{E}_{\boldsymbol{Z} \sim \mathbb{U}_{\mathcal{Z}}}(\mu) + \mathbb{E}_{\boldsymbol{Z} \sim \mathbb{U}_{\mathcal{Z}}} \log \mathbb{E}_{\boldsymbol{Z}' \sim \mathbb{U}_{\mathcal{Z}}}\left[1 + \frac{(k(\boldsymbol{Z}, \boldsymbol{Z}^{'}) - \mu)^2}{2})\right]
$$

$$
\leq \mathbb{E}_{\boldsymbol{Z} \sim \mathbb{U}_{\mathcal{Z}}}(\mu) + \mathbb{E}_{\boldsymbol{Z} \sim \mathbb{U}_{\mathcal{Z}}} \mathbb{E}_{\boldsymbol{Z}' \sim \mathbb{U}_{\mathcal{Z}}} \log\left[1 + \frac{(k(\boldsymbol{Z}, \boldsymbol{Z}^{'}) - \mu)^2}{2})\right]
$$

*Now expanding* $\log(1 + x)$ *around zero:*

$$
\mathbb{E}_{\boldsymbol{Z} \sim \mathbb{U}_{\mathcal{Z}}} \log \mathbb{E}_{\boldsymbol{Z}' \sim \mathbb{U}_{\mathcal{Z}}}[e^{k(\boldsymbol{Z}, \boldsymbol{Z}^{'})}] \leq \mathbb{E}_{\boldsymbol{Z} \sim \mathbb{U}_{\mathcal{Z}}}(\mu) + \mathbb{E}_{\boldsymbol{Z} \sim \mathbb{U}_{\mathcal{Z}}} \mathbb{E}_{\boldsymbol{Z}' \sim \mathbb{U}_{\mathcal{Z}}} \log\left[1 + \frac{\left(k(\boldsymbol{Z}, \boldsymbol{Z}^{'}) - \mu\right)^2}{2}\right]
$$

$$
= \mathbb{E}_{\boldsymbol{Z} \sim \mathbb{U}_{\mathcal{Z}}}(\mu) + \mathbb{E}_{\boldsymbol{Z} \sim \mathbb{U}_{\mathcal{Z}}} \mathbb{E}_{\boldsymbol{Z}' \sim \mathbb{U}_{\mathcal{Z}}}\left[\frac{\left(k(\boldsymbol{Z}, \boldsymbol{Z}^{'}) - \mu\right)^2}{2}\right]
$$

$$
= \mathbb{E}_{\boldsymbol{Z} \sim \mathbb{U}_{\mathcal{Z}}} \mathbb{E}_{\boldsymbol{Z}' \sim \mathbb{U}_{\mathcal{Z}}}\left[k(\boldsymbol{Z}, \boldsymbol{Z}^{'})\right] + \frac{1}{2}\mathbb{E}_{\boldsymbol{Z} \sim \mathbb{U}_{\mathcal{Z}}}\left[\mathbb{V}\mathrm{ar}_{\boldsymbol{Z}' \sim \mathbb{U}_{\mathcal{Z}}}\left[k(\boldsymbol{Z}, \boldsymbol{Z}^{'})\right]\right].
$$

*Thus, with the assumption that* $\Delta l = N_C$, *we can reformulate NCC-based loss as:*

$$
\mathcal{L}_{\mathrm{NCC}} \leq -\mathbb{E}_{\boldsymbol{Z} \sim \mathbb{U}_{\mathcal{Z}}} \mathbb{E}_{\boldsymbol{Z}^{''} \sim \mathbb{U}_{\mathcal{C}}}\left[k(\boldsymbol{Z}, \boldsymbol{Z}^{''})\right] + \mathbb{E}_{\boldsymbol{Z} \sim \mathbb{U}_{\mathcal{Z}}, \boldsymbol{Z}' \sim \mathbb{U}_{\mathcal{Z}}}\left[k(\boldsymbol{Z}, \boldsymbol{Z}^{'})\right] + \frac{1}{2}\mathbb{E}_{\boldsymbol{Z} \sim \mathbb{U}_{\mathcal{Z}}}\left[\mathbb{V}\mathrm{ar}_{\boldsymbol{Z}' \sim \mathbb{U}_{\mathcal{Z}}}\left[k(\boldsymbol{Z}, \boldsymbol{Z}^{'})\right]\right] + \log N_C
$$

$$
= -\mathrm{HSIC}(\boldsymbol{Z}, Y) + \frac{1}{2}\mathbb{E}_{\boldsymbol{Z} \sim \mathbb{U}_{\mathcal{Z}}}\left[\mathbb{V}\mathrm{ar}_{\boldsymbol{Z}' \sim \mathbb{U}_{\mathcal{Z}}}\left[k(\boldsymbol{Z}, \boldsymbol{Z}^{'})\right]\right] + \log N_C.
$$

## D DIFFERENCES BETWEEN SSL-HSIC AND MOKD

In this paper, we propose a bi-level optimization framework MOKD, which is built on a new interpretation of NCC-based loss from the perspective of kernel dependence measure, in order to

---

**Algorithm 1** Maximizing Optimized Kernel Dependence Algorithm

---

**Input:** pre-trained backbone $f_{\phi^*}$, number of inner iterations $n$, learning rate $\eta$, linear transformation parameters $h_\theta$, a list of bandwidths $\Sigma = \{\sigma_1, \sigma_2, ..., \sigma_T\}$, and $\epsilon = 1e - 5$.
**Output:** the optimal parameters for linear transformation head $\theta^*$.
*# Sample a task*
**Sample** a new task $\mathcal{T} = \{\{\boldsymbol{X}^{\mathrm{s}}, Y^{\mathrm{s}}\}, \{\boldsymbol{X}^{\mathrm{q}}, Y^{\mathrm{q}}\}\}$;
**Obtain** the representations: $\mathcal{Z} = \{h_\theta \circ f_{\phi^*}(\boldsymbol{x}_i)\}_{i=1}^{|\boldsymbol{X}^{\mathrm{s}}|}$
*# Inner optimization for test power maximization*
**Maximize** the test power of $\widehat{\mathrm{HSIC}}(\boldsymbol{Z}, Y; \sigma_{\boldsymbol{ZY}}, \theta)$ and $\widehat{\mathrm{HSIC}}(\boldsymbol{Z}, \boldsymbol{Z}; \sigma_{\boldsymbol{ZZ}}, \theta)$ with Eq. (6) and (7):
$$\sigma^*_{\boldsymbol{ZY}} = \max_\Sigma \frac{\widehat{\mathrm{HSIC}}(\boldsymbol{Z}, Y; \sigma_{\boldsymbol{ZY}}, \theta)}{\sqrt{v_{ZY} + \epsilon}}; \sigma^*_{\boldsymbol{ZZ}} = \max_\Sigma \frac{\widehat{\mathrm{HSIC}}(\boldsymbol{Z}, \boldsymbol{Z}; \sigma_{\boldsymbol{ZZ}}, \theta)}{\sqrt{v_{ZZ} + \epsilon}}$$
*# Outer optimization for dependence optimization*
**for** $i = 1$ **to** $n$ **do**
    **Compute** $\widehat{\mathrm{HSIC}}(\boldsymbol{Z}, Y, \sigma^*_{\boldsymbol{ZY}}, \theta)$ and $\widehat{\mathrm{HSIC}}(\boldsymbol{Z}, \boldsymbol{Z}; \sigma^*_{\boldsymbol{ZZ}}, \theta)$ with Eq. (6) for loss:
        $\mathcal{L}(\boldsymbol{Z}, Y; \theta) = -\widehat{\mathrm{HSIC}}(\boldsymbol{Z}, Y, \sigma^*_{\boldsymbol{ZY}}, \theta) + \gamma \widehat{\mathrm{HSIC}}(\boldsymbol{Z}, \boldsymbol{Z}; \sigma^*_{\boldsymbol{ZZ}}, \theta)$
    **Update** parameters:
        $\theta \leftarrow \theta - \eta \nabla_\theta \mathcal{L}(\boldsymbol{Z}, Y; \theta)$
**end for**

---

learn a set of class-specific representations, where the similarities among samples belonging to the same class are maximized while the similarties between samples from different classes are minimized, by optimizing the dependence respective between representations and labels and among all representations based on the optimized HSIC measures where the test power of the kernels used are maximized.

However, we notice that the outer optimization objective in Eq. (5) shares the similar format as the objective used in SSL-HSIC (Li et al., 2021b). From our perspective, the similar format is mainly resulted from two aspects. On the one side, the outer optimization objectives of NCC-based loss and InfoNCE share the same softmax-like structure. On the other hand, we adopt similar label kernel as SSL-HSIC via adapting it to few-shot classification settings. In fact, there are two major differences between them.

Firstly, the most obvious difference between SSL-HSIC and MOKD is that MOKD takes the test power of kernel HSIC into consideration. As aforementioned, a challenging of applying HSIC is that the kernels used may sometimes fail to accurately measure the dependence between the given two data samples. In turn, models may fail to learn a set of class-specific representations where the similarities among samples belonging to the same class are maximized while the similarities between samples from different classes are minimized. Such phenomenon may further induce uncertainty and result in misclassification of samples. Thus, by introducing test power maximization in HSIC, kernels' capability of detecting dependence between data samples are improved. This facilitates to increase the sensitivity of kernel HSIC to dependence and further contributes to dependence optimization.

In addition, SSL-HSIC and MOKD are derived from different learning frameworks and are designed for different goals. To be specific, SSL-HSIC is derived from the InfoNCE loss (Oord et al., 2018) that is designed for unsupervised contrastive learning and focuses on learning robust and discriminative representations of a sample by contrasting two different views of the sample. The ultimate goal of SSL-HSIC is to learn a good feature encoder for downstream tasks. However, MOKD is derived from NCC-based loss (a.k.a., Prototypical loss) (Snell et al., 2017) that is designed for supervised few-shot classification and aims at learning a set of class-specific representations where the similarities among samples within the same class is maximized while the similarities between samples from different classes are minimized. The ultimate goal of MOKD is to learn the optimal task-specific parameters (of a linear transformation head) for each task to extract a set of class-specific representations where the data clusters are well learned and the undesirable high similarities are alleviated.

In order to compare SSL-HSIC with MOKD, we further conduct an experiment to reveal the differences between the two learning frameworks. In this experiment, HSIC measures used in SSL-HSIC are estimated in the unbiased way mentioned in this paper. The results are reported in Table 4. As we can observe, MOKD outperforms SSL-HSIC and SSL-HSIC with Test Power Maximization on all datasets of Meta-Datasets. Moreover, an interesting phenomenon is that SSL-HSIC achieves better

Table 4: **Comparisons of MOKD and SSL-HSIC.**

| Datasets | ImageNet | Omniglot | Aircraft | Birds | DTD | QuickDraw | Fungi | VGG_Flower | Traffic Sign | MSCOCO | MNIST | CIFAR10 | CIFAR100 |
|---|---|---|---|---|---|---|---|---|---|---|---|---|---|
| MOKD | **57.3±1.1** | **94.2±0.5** | **88.4±0.5** | **80.4±0.8** | **76.5±0.7** | **82.2±0.6** | **68.6±1.0** | **92.5±0.5** | **64.5±1.1** | **55.5±1.0** | **95.1±0.4** | **72.8±0.8** | **63.9±1.0** |
| SSL-HSIC | 56.5±1.2 | 92.0±0.9 | 87.3±0.7 | 78.1±1.1 | 75.2±0.8 | 81.4±0.7 | 63.5±1.2 | 90.9±0.8 | 59.7±1.3 | 51.4±1.1 | 93.4±0.6 | 70.0±1.1 | 61.8±1.1 |
| SSL-HSIC(TPM) | 56.9±1.1 | 92.6±0.9 | 87.5±0.6 | 79.8±0.9 | 75.7±0.7 | 82.0±0.7 | 67.1±1.0 | 91.4±0.6 | 62.4±1.0 | 53.6±1.0 | 94.3±0.5 | 71.5±0.8 | 63.5±1.0 |

performance when applying test power maximization to kernels used in SSL-HSIC. This strongly demonstrates that test power facilitates to capture the dependence between data samples and in turn learn better representations for each class in the given support set.

# E  MORE SETTINGS FOR CFC

In this section, we provide more details about cross-domain few-shot classification task settings.

## E.1  INTRODUCTION TO META-DATASET

Meta-Dataset was firstly proposed by Triantafillou et al. (2020) as a cross-domain few-shot classification benchmark. The selected datasets are free and easy to obtain, and span a variety of visual concepts and with different degrees in fine-grain. The original Meta-Dataset is composed with 10 different datasets which are ILSVRC_2012 (ImageNet) (Russakovsky et al., 2015), Omniglot (Lake et al., 2015), FGVC_Aircraft (Aircraft) (Maji et al., 2013), CUB_200-2011 (CU_Birds) (Wah et al., 2011), Describable Textures (DTD) (Cimpoi et al., 2014), Quick Draw (Jongejan et al., 2016), FGVCx Fungi (Fungi) (Schroeder & Cui, 2018), VGG_Flower (Flower) (Nilsback & Zisserman, 2008), Traffic Sign (Houben et al., 2013), MSCOCO (Lin et al., 2014). Then, MNIST (LeCun et al., 1998), CIFAR-10 (Krizhevsky et al., 2009) and CIFAR-100 (Krizhevsky et al., 2009) were added by Requeima et al. (2019).

## E.2  TASK SETTINGS

For vary-way vary-shot tasks, there are two main experimental settings: 'Train on All Datasets' and 'Train on ImageNet Only'. In 'Train on All Datasets' settings, the backbone we use is the multi-domain backbone which has observed training data of all 8 domains (ImageNet, Omniglot, Aircraft, CU_Birds, DTD, QuickDraw, Fungi and VGG_Flower). In 'Train on ImageNet Only' settings, the backbone we use is the single-domain backbone which is trained only on the training data of ImageNet dataset. For both set of settings, during meta-test phase, the evaluation is performed on the test data of seen domains and data from unseen domains.

In addition, for simplicity, the default setting in this paper is 'Train on All Datasets' settings if we do not provide any specific explanation.

## E.3  SPLIT SETTINGS

The rules for splitting the datasets in this paper are consistent with those in Triantafillou et al. (2020). For example, under 'Train on all datasets' settings, ImageNet, Omniglot, Aircraft, Birds, DTD, QuickDraw, Fungi and VGG_Flower are preserved as 'seen domains' where the training set of each dataset are accessible for training the backbone. Each dataset of seen domain is divided into training set, validation set and test set roughly with the proportions of 75%, 15% and 15%. Specifically, for ImageNet, Meta-Dataset constructs a sub-graph of the overall DAG that describes the relationships among all 82115 'synets' in ILSVRC_2012. Then, the entire graph is cut into three pieces for training, validation and test without overlap.

## E.4  VARY-WAY VARY-SHOT SETTINGS

Vary-way vary-shot task is a popular and basic task setting in cross-domain few-shot classification (Triantafillou et al., 2020; Dvornik et al., 2020; Liu et al., 2021a; Doersch et al., 2020; Li et al., 2021a; 2022). Such task settings stimulate the common daily situations where there exist distribution gaps among tasks and the data in the given task are imbalanced. Compared with conventional few-shot classification task settings where the numbers of ways and shots are fixed and tasks for test

are sampled from unseen data with the same distribution, vary-way vary-shot task is more challenging due to imbalanced data and distributional discrepancies between source and target domains.

In the context of cross-domain few-shot classification, a vary-way vary-shot task is sampled from a single dataset for each learning episode. Generally, the sample process of a vary-way vary-shot task mainly includes two independent steps: sampling a set of classes and sampling support and query data from the sampled classes. We only provide a brief introduction to the task sampling process, for more details, please refer to the paper of Meta-Dataset (Triantafillou et al., 2020).

**Class Sampling**   Given a dataset, the number of ways (classes) $N_C$ is sampled uniformly from the interval $[5, N_{max}]$, where $N_{max}$ denotes the maximum of the number of classes. Usually, $N_{max}$ is either 50 or as many classes as there are available.

**Support and Query Data Sampling**   After a set of classes is sampled, the numbers of shots for support and query sets are respectively determined by the following rules.

**Compute query set size.** In vary-way vary-shot task settings, the number of query data of each class in a task is fixed to the same number. The fixed number should be no more than the half of the total number of data in the given class so that there are still roughly 50% data being used as support data. The process is formulated as:

$$q = \min \left\{ 10, \left( \min_{c \in \mathcal{C}} \lfloor 0.5 * |c| \rfloor \right) \right\},$$

where $\mathcal{C}$ denotes a set of selected classes, $c$ denotes a single class and $|c|$ denotes the number of images in the given class $c$. In order to avoid too large query set, the maximum of the number of query data of each class is set to 10.

**Compute support set size.** The computation of support set size is formulated as:

$$s = \min \left\{ 500, \sum_{c \in \mathcal{C}} \lceil \beta \min \{100, |c| - q\} \rceil \right\},$$

where $\beta$ is a coefficient sampled uniformly from $(0, 1]$. In vary-way vary-shot task settings, the total number of data in a support set of a task is no more than 500. For each class in the selected set, the number of shots is determined by its remaining data where query data have been excluded. The maximum of the number of shots for each class is 100. The coefficient $\beta$ is used to sample smaller number of support data and generate a task with imbalanced number of shots.

**Data Sampling for Each Class.** After the support set size is determined, the number of shots for each class is calculated. First of all, $N_C$ random scalars $\{\alpha_1, \alpha_2, ..., \alpha_{N_C}\}$ are uniformly sampled from the interval $[\log(0.5), \log(2))$. Then, their 'contributions' to the support set are calculated as:

$$R_c = \frac{\exp(\alpha_c)|c|}{\sum_{c' \in \mathcal{C}} \exp(\alpha_{c'})|c'|}.$$

Then, the number of shots for class $c$ can be calculated by:

$$K_c = \min \left\{ \lfloor R_c * (s - |\mathcal{C}|) \rfloor + 1, |c| - q \right\}.$$

The term $R_c * (s - |\mathcal{C}|) \rfloor + 1$ is to guarantee that there is at least one sample being select for the class.

## F   MORE EXPERIMENTAL SETTINGS

### F.1   PRE-TRAINED BACKBONE

In this paper, we directly use both multi-domain and single-domain ResNet-18 (He et al., 2016) backbones provided by URL repository[1] for simplicity and fairness. Two kinds of backbones are respectively applied in our experiments according to different task settings. For 'Train on ImageNet Only' settings, the pre-trained backbone applied is a single domain-specific backbone which is trained

---

[1]https://github.com/VICO-UoE/URL

only on ImageNet dataset. For 'Train on All Datasets' settings, the pre-trained backbone applied is a multi-domain backbone. The multi-domain backbone is distilled from 8 single domain-specific pre-trained backbones. More details about model distillation are available in Li et al. (2021a).

For simplicity, except for specific explanation, the experiments are conducted on the multi-domain backbone under the 'Train on All Datasets' settings. In practice, we directly use both multi-domain and single-domain backbones provided in URL repository in order to make fair comparisons.

## F.2    MORE IMPLEMENTATION DETAILS

In this paper, we follow most settings in URL (Li et al., 2021a) to train a simple linear head on top of a pre-trained backbone. All experiments are conducted on an NVIDIA GeForce RTX 3090 GPU.

**Initialization & Learning rate.**    For each adaptation episode, we re-initialize the linear transformation layer as an identity matrix and learn a set of task-specific parameters for the given task. The optimizer used in MOKD is Adadelta (Zeiler, 2012). The learning rate is 1.0 for Traffic Sign and MNIST while 0.25 for the remaining datasets. Besides, the weight decay is set to 0.25 for seen domains while 0.0 for unseen domains.

**Values of $\gamma$.**    In vary-way vary-shot task settings, we intuitively set $\gamma$ to 1.0 for Omniglot, Aircraft, CU_Birds, Quick Draw and MNIST while 3.0 for other datasets. Since datasets like Omniglot and Aircraft are simple and the main object of each image is salient, small $\gamma$ is enough. In contrast, since datasets like ImageNet and Fungi are complex and each image contains too much semantic information, large $\gamma$ is required to penalize the high-variance representations and alleviate the overfitting phenomenon.

In addition, in vary-way 5-shot and 5-way 1-shot task settings, we respectively set $\gamma$ to 1.0 and 0 for all datasets since there are only few data samples in each task.

**Hardware & Seed settings.**    In this paper, all experiments are performed on an NVIDIA GeForce RTX 3090 GPU. The GPU memory required for running MOKD is about 5 GB. For fairness, all baselines of URL and experiments on MOKD are performed with seeds 41, 42, 43, 44, 45.

## F.3    ADAPTIVE BANDWIDTH SELECTION

Bandwidth is an essential component of a kernel (such as Gaussian kernel and IMQ kernel) since it closely corresponds to the test power of the kernel. It is widely believed that kernels with large test power are more sensitive to the dependence among data. Since the goal in this paper is to maximize dependence among samples belonging to the same class and minimize the dependence among all samples, a viable way is to select a suitable bandwidth to maximize the test power of the kernel. To this end, we firstly perform bandwidth selection before optimizing the objective loss to maximize its test power so that the optimized kernel is much more sensitive to the dependence. To be concrete, the test power is maximized by selecting an optimal bandwidth to maximize $\frac{\mathrm{HSIC}(\cdot,\cdot;\sigma,\theta)}{\sqrt{v+\epsilon}}$, where, $\sigma$ denotes the bandwidth, $v$ denotes the variance of HSIC, and $\epsilon$ is a constant that aims to avoid $v \leq 0$.

First of all, bandwidth is initialized as the median of the Gram matrix obtained with the data. Then, we manually set a list of coefficients to scale the median as the new bandwidth. To be concrete, the scale coefficient is selected from the list [0.001, 0.01, 0.1, 0.2, 0.25, 0.5, 0.75, 0.8, 0.9, 1.0, 1.25, 1.5, 2.0, 5.0, 10.0]. Finally, by iteratively calculating $\frac{\mathrm{HSIC}(\cdot,\cdot;\sigma,\theta)}{\sqrt{v+\epsilon}}$ where $\sigma$ is a scaled median, we select the $\sigma$ which obtains the largest test value of $\frac{\mathrm{HSIC}(\cdot,\cdot;\sigma,\theta)}{\sqrt{v+\epsilon}}$ as the optimal bandwidth.

The reason that we choose grid search method for the optimal bandwidth is the efficiency of MOKD. Optimizing bandwidth with auto optimizer requires extra hyperparameter selection and gradient descent steps, and these extra work will make the algorithm complicated and time-consuming.

## G   DETAILED EXPERIMENTAL RESULTS

### G.1   RESULTS UNDER VARY-WAY VARY-SHOT SETTINGS

In this section, we evaluate MOKD on vary-way vary-shot tasks under both 'Train on All Datasets' and 'Train on ImageNet Only' settings. To be clear, we mark seen domains with green while unseen domains with red.

### G.1.1   RESULTS UNDER TRAIN ON IMAGENET ONLY SETTINGS

The empirical results under 'Train on ImageNet Only' settings are reported in Table 1 with mean accuracy and 95% confidence. Here, we provide more detailed analysis regarding the results.

Generally, MOKD achieves the best performance among all approaches on 10 out of 13 datasets, including ImageNet, Omniglot, Textures (DTD), Quick Draw, Fungi, VGG_Flower, MSCOCO, MNIST, CIFAR10 and CIFAR100, and ranks is 1.3 in average of all datasets. Compared with URL, where MOKD is based, MOKD outperforms URL on almost all datasets. Specifically, compared with URL, MOKD obtains 1.5%, 0.2%, 0.7%, 0.9%, 3.3%, 0.8%, 1.6%, 0.4%, 2.3%, 2.1%, 1.2% and 1.1% improvements respectively from Omniglot to CIFAR-100. For ImageNet, which is the seen domain, MOKD gets the same results as URL and outperforms other previous works.

An interesting phenomenon is that MOKD performs better than URL on unseen domains compared with the results on seen domains. As we can see from table, MOKD achieves 1.5% improvements in average on unseen domains. Such phenomenon indicates that MOKD owns better generalization ability than previous works. Due to there exist distribution gaps between useen domains and unseen domains, it is challenging for a model to perform well on the domains that it has never observed before. We guess the reason for such phenomenon is that MOKD directly optimizes the dependencen respectively between representations and labels and representation themselves with the optimized kernel HSIC where the test power is maximized to be more sensitive to dependence. Thus, it is able to learn a set of better representations where similarities among samples within the same class are maximized while similarities between samples from different classes are minimized via capturing the accurate dependence between representations and labels.

### G.1.2   RESULTS UNDER TRAIN ON ALL DATASETS SETTINGS

The results under 'Train on All Datasets' settings are reported in Table 2 with mean accuracy and 95% confidence. Here, we are intended to provide more detailed analysis regarding empirical results.

According to the table, it is easy to observe that MOKD achieves the best performance in average and ranks 1.8 among all baselines. Compared with URL where our proposed MOKD is based, MOKD outperforms URL on 10 out of 13 datasets. Specifically, MOKD achieves 0.1%, 0.2%, 0.2%, 0.3%, 0.6%, 1.2%, 1.1%, 0.4%, 0.9% and 1.0% improvements respectively on Omniglot, Aircraft, CU_Birds, Textures (DTD), VGG_Flower, Traffic Sign, MSCOCO, MNIST, CIFAR10 and CIFAR100 datasets. Besides, compared with 2LM which is a recent new state-of-the-art method in cross-domain few-shot classification community, MOKD still achieves better performance on 8 out of 13 datasets.

Consistent with the results under 'Train on ImageNet Only' settings, MOKD also obtained better performance on unseen domains (Traffic Sign, MSCOCO, MNIST, CIFAR10 and CIFAR100) under 'Train on All Datasets' settings. Specifically, under 'Train on All Datasets' settings, MOKD achieves 1.2%, 1.3%, 0.4%, 0.9% and 1.0% improvements on Traffic Sign, MSCOCO, MNIST, CIFAR10 and CIFAR100 datasets. Such phenomenon consistently demonstrates that MOKD is able to obtain better generalization performance on previously unseen domains with only few learning adaptation steps.

Although MOKD achieves impressive performance on Meta-Dataset benchmark, we also notice that slight overfitting happens on Fungi dataset (see Fig. 5(g)).

### G.1.3   DISCUSSION ABOUT WHY MOKD GENERALIZES WELL ON UNSEEN DOMAINS

According to the results under both 'Train on all datasets' and 'Train on ImageNet only' settings, we observe that MOKD achieves better generalization performance on unseen domains than that on seen domains. From our perspective, the reasons for this phenomenon are collectively determined by

Table 5: **Comparisons of performance gaps between initial and final steps.**

| Datasets | ImageNet | Omniglot | Aircraft | Birds | DTD | QuickDraw | Fungi | VGG_Flower | Traffic Sign | MSCOCO | MNIST | CIFAR10 | CIFAR100 |
|---|---|---|---|---|---|---|---|---|---|---|---|---|---|
| URL | 2.00 | 0.15 | 1.36 | 0.27 | 2.42 | 0.19 | 1.40 | 0.58 | 13.94 | 2.30 | 3.59 | 2.77 | 3.52 |
| MOKD | 2.00 | 0.23 | 1.55 | 0.46 | 2.68 | 0.25 | 1.22 | 1.22 | 14.99 | 3.70 | 4.19 | 3.67 | 4.64 |

both pre-trained backbones and optimization objective of MOKD. Specifically, on the one side, the optimization objective of MOKD proposed in Eq. (5) is more powerful in exploring class-specific representations; on the other side, the pre-trained backbones limit feature exploration to some extent.

From the perspective of the optimization objective of MOKD, as we have mentioned in Theorem 3 of our paper, maximizing $\mathrm{HSIC}(\boldsymbol{Z}, Y)$ is equivalent to exploring a set of representations that matches the cluster structure of the given task. Meanwhile, since test power maximization is further taken into consideration, MOKD owns more powerful ability in exploring such representations compared with NCC-based loss. Our visualization results in Fig. 1 and 8 and 9 have demonstrated this.

However, the performance is not only simply determined by the optimization objective in Eq. (5), but also decided by the pre-trained backbones. In our paper, the backbone used under 'Train on all datasets' settings is pre-trained on 8 datasets, including ILSVRC_2012, Omniglot, Aircraft, CU_Birds, DTD, Quick Draw, Fungi and VGG Flowers. Since the distribution is shared between the training and test sets of a single dataset, it is easy for the pre-trained backbone to extract good features, where the cluster structures are definite, from test data of seen domains. However, such advantage may somewhat constrain the function space that can be explored by the linear transformation head. An intuitive explanation for this conjecture is that a loss will converge to the local optimal if the initial point is close to that local optimal. In contrast, since data from unseen domains have never been observed by the pre-trained model, the features extracted from the pre-trained backbone are not so good. Thus, it is more possible for MOKD to find better results.

As a simple demonstration, we compare the performance gaps between the initial and final adaptation steps. The results are obtained with random seed 42 from both URL and MOKD methods. Since both URL and MOKD initialize the linear head as identity matrix, the initial accuracies of them before performing adaptation are same (as shown in Fig. 5 in our paper).

Intuitively, a small performance gap means the extracted features from the pre-trained backbone are good enough for direct classification. Otherwise, the extracted features are not so good. According to the table, we notice that the gaps on seen domains are generally smaller than unseen domains for both URL and MOKD, which demonstrates that the extracted features from unseen domain data are not so good. Based on this obervation, since MOKD owns more powerful ability in exploring a set of representations that matches the cluster structure of the given task, better improvements are obtained on unseen domians.

## G.2 FURTHER STUDIES ON VARY-WAY 5-SHOT AND 5-WAY 1-SHOT

In this section, we further conduct experiments on more challenging 5-way 1-shot and vary-way 5-shot tasks under the settings of 'Train on All Datasets'. These tasks are difficult for our proposed MOKD since scarce data have negative effect on HSIC estimation. Besides, maximizing dependence between representations and labels with only few data samples may result in learning biased class-specific representations. The results are reported in Table 6.

**Vary-way 5-shot** According to Table 6, MOKD achieves best performance on 6 out of 13 datasets and ranks 1.8 among all baselines on vary-way 5-shot task settings. Generally, MOKD obtains comparable results on Omniglot and Quick Draw compared with the best, but obtains quite good results on Aircraft, CU_Birds, VGG_Flower and Traffic Sign with improvements $0.5\%$, $0.7\%$, $0.6\%$ and $2.5\%$ respectively. Such results show that MOKD is able to perform well even if the available data are relatively scarce. However, compared with the results under vary-way vary-shot settings, MOKD fails to achieve impressive performance totally.

**Five-way One-shot** The results regarding 5-way 1-shot tasks are reported in Table 6. According to the table, we observe that MOKD fails to achieve the best performance under 5-way 1-shot task settings. We notice that overfitting took place when performing MOKD on 5-way 1-shot tasks. The reason for such phenomenon is that there is only one sample available for training and MOKD tends

Table 6: **Results on vary-way 5-shot and 5-way 1-shot task settings (Trained on All Datasets).** Mean accuracy, 95% confidence interval reported.

| Datasets | Vary-way 5-shot | | | | | 5-way 1-shot | | | | |
|---|---|---|---|---|---|---|---|---|---|---|
| | Sim-CNAPS | SUR | URT | URL | **MOKD** | Sim-CNAPS | SUR | URT | URL | **MOKD** |
| ImageNet | 47.2±1.0 | 46.7±1.0 | **48.6±1.0** | 47.8±1.0 | 47.5±1.0 | 42.6±0.9 | 40.7±1.0 | **47.4±1.0** | 46.5±1.0 | 46.0±1.0 |
| Omniglot | 95.1±0.3 | 95.8±0.3 | **96.0±0.3** | 95.8±0.3 | **96.0±0.3** | 93.1±0.5 | 93.0±0.7 | **95.6±0.5** | 95.5±0.5 | 95.5±0.5 |
| Aircraft | 74.6±0.6 | 82.1±0.6 | 81.2±0.6 | 83.9±0.5 | **84.4±0.5** | 65.8±0.9 | 67.1±1.4 | 77.9±0.9 | **78.6±0.9** | **78.6±0.9** |
| Birds | 69.6±0.7 | 62.8±0.9 | 71.2±0.7 | 76.1±0.7 | **76.8±0.6** | 67.9±0.9 | 59.2±1.0 | 70.9±0.9 | **76.2±0.9** | 75.9±0.9 |
| Textures | 57.5±0.7 | 60.2±0.7 | 65.2±0.7 | **66.8±0.6** | 66.3±0.6 | 42.2±0.8 | 42.5±0.8 | 49.4±0.9 | **52.0±0.9** | 51.4±0.9 |
| Quick Draw | 70.9±0.6 | 79.0±0.5 | **79.2±0.5** | 78.3±0.5 | 78.9±0.5 | 70.5±0.9 | **79.8±0.9** | 79.6±0.9 | 79.1±0.9 | 78.9±0.9 |
| Fungi | 50.3±1.0 | 66.5±0.8 | 66.9±0.9 | 68.7±0.9 | **68.8±0.9** | 58.3±0.1 | 64.8±1.1 | 71.0±1.0 | **71.4±1.0** | 71.1±1.0 |
| VGG Flower | 86.5±0.4 | 76.9±0.6 | 82.4±0.5 | 88.5±0.4 | **89.1±0.4** | 79.9±0.7 | 65.0±1.0 | 72.7±1.0 | **80.3±0.8** | 79.8±0.8 |
| Traffic Sign | 55.2±0.8 | 44.9±0.9 | 45.1±0.9 | 56.7±0.8 | **59.2±0.8** | 55.3±0.9 | 44.6±0.9 | 52.7±0.9 | **57.4±0.9** | 57.0±0.9 |
| MSCOCO | 49.2±0.8 | 48.1±0.9 | **52.3±0.9** | 51.3±0.8 | 51.8±0.8 | 48.8±0.9 | 47.8±1.1 | **56.9±1.1** | 52.1±1.0 | 50.9±0.8 |
| MNIST | 88.9±0.4 | **90.1±0.4** | 86.5±0.5 | 88.5±0.4 | 89.4±0.3 | **80.1±0.9** | 77.1±0.9 | 75.6±0.9 | 73.3±0.8 | 72.5±0.9 |
| CIFAR-10 | **66.1±0.7** | 50.3±1.0 | 61.4±0.7 | 59.6±0.7 | 58.8±0.7 | **50.3±0.9** | 35.8±0.8 | 47.3±0.9 | 48.6±0.8 | 47.3±0.8 |
| CIFAR-100 | 53.8±0.9 | 46.4±0.9 | 52.5±0.9 | **55.8±0.9** | 55.3±0.9 | 53.8±0.9 | 42.9±1.0 | 54.9±1.1 | **61.5±1.0** | 60.2±1.0 |
| Average Seen | 69.0 | 71.2 | 73.8 | 75.7 | **76.0** | 65.0 | 64.0 | 70.6 | **72.5** | 72.2 |
| Average Unseen | 62.6 | 56.0 | 59.6 | 62.3 | **63.0** | 57.7 | 49.6 | 57.5 | **58.4** | 57.5 |
| Average All | 66.5 | 65.4 | 68.3 | 70.6 | **71.0** | 62.2 | 58.5 | 65.5 | **67.1** | 66.5 |
| Average Rank | 4.1 | 3.8 | 2.8 | 2.2 | **1.8** | 3.6 | 4.2 | 2.5 | **1.7** | 2.8 |

[1] Both the results on URL and MOKD are the average of 5 random seed. The ranks only consider the first 10 datasets.

Table 7: **Analyses on $\gamma$ (Trained on All Datasets).** Mean accuracy, 95% confidence interval are reported.

| Datasets | $\gamma = 0.0$ | $\gamma = 0.5$ | $\gamma = 1.0$ | $\gamma = 2.0$ | $\gamma = 3.0$ | $\gamma = 4.0$ | $\gamma = 5.0$ |
|---|---|---|---|---|---|---|---|
| ImageNet | 53.6±1.0 | 56.2±1.1 | 57.0±1.1 | 57.2±1.1 | **57.3±1.1** | **57.3±1.1** | **57.3±1.1** |
| Omniglot | **94.4±0.5** | **94.4±0.5** | 94.2±0.5 | 93.9±0.5 | 93.7±0.5 | 93.5±0.5 | 93.3±0.5 |
| Aircraft | 85.2±0.5 | 87.6±0.5 | **88.4±0.5** | 88.3±0.5 | 88.2±0.5 | 88.0±0.5 | 87.9±0.5 |
| Birds | 77.8±0.7 | 80.3±0.7 | **80.4±0.8** | 80.3±0.8 | 80.1±0.8 | 79.9±0.8 | 79.8±0.8 |
| Textures | 73.2±0.7 | 75.4±0.7 | 76.1±0.7 | 76.3±0.7 | **76.5±0.7** | **76.5±0.7** | **76.5±0.7** |
| Quick Draw | 80.5±0.6 | 82.1±0.6 | **82.3±0.6** | **82.3±0.6** | 82.2±0.6 | 82.1±0.6 | 82.0±0.6 |
| Fungi | 61.9±0.9 | 65.1±1.0 | 66.8±1.0 | 68.1±1.0 | 68.6±1.0 | **68.7±1.0** | **68.7±1.0** |
| VGG Flower | 88.8±0.5 | 91.5±0.5 | 92.1±0.5 | 92.4±0.5 | **92.5±0.5** | **92.5±0.5** | 92.3±0.5 |
| Traffic Sign | 48.7±1.0 | 62.9±1.1 | 64.1±1.1 | **64.6±1.1** | 64.5±1.1 | 64.2±1.1 | 64.0±1.1 |
| MSCOCO | 44.7±1.0 | 51.3±1.0 | 53.4±1.0 | 55.0±1.0 | **55.5±1.0** | 55.4±1.0 | 55.3±1.0 |
| MNIST | 91.7±0.5 | 95.0±0.4 | **95.1±0.4** | 94.6±0.4 | 94.5±0.4 | 94.3±0.4 | 94.3±0.4 |
| CIFAR-10 | 66.2±0.8 | 71.0±0.8 | 72.3±0.8 | 72.4±0.8 | 72.8±0.8 | **72.9±0.8** | **72.9±0.8** |
| CIFAR-100 | 57.1±1.0 | 62.5±1.0 | 63.5±1.0 | **64.0±1.0** | 63.9±1.0 | 63.8±1.0 | 63.6±1.0 |
| Average Seen | 76.9 | 79.1 | 79.7 | 79.8 | **79.9** | 79.8 | 79.7 |
| Average Unseen | 61.7 | 68.4 | 69.7 | 70.1 | **70.2** | 70.1 | 70.0 |
| Average All | 71.1 | 75.0 | 75.8 | 76.1 | **76.2** | 76.1 | 76.0 |

to learn excessively biased representations for each class. Even so, MOKD still outperforms other baselines except URL.

**Some remarks regarding the empirical results.** As shown in both vary-way 5-shot and 5-way 1-shot task settings, MOKD fails to significantly outperform all baselines. According to the learning curves in our experiments, we find that MOKD tends to overfit the data under these two settings. An reasonable for such phenomenon is that too few data samples have negative effect on HSIC estimation. For example, in of 5-way 1-shot task settings, $\text{HSIC}(Z, Y)$ has to explore representations from only 5 data samples to match the cluster structures of the given support set. Thus, it is highly possible that the learned representations will be extremely biased since there isn't any other samples for each class. Thus, the generalization performance drops. From this perspective, we know that the proposed MOKD is not quite suitable for tasks with scarce data samples in each class since the estimated HSIC is not reliable enough to learn a set of good representations that match the cluster structures.

### G.3 ANALYSES ON GAMMA

In our work, $\gamma$ functions as a coefficient of regularization term $\text{HSIC}(\boldsymbol{Z}, \boldsymbol{Z})$. Since $\text{HSIC}(\boldsymbol{Z}, \boldsymbol{Z})$ mainly facilitates to penalize high-variance representations and remove common information shared

Table 8: **Comparisons of running time between MOKD and URL**. (sec. per task)

| Datasets | ImageNet | Omniglot | Aircraft | Birds | DTD | QuickDraw | Fungi | VGG_Flower | Traffic Sign | MSCOCO | MNIST | CIFAR10 | CIFAR100 |
|---|---|---|---|---|---|---|---|---|---|---|---|---|---|
| URL | 0.7 | 0.8 | 0.5 | 0.7 | 0.4 | 1.1 | 1.0 | 0.5 | 1.0 | 0.9 | 0.5 | 0.5 | 1.1 |
| MOKD | 0.9 | 0.7 | 0.8 | 0.8 | 0.8 | 0.9 | 0.8 | 0.8 | 0.9 | 0.9 | 0.8 | 0.8 | 0.9 |

Table 9: **Ablation study on test power maximization.** Mean accuracy, 95% confidence interval are reported.

| Datasets | MOKD w/o Test Power | MOKD w Test Power |
|---|---|---|
| ImageNet | 56.4±1.1 | **57.2±1.1** |
| Omniglot | 93.9±0.5 | **94.2±0.5** |
| Aircraft | 88.1±0.5 | **88.3±0.5** |
| Birds | 80.1±0.8 | **80.2±0.8** |
| Textures | **75.9±0.7** | 75.7±0.7 |
| Quick Draw | 82.0±0.6 | **82.1±0.6** |
| Fungi | 64.1±1.1 | **68.9±1.0** |
| VGG Flower | **91.9±0.5** | 91.8±0.5 |
| Traffic Sign | 62.8±1.2 | **64.1±1.0** |
| MSCOCO | 52.9±1.1 | **55.3±1.0** |
| MNIST | **95.3±0.4** | 95.0±0.4 |
| CIFAR-10 | 72.1±0.8 | **73.0±0.8** |
| CIFAR-100 | 62.0±1.0 | **63.0±1.0** |
| Average Seen | 79.0 | **79.8** |
| Average Unseen | 69.0 | **70.1** |
| Average All | 75.2 | **76.1** |

among samples, $\gamma$ intuitively determines the power of penalization and suppression imposed to high-variance representations and common features shared across samples. In order to figure out how datasets in Meta-Dataset react to $\gamma$, we run MOKD with different $\gamma$ values under 'Train on all datasets' settings. The results are reported in Table 7 and Fig. 6 and 7.

According to the numerical results reported in Table 7, a general conclusion we can summarize is that different datasets prefer different values of $\gamma$. To be concrete, for simple datasets, such as Omniglot, Aircraft and MNIST, small $\gamma$ is preferred since images in these datasets are simple and the main object of each image is evident. However, for complicated datasets, such as ImageNet, Fungi and MSCOCO, large $\gamma$ is better since images in these datasets contain abundant objects and semantic information. In most cases, these semantic information is useless and sometimes may have negative effect on achieving good performance.

In addition, a special case is that it is equivalent to perform an ablation study on $\mathrm{HSIC}(\boldsymbol{Z}, \boldsymbol{Z})$ when $\gamma$ is set to zero. As shown in the table, the performance drops drastically on most datasets. By further plot the learning curves (see Fig. 6), we find that overfitting happens on these datasets when the $\mathrm{HSIC}(\boldsymbol{Z}, \boldsymbol{Z})$ term is removed. Thus, such phenomenon demonstrates that $\mathrm{HSIC}(\boldsymbol{Z}, \boldsymbol{Z})$ contributes to penalizing high-variance kernelized representations and alleviating the overfitting phenomenon.

**Further discussion about $\gamma$.** According to the results reported in Table 7, an interesting phenomenon is that different datasets achieve their best performance with different gamma values. From our perspective, the reasons for such phenomenon mainly include two aspects.

On the one side, as aforementioned, there exist high similarities among samples when performing classification with NCC-based loss. A reasonable conjecture for such phenomenon is the trivial common features shared across samples. On the other side, according to Fig. 6, merely maximizing $\mathrm{HSIC}(\boldsymbol{Z}, Y)$ results in overfitting phenomenon.

For complicated datasets, such as ImageNet and MSCOCO, there are many objects and abundant semantic information in images. However, most of semantic information is useless and sometimes has negative effect on representation learning. Thus, when tasks are sampled from these datasets, it is challenging for the model to learn definite and discriminative representations for each class. This will in turn result in uncertainties. For example, Fig. 8(g) and 9(d) have demonstrated this. Meanwhile,

due to the scarce data in few-shot classification tasks and strong power of $\text{HSIC}(\boldsymbol{Z}, Y)$, the model tends to learn high-variance representations and overfit the data.

Thus, large gamma value is essential for these complicated datasets. On the one side, according to Theorem 4, $\text{HSIC}(\boldsymbol{Z}, \boldsymbol{Z})$ facilitates to penalize high-variance kernelized representations for further alleviating overfitting phenomenon. Secondly, according to its definition, $\text{HSIC}(\boldsymbol{Z}, \boldsymbol{Z})$ measures the dependence between two sets of data. Thus, minimizing $\text{HSIC}(\boldsymbol{Z}, \boldsymbol{Z})$ drives the model to learn discriminative features for each single sample so that samples are "independent" to each other. This further helps remove the trivial common features shared across samples and in turn alleviate the high similarities among samples. In contrast, since simple datasets, such as Omniglot and Aircraft, owns evident and definite semantic area, small $\gamma$ is enough.

Thus, in our work, we set small $\gamma$ for those simple datasets, such as Aircraft, Omniglot and MNIST. However, for those complex datasets, such as ImageNet and MSCOCO, we set large $\gamma = 3$ (inspired by SSL-HSIC (Li et al., 2021b)).

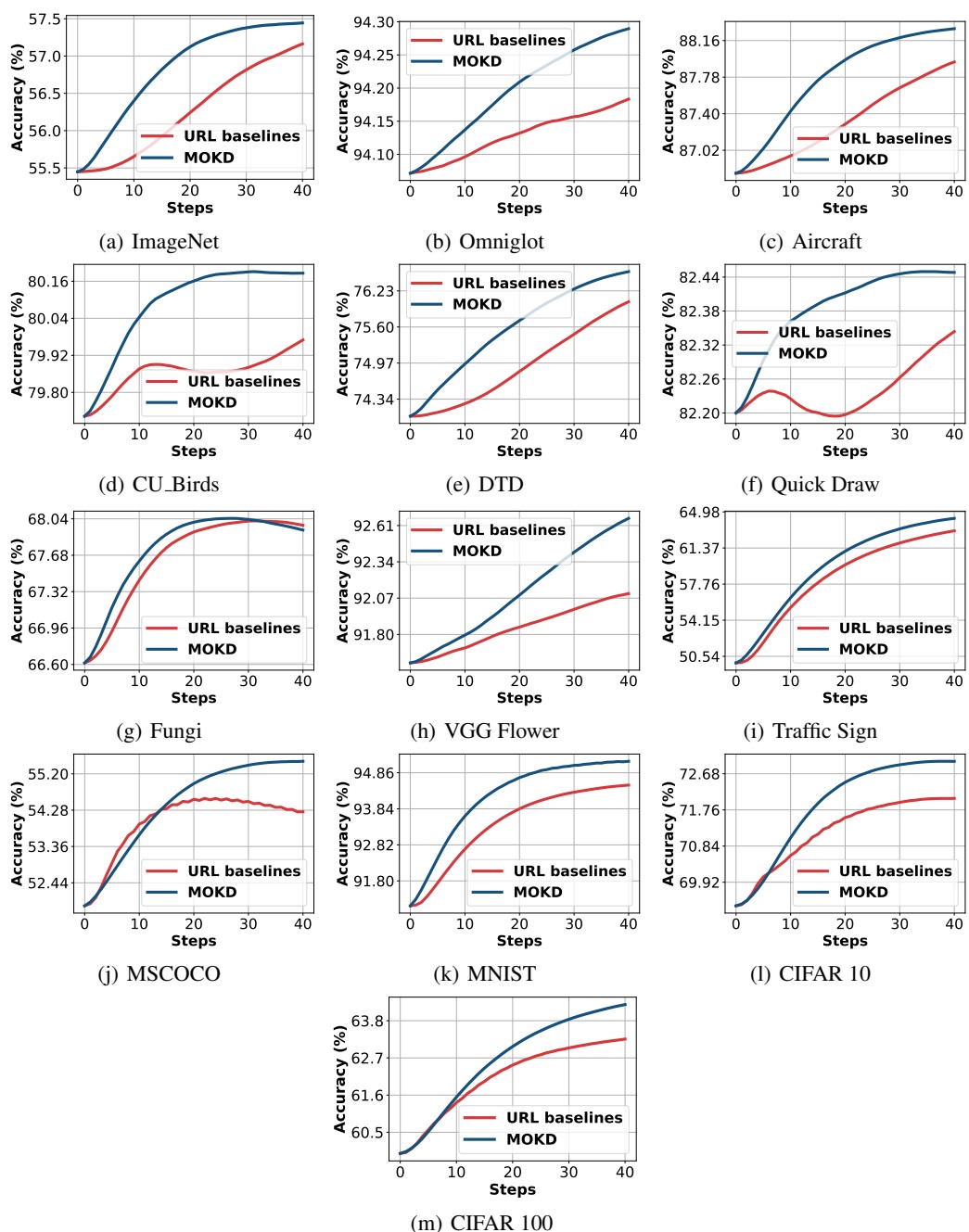

Figure 5: Test accuracy curves of Meta-Dataset with respect to the steps under 'Train on All Datasets' settings. As shown in figures, MOKD evidently achieves better learning process and convergence performance compared with URL baseline.

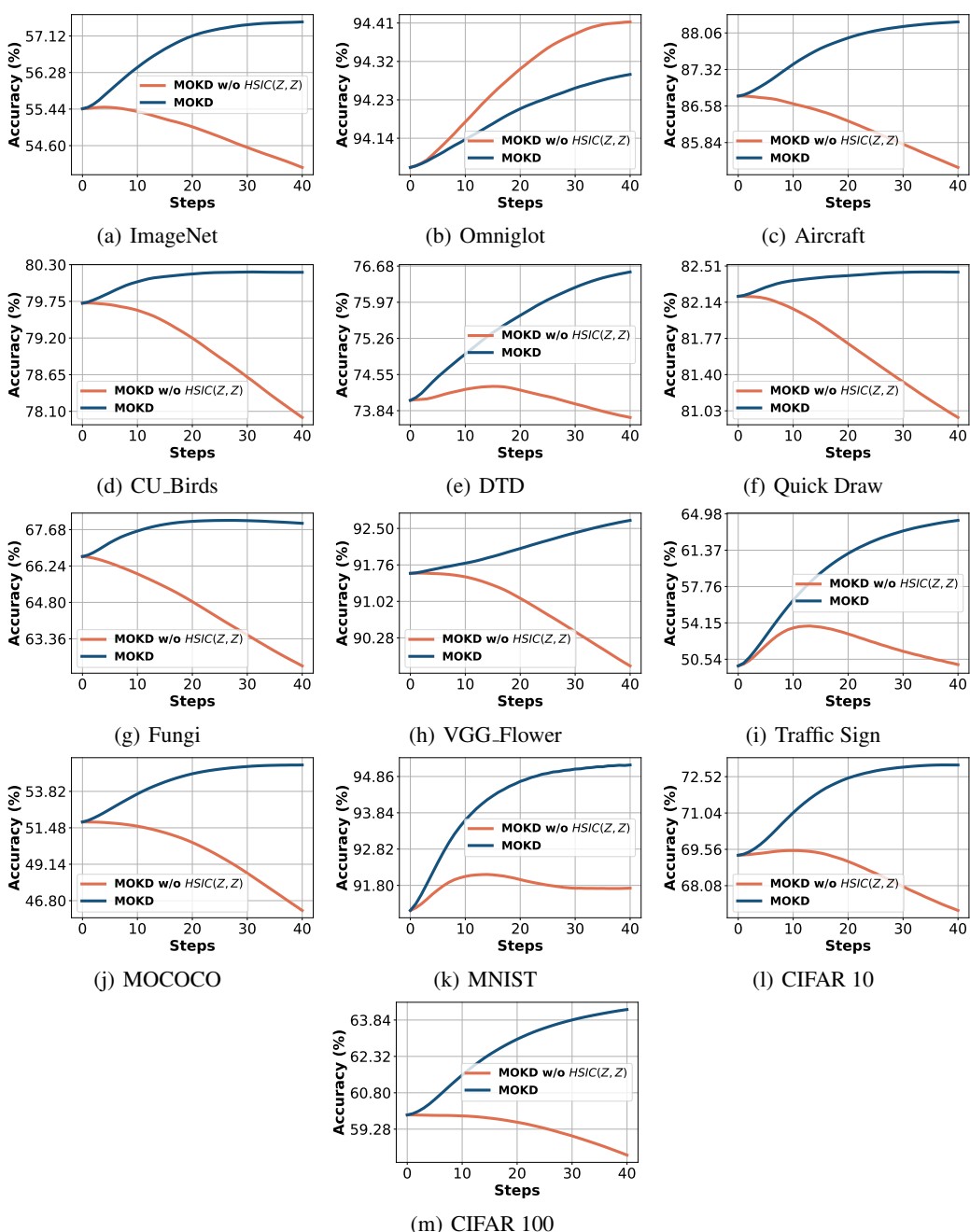

Figure 6: **Part test accuracy curves of datasets in Meta-Dataset.** The learning curves show that when $\mathrm{HSIC}(\boldsymbol{Z}, \boldsymbol{Z})$ is removed, MOKD tends to overfit the training data and achieves bad generalization performance.

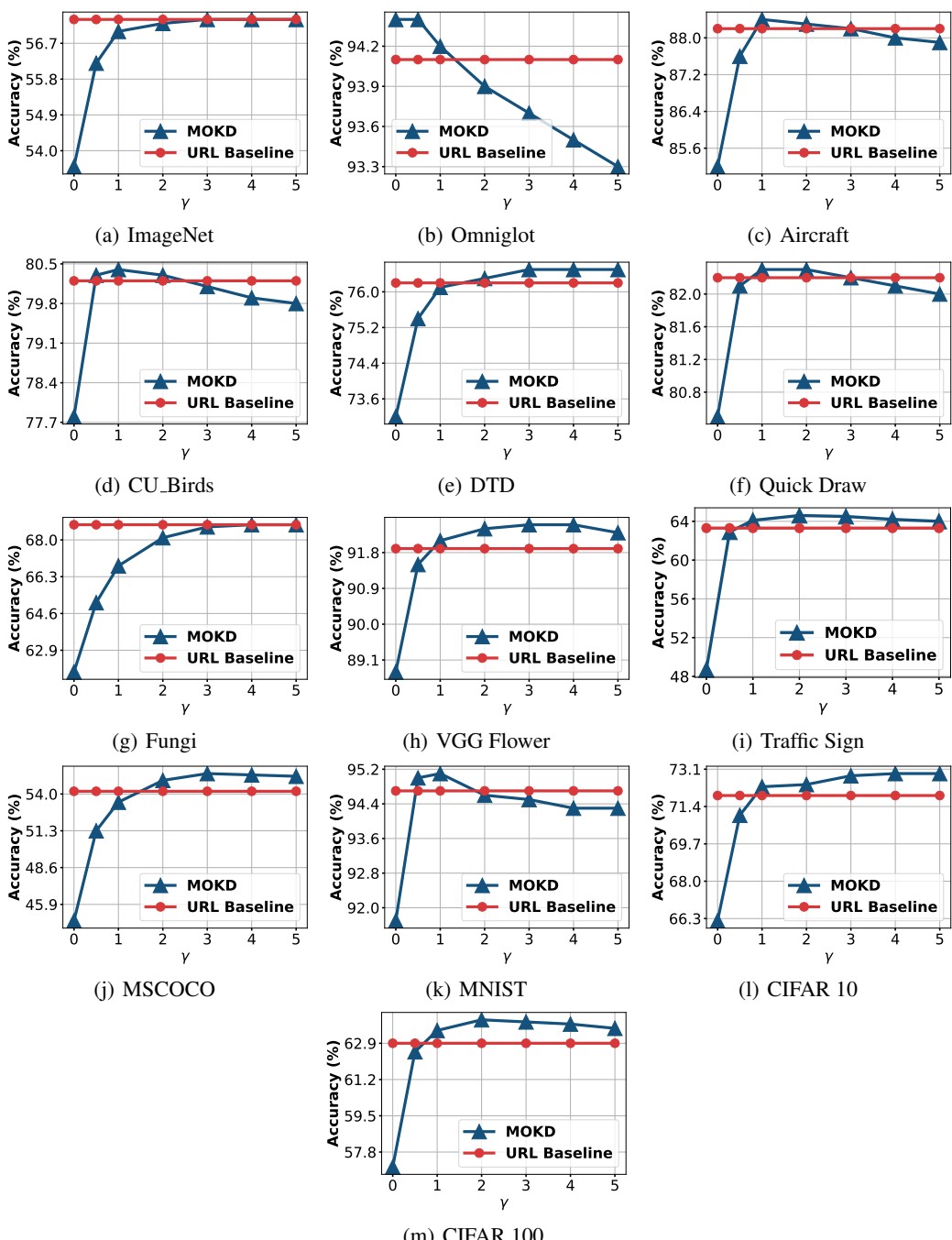

Figure 7: Illustration of effect of $\gamma$ on all datasets in Meta-Dataset.

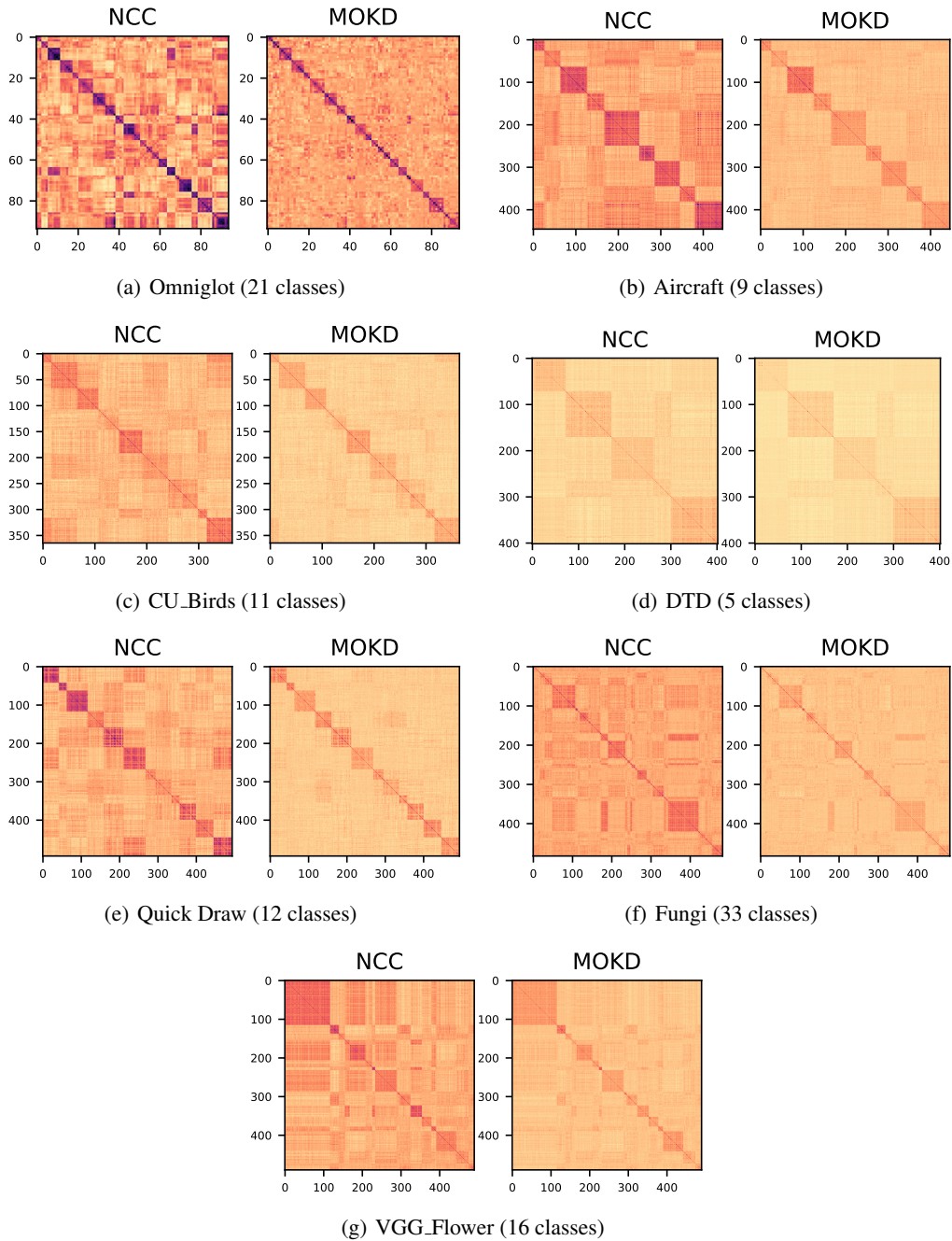

Figure 8: Visualization results of similarity matrices of representations repsectively learned with NCC-based loss and MOKD on seen domains.

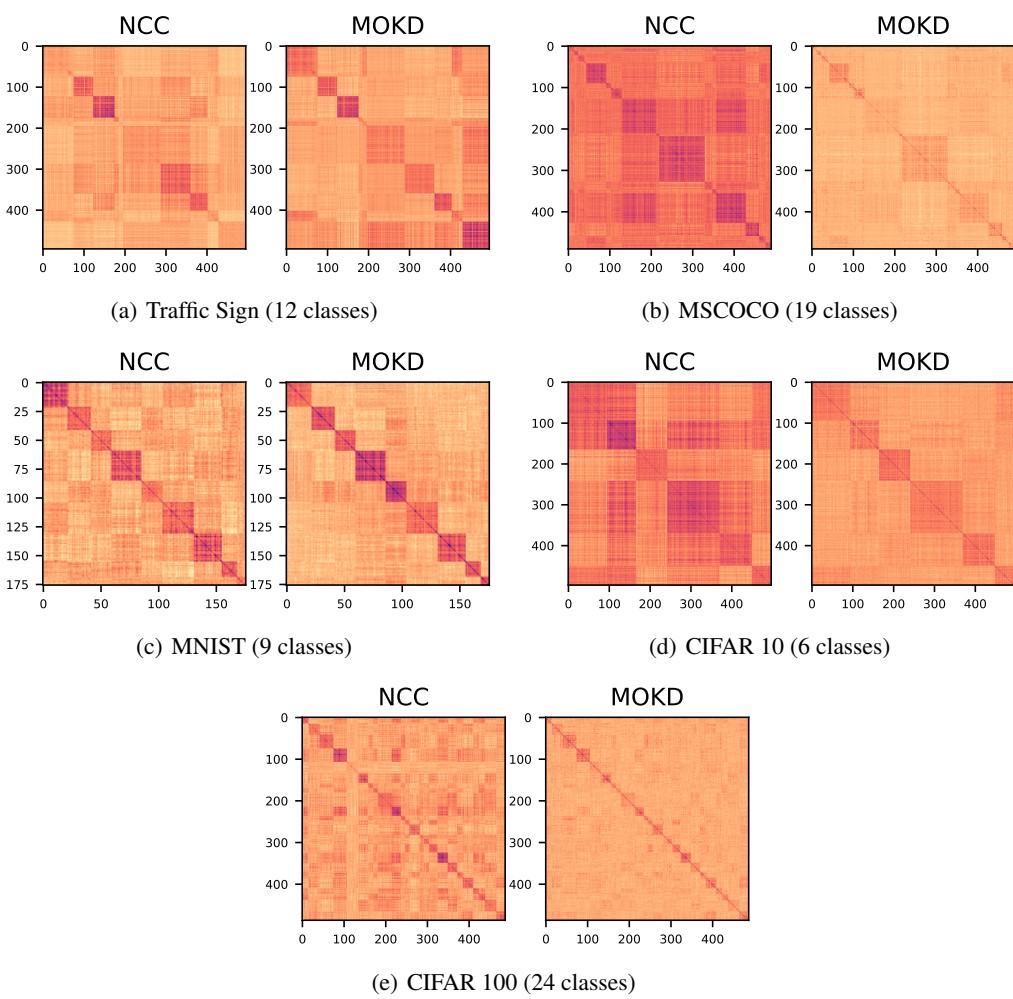

Figure 9: Visualization results of similarity matrices of representations repsectively learned with NCC-based loss and MOKD on unseen domains.

