# OpenReview forum: "Cross-domain Few-shot Classification via Maximization Optimized Kernel Dependence"
_ICLR.cc/2024/Conference — Submitted to ICLR 2024_

### Official Review · Reviewer_qXag · 2023-10-29

**Soundness:** 4 excellent
**Presentation:** 3 good
**Contribution:** 4 excellent
**Rating:** 8
**Confidence:** 4

**Summary:**

In the realm of cross-domain few-shot classification, the nearest centroid classifier (NCC) endeavors to develop representations that craft a metric space, facilitating few-shot classification through the assessment of similarities between samples and the prototype of each respective class. A foundational idea behind NCC is the gravitational pull of each sample towards its respective class centroid while simultaneously being repelled from other classes. Nonetheless, the authors observed alarming similarities between NCC-derived representations of samples hailing from disparate classes. Such inadvertent similarities could sow seeds of doubt, potentially culminating in the misclassification of samples. Addressing this quandary, they put forth a dual-layer optimization framework coined as maximizing optimized kernel dependence (MOKD). MOKD aims to refine the perceived similarities (dependence) amongst samples, maximizing the likeness among samples of a shared class while simultaneously suppressing similarities between samples of distinct classes. In its initial phase, MOKD optimizes the kernel Hilbert-Schmidt Independence Criterion (HSIC), amplifying its test power to attain an influential kernel dependence measure known as optimized kernel HSIC (opt-HSIC). Subsequent to this, an optimization challenge concerning the opt-HSIC is undertaken to accentuate the similarities among samples from identical classes while concurrently minimizing similarities across the entire sample pool. Given the pronounced test power of kernel HSIC, which displays heightened sensitivity to dependence, it can adeptly gauge the dependence within the representations of samples. Rigorous tests carried out on the esteemed Meta-Dataset benchmark confirm that MOKD not only exhibits superior generalization prowess across uncharted domains in a multitude of task settings but also excels in crafting superior data clusters.

**Strengths:**

1 Cross-domain few-shot classification (CFC) is an important problem setting, where researchers want to find a good way to let a pre-trained model adapt to down stream tasks quickly with a few samples. The problem itself is even more important after many foundation models are proposed recently.

2 This paper focuses on the most important solution pipeline of CFC, and first identifies its issues via checking the similarity between same-label classes and different-label classes. Then, an optimized measure is proposed to solve this issue. The proposed solution is well-motivated by the experiments and the theory, making this paper make a solid contribution to the field.

3 Connecting HSIC to NCC (in a theoretical view) is very interesting, which provides another insight to understand NCC and finally motivates the proposed solution.

4. This paper conducted extensive experiments to verify the proposed solution, which is empirically solid as well. The improvement on unseen domain is impressive, verifying that the proposed solution can obtain better representations for down stream tasks.

**Weaknesses:**

1 In Figure 1, we can observe that there are less noise in the MOKD case. However, it seems that the intra similarity is also smaller than (a). Can the authors explain this more?

2 What are implicit assumption this paper adopts to make few-shot learning be successful? Normally, we might have meta-distribution assumption or distribution-closeness assumption.

3 HSIC, as a valid statistic, can always be correct to evaluate the dependence among observed data. What is the exact function of maximizing test power in your method? What is the meaning of test power? It can be found that the opt-HSIC is indeed helpful. Thus, the reason is very interesting to be explained well.

4 How do you obtain the 95% confidence interval in all tables? Details might be needed here.

5 Figure 4 can be revised to avoid that the number 4.3 surpasses the figure.

**Questions:**

1 In Figure 1, we can observe that there are less noise in the MOKD case. However, it seems that the intra similarity is also smaller than (a). Can the authors explain this more?

2 What are implicit assumption this paper adopts to make few-shot learning be successful? Normally, we might have meta-distribution assumption or distribution-closeness assumption.

3 HSIC, as a valid statistic, can always be correct to evaluate the dependence among observed data. What is the exact function of maximizing test power in your method? What is the meaning of test power? It can be found that the opt-HSIC is indeed helpful. Thus, the reason is very interesting to be explained well.

4 How do you obtain the 95% confidence interval in all tables? Details might be needed here.

---

> ### Author Response · Authors · 2023-11-19
> **Rebuttal from authors to reviewer qXag (1/3)**
>
> Thank you for your efforts in reviewing our work. We appreciate for your valuable and insightful suggestions and questions. Now, we post our responses regarding your concerns in the following.
>
> > __Weakness 1.__ In Figure 1, we can observe that there are less noise in the MOKD case. However, it seems that the intra similarity is also smaller than (a). Can the authors explain this more?
>
> Per your question, we think that this phenomenon results from ${\rm HSIC}(\boldsymbol{Z}, \boldsymbol{Z})$.
>
> According to the definition of HSIC mentioned in our paper, ${\rm HSIC}(\boldsymbol{Z}, \boldsymbol{Z})$ measures the dependence among samples. Thus, minimizing ${\rm HSIC}(\boldsymbol{Z}, \boldsymbol{Z})$ is equivalent to driving the model to learn discriminative features for each sample so that samples tend to be mutually 'independent'.
>
> Thus, __when minimizing Eq.(5), the dependence, not only between data samples from different classes but also among samples within the same class, is minimized. Thus, the intra similarities becoming smaller__. However, we can still observe that the high similarities within the same class are dominant. Therefore, such phenomenon is acceptable.
>
> > __Weakness 2.__ What are implicit assumption this paper adopts to make few-shot learning be successful? Normally, we might have meta-distribution assumption or distribution-closeness assumption.
>
> Per your concern, from our perspective, there might be three implicit assumptions adopted in our paper.
>
> - __[High-level similarity.]__ First of all, we adopt the impilict assumption that downstream tasks (from both seen and unseen domains during meta-test time) are similar in high-level to those observed during pre-training phase, which is widely adopted in cross-domain few-shot classification works.
>
> - __[Independent class-specific distributions.]__ Then, the second implicit assumption adopted is that the distributions among different classes in a given task are mutually independent. In other words, we assume that the true class-specific representations are discriminative enough.
>
> - __[Assumption for analysis.]__ The final implicit assumption is proposed in Theorem 4. We assume that $k(\boldsymbol{Z}, \boldsymbol{Z}^{'})$ does not deviate much from $\mathbb{E}\_{{\boldsymbol{Z}^{'}\sim\mathbb{U}\_\mathcal{Z}}}[k(\boldsymbol{Z}, \boldsymbol{Z}^{'})]$ in order to perform Taylor expansion and use $\lim\_{x\to 0}\log(1+x)=x$.

---

> ### Author Response · Authors · 2023-11-19
> **Rebuttal from authors to reviewer qXag (2/3)**
>
> > __Weakness 3.__ HSIC, as a valid statistic, can always be correct to evaluate the dependence among observed data. What is the exact function of maximizing test power in your method? What is the meaning of test power? It can be found that the opt-HSIC is indeed helpful. Thus, the reason is very interesting to be explained well.
>
> Per your concern, we have provided a detailed introduction in our __related work section__.
>
> __[Why we need to maximize test power.]__ HSIC is correct to evaluate the dependence among two distributions. However, in practice, we do not have distributions and only have observed data. Thus, we hope to use less data to see the dependence among observed data, and maximizing the test power can help achieve that. HSIC can better capture the depedency for the observed data if its kernel is optimized by maximizing the test power.
>
> __[Exact function of TPM.]__ Based on the meaning of test power aforementioned, given an unbiased HSIC estimator (e.g. U-statistic estimator), under the hypothesis that the two distributions are dependent, the central limit theorem holds: $\sqrt{m}(\widehat{{\rm HSIC}}^2_{\rm u}-{\rm HSIC}^2)\stackrel{d}{\longrightarrow}\mathcal{N}(0, v^2)$, where $m$ denotes the number of samples, $\widehat{{\rm HSIC}}_{\rm u}$ denotes the HSIC value estimated with unbiased estimator, ${\rm HSIC}$ denote the real HSIC value, $v^2$ denotes the variance, $\stackrel{d}{\longrightarrow}$ denotes convergence in distribution.
>
> This CLT further implies the test power can be formulated in a format of __probability__: ${\rm Pr}\left(m\widehat{{\rm HSIC}}^2_{\rm u} >  r\right)\rightarrow\Phi\left(\frac{\sqrt{m}{\rm HSIC}^2}{v}-\frac{r}{\sqrt{m}v}\right)$, where $r$ denotes a rejection threshold and $\Phi$ denotes the standard normal CDF.
>
> Thus, __maximizing test power is equivalent to maximzing the probability ${\rm Pr}\left(m\widehat{{\rm HSIC}}^2_{\rm u} >  r\right)$__. Since the rejection threshold $r$ will converge to a constant, and ${\rm HSIC}$, $v$ are constants, for reasonably large $m$, the test power is dominated by the first term. __Thus, a feasible way to maximize the test power is to find a kernel function to maximize ${{\rm HSIC}^2}/{v}$__. In this way, in Eq.(5), we perform $\max\_{\sigma\_{\boldsymbol{Z}Y}}\frac{{\rm HSIC}(\boldsymbol{Z}, Y; \sigma\_{\boldsymbol{Z}Y}, \theta)}{\sqrt{v\_{\boldsymbol{Z}Y}+\epsilon}}, \max\_{\sigma\_{\boldsymbol{Z}\boldsymbol{Z}}}\frac{{\rm HSIC}(\boldsymbol{Z}, \boldsymbol{Z}; \sigma\_{\boldsymbol{Z}\boldsymbol{Z}}, \theta)}{\sqrt{v\_{\boldsymbol{Z}\boldsymbol{Z}}+\epsilon}}$ to respectively maximize the test power of the two HSIC measures.

---

> ### Author Response · Authors · 2023-11-19
> **Rebuttal from authors to reviewer qXag (3/3)**
>
> > __Weakness 4.__ How do you obtain the 95% confidence interval in all tables? Details might be needed here.
>
> Per your concern, 95% confidence interval is widely used in few-shot learning works. In our work, we follow previous works to calculate 95% confidence interval as done in MAML[1].
>
> During meta-test phase, we evaluate MOKD on each dataset with 600 randomly sampled tasks and record the accuracies. Then, for each dataset, with the accuracies obtained above, we can calculate the average accuracy $\mu$, and the standard deviation $\nu$. Then, the 95% confidence interval is calculated as: $\frac{1.96*\nu}{\sqrt{600}}$. Our experiemntal results are reported in a combination of mean accuracy and 95% confidence interval: $\mu \pm \frac{1.96*\nu}{\sqrt{600}}$.
>
>
> #### Reference
> [1] Finn et al. Model-agnostic meta-learning for fast adaptation of deep networks, ICML 2017.
>
> > __Weakness 5.__ Figure 4 can be revised to avoid that the number 4.3 surpasses the figure.
>
> Thank you for your suggestion, we will modify Fig.4 in our future version.

---

> > ### Comment · Reviewer_qXag · 2023-11-21
> >
> > Thanks for the detailed response and clarification of authors, which have addressed my concerns. Overall, I think this is a good contribution to this field, with interesting motivations, theoretical analyses, and empirical results.

---

> > > ### Author Response · Authors · 2023-11-21
> > > **Thank you for your feedback!**
> > >
> > > Dear Reviewer qXag,
> > >
> > > Thank you again for your time and efforts in reviewing our work. We really appreciate for your positive feedback towards our work.
> > >
> > > If you have any question in the future, we will be glad for discussion.
> > >
> > > Best wishes,
> > >
> > > Authors of Paper 2357

---

### Official Review · Reviewer_EDLJ · 2023-10-30

**Soundness:** 3 good
**Presentation:** 3 good
**Contribution:** 3 good
**Rating:** 6
**Confidence:** 3

**Summary:**

This paper proposes to replace the Nearest Centroid Classifier (NCC) loss with a loss based on Hilbert-Schmidt Independence Criterion (HSIC). The proposed loss called Maximizing Optimized Kernel Dependence (MOKD) is a bi-level optimization framework. MOKD first optimizes kernel parameters by test power maximization. Then, HSIC loss maximizes dependence between representations and labels and minimizes support among all samples based on optimized kernel dependence measure. Experiments on Meta-Dataset show the MOKD outperforms state-of-the-art methods on cross-domain few-shot classification methods and ablation studies.

**Strengths:**

-	The method of replacing NCC loss with HSIC loss is based on theoretical analysis (the upper bound analysis of NCC loss.)
-	The results on Meta-Dataset are better than state-of-the-art methods.
-	MOKD could produce prototypes that are better clustered compared to NCC-based loss.
-	Kernel parameter optimization by the Test Power Maximization (TPM) seems novel, and the ablation study confirms its effects.
-	MOKD does not significantly increase the running time compared with the URL baseline.

**Weaknesses:**

-	This paper proposes to learn the Nearest Centroid Classifier (NCC) by HSIC. This point seems not directly related to the Cross-domain problem.
-	The performance improvement by MOKD on the Trained on All Datasets setting is small.

**Questions:**

- Why would MOKD generalize unseen domains well?
- How is the classification performed on the test dataset? Is it the same as NCC?  Does the TPM is also conducted on the test dataset?

---

> ### Author Response · Authors · 2023-11-19
> **Rebuttal from authors to reviewer EDLJ (1/3)**
>
> Thank you for your efforts in reviewing our work. We appreciate for your valuable and insightful comments and questions. Now, we post our responses regarding your concerns in the following.
>
> > __Weakness 1.__ This paper proposes to learn the Nearest Centroid Classifier (NCC) by HSIC. This point seems not directly related to the Cross-domain problem.
>
> Per your concern, MOKD is closely related to cross-domain problem with two main aspects.
>
> - __[Literature perspective.]__ First of all, NCC-based is first proposed to solve few-shot classification problem[1]. Then, __Triantafillou et al.[2] introduces NCC-based loss into cross-domain few-shot classification tasks to avoid initializing a classifier layer with fixed number of classes, because the number of ways and shots are different across tasks (refer to Appendix E.4 for details)__. Thus, NCC-based loss is closely associated with cross-domain few-shot classifition tasks in existing literatures (refer to related works in Appendix A).
>
> - __[Distribution gap perspective.]__ In addition, one of challenges of cross-domain few-shot classificaion is __distribution gaps between source and target domains__. In our proposed MOKD method, since opt-HSIC, where test power is maximized, is more sensitive to dependence, it is able to explore a set of representations that matches the cluster structures of the given task. Thus, a set of better class-specific representations are learned from previously unseen domain data for further classification of query data. The empirical results in our paper also demonstrate that __MOKD achieves better generalization performance on unseen domains__.
>
> #### Reference
>
> [1] Snell et al., Prototypical networks for few-shot learning, NeurIPS 2017.
>
> [2] Triantafillou et al., Meta-dataset: a dataset of datasets for learning to learn from few-samples, ICLR 2020.

---

> ### Author Response · Authors · 2023-11-19
> **Rebuttal from authors to reviewer EDLJ (2/3)**
>
> > __Weakness 2 & Q1.__ The performance improvements on seen domains are small while evident on unseen domains?
>
> __[A summary view.]__ Per your question about the differences of performance improvements between seen and unseen domains, we think that __the reasons for such phenomenon are collectively determined by both pre-trained backbone and optimization objective of MOKD__. On the one side, __the optimization objective of MOKD owns more powerful ability in exploring class-specific representations__. On the other side, __The pre-trained backbone limits the feature exploration to some extent__.
>
> __[Power of ${\rm HSIC}(\boldsymbol{Z}, Y)$.]__ As we have mentioned in our paper, __maximizing ${\rm HSIC}(\boldsymbol{Z}, Y)$ is equivalent to exploring a set of representations that matches the cluster structure of the given task__. Meanwhile, since test power maximization is further taken into consideration, MOKD owns more powerful ability in exploring such representations compared with NCC-based loss.
>
> __[Limitation of pre-trained backbone.]__ However, __the performance is not only simply determined by the optimization objective in Eq.(5), but also decided by the pre-trained backbone__. In our paper, the backbone used under "Train on all datasets" settings is pre-trained on 8 datasets, including ILSVRC_2012, Omniglot, Aircraft, CU_Birds, DTD, Quick Draw, Fungi and VGG Flowers.
>
> Since the distribution is shared between the training and test sets of a single dataset, it is easy for the pre-trained backbone to extract good features, where the cluster structures are definite, from test data of seen domains. __However, such features may somewhat constrain the function space that can be explored by the linear transformation head__. An intuitive explanation for this conjecture is that a loss will converge to the local optimal if the initial point is close to that local optimal. In contrast, since data from unseen domains have never been observed by the pre-trained model, the features extracted from the pre-trained backbone are not so good. Thus, it is more possible for MOKD to find better results.
>
> __[Quantative results for limitation of pre-trained backbone.]__ As a simple demonstration, we compare the __performance gaps between the initial and final adaptation steps__. The results are obtained with random seed 42 from both URL and MOKD methods. __Since both URL and MOKD initialize the linear head as identity matrix, the initial accuracies of them before performing adaptation are same (as shown in Fig. 6 in our paper)__.
>
> Intuitively, __a small performance gap means the extracted features from the pre-trained backbone are good enough for direct classification. Otherwise, the extracted features are not so good__.
>
> According to the table, we notice that the gaps on seen domains are generally smaller than unseen domains for both URL and MOKD, which demonstrates that the extracted features from unseen domain data are not so good.
>
> Based on this obervation, __since MOKD owns more powerful ability in exploring a set of representations that matches the cluster structure of the given task, better improvements are obtained on unseen domians__.
>
> | **Datasets** | **ImgN** | **Omni** | **AirC** | **Brds** | **DTD** | **QikD** | **fung** | **VGGF** | **TS** | **MSCO** | **MST** | **CF10** | **CF100** |
> | -------- | -------- | -------- | -------- | -------- | -------- | -------- | -------- | -------- | -------- | -------- | -------- | -------- | -------- |
> | URL | $2.00$ | $0.15$ | $1.36$ | $0.27$ | $2.42$ | $0.19$ | **$1.40$** | $0.58$ | $13.94$ | $2.30$ | $3.59$ | $2.77$ | $3.52$ |
> | MOKD | $2.00$ | $0.23$ | **$1.55$** | **$0.46$** | **$2.68$** | $0.25$ | $1.22$ | **$1.22$** | **$14.99$** | **$3.70$** | **$4.19$** | **$3.67$** | **$4.64$** |

---

> ### Author Response · Authors · 2023-11-19
> **Rebuttal from authors to reviewer EDLJ (3/3)**
>
> > __Q2.__ How is the classification performed on the test dataset? Is it the same as NCC? Does the TPM is also conducted on the test dataset?
>
> Per your question, during meta-test time, given a task which is sampled from test dataset and includes support data (for adaptation) and query data (for evaluation), __MOKD firstly maximizes the test power of kernel dependence measures via respectivly maximizing $\frac{{\rm HSIC}(\boldsymbol{Z}, Y; \sigma\_{\boldsymbol{Z}Y}, \theta)}{\sqrt{v\_{\boldsymbol{Z}Y}+\epsilon}}$ and $\frac{{\rm HSIC}(\boldsymbol{Z}, \boldsymbol{Z}; \sigma\_{\boldsymbol{Z}\boldsymbol{Z}}, \theta)}{\sqrt{v\_{\boldsymbol{Z}\boldsymbol{Z}}+\epsilon}}$ with the support data__, and __then iteratively update the model on support data via performing gradient descent on the outer optimization objective in Eq.(5)__. The classification is performed by measuring similarities between query data and prototypes obtained via averaging support data within the same class, which is same as NCC. However, __the classification results are not involved in the optimization of the model__.

---

> ### Author Response · Authors · 2023-11-22
> **Any further concern or question**
>
> Dear Reviewer EDLJ,
>
> Thank you again for your efforts in reviewing our work and valuable comments. We have provided detailed responses to your concerns and questions regarding our work. We hope that our responses help address your concerns.
>
> Since the deadline of discussion (22 Nov) is approaching, if you still have any concern or question regarding our work, please let us know. We are glad to help address any concern or question.
>
> Best regards,
>
> Authors of Paper 2357

---

> > ### Comment · Reviewer_EDLJ · 2023-11-22
> >
> > Thank you for the detailed explanations. These are sufficient for me.

---

> > > ### Author Response · Authors · 2023-11-22
> > > **Thank you for your feedback**
> > >
> > > Dear Reviewer EDLJ,
> > >
> > > Thank you for your feedback. We really appreciate for your time and efforts in reviewing our work. Your valuable comments help us further improve the quality of our work. We will update the discussion in our future revised version.
> > >
> > > Best wishes,
> > >
> > > Authors of Paper 2357

---

### Official Review · Reviewer_hhDT · 2023-10-30

**Soundness:** 3 good
**Presentation:** 3 good
**Contribution:** 3 good
**Rating:** 6
**Confidence:** 4

**Summary:**

This paper proposes a cross-domain few-shot classification method that is called maximizing optimized kernel dependence (MOKD). The proposed method consists in an optimization problem based on Hilbert-Schmidt Intendance Criterion (HSIC), in which the similarities among samples of the same class are minimized while the similarities among all samples are maximized. Experiments on standard benchmark of Meta-Dataset show that MOKD achieves better performance than the competing methods.

**Strengths:**

(1) Hilbert-Schmidt Intendance Criterion (HSIC) is a powerful kernel method for measuring dependence between two random variables. This measure is rarely used in few-shot learning (FSL) and this paper makes an interesting exploration by introducing it into FSL.

(2) The paper derives the objective of optimized kernel HSIC (opt-HSIC) based on Theorem 1, 3 and 4, clarifying its connection with the nearest centroid classifier (NCC).

**Weaknesses:**

The proposed MOKD method is very sensitive to parameter $\gamma$ (see figure 8 in appendix). This point is not raised especially in main paper: in section 5.2, only benefits of incorporating HSIC term is mentioned but not the sensibility of its contribution. It would interesting to elaborate more in the main paper on this point.

**Questions:**

1. Could you elaborate more on the senstivity to the $\gamma$ parameter? Especially how is is set for the various datasets.

---

> ### Author Response · Authors · 2023-11-19
> **Rebuttal from authors to reviewer hhDT**
>
> Thank you for your efforts in reviewing our work. We appreciate for your valuable and insightful questions. Now, we post our responses regarding your concerns in the following. __Q1.__ Could you elaborate more on the senstivity to the parameter? Especially how is is set for the various datasets.
>
> >__Weakness 1.__ The proposed MOKD method is very sensitive to parameter (see figure 8 in appendix). This point is not raised especially in main paper: in section 5.2, only benefits of incorporating HSIC term is mentioned but not the sensibility of its contribution. It would interesting to elaborate more in the main paper on this point.
> > __Q1.__ Could you elaborate more on the senstivity to the parameter? Especially how is is set for the various datasets.
>
> __[Response for suggestion.]__ Thank you for your suggestion. Due to the space limits, we had to put these contents in appendix. We will add the discussion in our paper and consider to put them in main paper in our future version.
>
> __[Response for Q1.]__ Per your concern about the sensitivity of $\gamma$, in order to better explain this question, we would like to first state two observations from our empirical results:
> - __MOKD is not sensitive to parameter $\gamma$ on a given dataset__. For example, the accuracy on ImageNet does not change significantly when $\gamma$ is set larger than 2. Such phenomenon also exists on other datasets. __In fact, the performance is stable on a certain range of gamma values for each dataset__.
> - __From the perspective of total performance on meta-dataset, MOKD is also not sensitive to parameter $\gamma$__. For example, according to Table 6, if we set $\gamma=3.0$ for all datasets, the average performance on seen, unseen and all datasets does not change significantly.
>
> Thus, we think that the 'sensitivity' you mentioned is that __different datasets achieve their best performance with different gamma values__. From our perspective, there are two main aspects determining the values of $\gamma$ for each dataset in meta-dataset benchmark.
> - __The high similarities among samples__. As we mentioned in our paper, there exists high similarities when performing CFC with NCC-based loss. __A reasonable conjecture about the cause of such phenomenon is trivial common features shared across samples__.
> - __Potential overfitting resulted from ${\rm HSIC}(\boldsymbol{Z}, Y)$__. According to our empirical results in Fig.7, only maximizing ${\rm HSIC}(\boldsymbol{Z}, Y)$ results in overfitting.
>
> For complicated datasets, such as ImageNet and MSCOCO, there are many objects and abundant semantic information in images. However, most of semantic information is useless and sometimes has negative effect on representation learning. Thus, when tasks are sampled from these datasets, __it is challenging for the model to learn definite and discriminative representations for each class. This will in turn result in uncertainties__. For example, Fig.5(a), Fig.9(g) and Fig.9(i) have demonstrated this. Meanwhile, __due to the scarce data in few-shot classification tasks and strong power of ${\rm HSIC}(\boldsymbol{Z}, Y)$, the model tends to learn high-variance representations and overfit the data__.
>
> Thus, large gamma value is essential for these complicated datasets. On the one side, as shown in Theorem 4, __${\rm HSIC}(\boldsymbol{Z}, \boldsymbol{Z})$ helps penalize high-variance representations, which contributes to alleviating overfitting__. On the other side, according to the definition of HSIC, __minimizing ${\rm HSIC}(\boldsymbol{Z}, \boldsymbol{Z})$ drives model to learn discriminative features for each single sample so that samples are 'independent' with each other__. This further helps remove the trivial common features shared across samples and in turn alleviate the high similarities among samples.
>
>
> In contrast, since simple datasets, such as Omniglot and Aircraft, owns evident and definite semantic area, it is less possible for the model to learn high-variance features. Thus, small $\gamma$ is enough.
>
> Thus, in our work, we set small $\gamma$ for those simple datasets, such as Aircraft, Omniglot and MNIST. However, for those complex datasets, such as ImageNet and MSCOCO, we set large $\gamma=3$ (inspired by SSL-HSIC).

---

> ### Author Response · Authors · 2023-11-22
> **Any further concern or question**
>
> Dear Reviewer hhDT,
>
> Thank you again for your efforts in reviewing our work and valuable comments. We have provided detailed responses to your concerns and questions regarding our work. We hope that our responses help address your concerns.
>
> Since the deadline of discussion (22 Nov) is approaching,  if you still have any concern or question regarding our work, please let us know. We are glad to help address any concern or question.
>
> Best regards,
>
> Authors of Paper 2357

---

> > ### Comment · Reviewer_hhDT · 2023-11-22
> >
> > Thanks for this clarification. This addressed my concerns.

---

> > > ### Author Response · Authors · 2023-11-22
> > > **Thank you for your feedback**
> > >
> > > Dear Reviewer hhDT,
> > >
> > > Thank you for your feedback. We will update the discussion in our revised paper.
> > >
> > > Thank you again for your time and efforts in reviewing our work!
> > >
> > > Best regards,
> > >
> > > Authors of Paper 2357

---

### Official Review · Reviewer_rkmN · 2023-10-31

**Soundness:** 3 good
**Presentation:** 2 fair
**Contribution:** 2 fair
**Rating:** 5
**Confidence:** 4

**Summary:**

In this paper, the authors propose a method called MOKD (Maximizing Optimized Kernel Dependence) which learns a set of class-specific representations by maximizing the dependence between feature representations and minimizing the dependence among all samples based on the optimized kernel dependence measures. To this end, they utilize Hilbert-Schmidt Independence Criterion (HSIC) which attempt to maximize optimized kernel dependence which is recently adopted in SSL-HSIC (Li et al., 2021b). The method can be seen as an extension of this method. I would not be surprised if the authors are same. The only difference is the utilization of test power in the revised method. The authors report better accuracies compared to SSL-HSIC.

**Strengths:**

Basic strengths of the paper can be summarized as follows:
i) The method is basically built on strong theoretical foundations whose efficacy is proved in the literature. The authors utilize the Hilbert-Schmidt Independence Criterion (HSIC) and kernel dependence maximization and efficacy of these concepts are already well demonstrated in the similar classification problems.
ii) Demonstrating the ties between NCC and the proposed method is a plus.
iii) Introducing test power concept seems novel.
iv) The authors report better accuracies compared to the related methods including SSl-HSIC.

**Weaknesses:**

Main weaknesses of the paper can be summarized as follows:
i) There is a very similar method (SSL-HSIC) using the same concepts. As far as I understand the only difference is utilization of test power concept. Therefore, this method is revised version of another method, which limits the novelty.
ii) There are some wrong statements in the paper. The authors define k(z,c_c) as the kernel function given at the likelihood function used in Equation (1). I checked the paper again. The correct term must be exp(-d(z,c_c)) and d(z,c_c) is actually distance between z and c_c. Furthermore, the authors (Li et al., 2021a) use Mahalanobis distance not the cosine distance. Please note that the whole term, exp((-d(z,c_c)) looks like a Gaussian kernel function without a width term. They must be corrected. If the authors used this terminology in their proofs, they must be corrected as well.
iii) All the arguments given under Theorem 1 are not new, they are given in the main NCC paper. Similarly, the arguments given below Theorem also exist in (Li et al., 2021b).
iv) The optimization problem given at (5) is a constrained optimization problem and I wonder how the authors enforce these two constraints. For the first one, they convert it to selection of a width term if I understand correctly. What about the second constraint? Please give more information on optimization of the problem.
v) The authors compare their proposed method to URL, but I wonder why they do not compare against SSL-HSIC and NCC?

**Questions:**

1) The authors compare their proposed method to URL, but I wonder why they do not compare against SSL-HSIC and NCC?
2) Please explain how the optimization problem is solved in more details (e.g., how the contraints are enforced?)

---

> ### Author Response · Authors · 2023-11-19
> **Rebuttal from authors to reviewer rkmN (Weakness 1 (1/2))**
>
> Thank you for your efforts in reviewing our work. We appreciate for your valuable comments. Now, we post our responses regarding your concerns in the following.
>
> > __Weakness 1.__ There is a very similar method (SSL-HSIC) using the same concepts. As far as I understand the only difference is utilization of test power concept. Therefore, this method is revised version of another method, which limits the novelty.
>
> __[A summary view.]__ Per your concern regarding the same concepts, Hilbert-Schmidt Independence Criterion is a widely used independence measure in machine learning areas. There are several works adopting HSIC as loss function [1] or gradients [2]. However, in this paper, the main reasons that MOKD is similar to SSL-HSIC are that
> - __(i) the outer objective of NCC-based loss $-log\frac{e^{k(\boldsymbol{z}, \boldsymbol{c})}}{\sum\_{c^{'}} e^{k(\boldsymbol{z}, \boldsymbol{c}^{'})}}$ and InfoNCE loss $-log\frac{e^{k(\boldsymbol{z}, \boldsymbol{z}_+)}}{\sum\_{z^{'}} e^{k(\boldsymbol{z}, \boldsymbol{z}^{'})}}$ share the similar softmax-like structure__;
> - __(ii) we adopt similar label kernel as SSL-HSIC via adapting it to few-shot classification settings__.
>
> Thus, MOKD is not a simply revised version of SSL-HSIC. __In fact, MOKD is derived from NCC-based loss under the settings of few-shot classification tasks__.
>
> __[New interpretation of NCC-based loss.]__ As far as we know, existing mainstream works regarding cross-domain few-shot classification tasks (vary-way vary-shot settings) mainly focus on improving performance from the angle of model modules based on NCC-based loss.
>
> However, in our work, __MOKD proposes a new perspective to interpret NCC-based loss from the angle of dependence measures and seek to learn better class-specific representations via performing optimization on these dependence measures__. The interpretation mainly reveals two important aspects:
> - __The original NCC-based loss can be expressed in a format of dependence measures (Hilbert-Schmidt Independence Criterion)__;
> - __The dependence measure plays an essential role in learning a set of representations, where similarities among samples within the same class are maximized while similarities between samples from different classes are minimized, from few labeled data__.
>
> __[How the new interpretation motivate us.]__ According to Fig. 1, the qualitive results show that there exist high similarities among samples. __Such phenomenon implies that conventional NCC-based loss fails to learn class-specific representations from the given few labeled data__.
>
> A feasible explanantion for such phenomenon is that __conventional NCC-based loss fails to accurately measure the dependence between data samples and in turn fails to explore a set of representations that matches the implicit cluster structure of the given task__.
>
> In order to solve this problem, a direct way is enhancing independence measures' capbility in dependence detection. __This motivates us to consider maximizing the test power of kernel dependence, which is able to increase the sensitivity of kernel dependence to dependence__.
>
> According to the empirical results reported in experiment section, the utilization of HSIC measures where test power is maxmized indeed facilitates to improve the generalization performance on meta-dataset and learn more clear and definite class clusters for each class in a given task.

---

> > ### Author Response · Authors · 2023-11-19
> > **Rebuttal from authors to reviewer rkmN (Weakness 1 (2/2))**
> >
> > __[Differences between SSL-HSIC and MOKD.]__ In fact, __we have discussed the differences between SSL-HSIC and MOKD in Section 4.1 and Appendix D.__ From our perspective, the differences between MOKD and SSL-HSIC mainly include two aspects.
> > - __[Introduction of TPM.]__ First of all, __the most obvious difference is that we utilize test power maximization (TPM) to enhance kernel HSIC measures' ability of detecting dependence between two batches of data.__ This further contributes to dependence optimization and learning better class-specific representations where similarities among samples within the same class are maximized while similarities between samples from different classes are minimized.
> >
> > - __[Derivations and ultimate goals.]__ In addition, __the derivations and ultimate goals of SSL-HSIC and MOKD are also different.__ SSL-HSIC is derived from unsupervised contrastive learning paradigm. Since each sample is treated as an individual class in typical contrastive learning, __optimizing SSL-HSIC is equivalent to learning robust and discriminative features in instance-level for downstream tasks via comparing two different views of a data sample.__ However, MOKD is derived from supervised few-shot classification framework which __aims at learning a set of class-specific features, where similarities among samples within the same class are maximized while similarities between samples from different classes are minimized, to further perform classification on query data in a given task.__ Thus, __optimizing MOKD framework is equivalent to exploring class-specific features based on kernels that are sensitive to dependence.__
> >
> > __[Experiments: Comparisons between SSL-HSIC and MOKD.]__ In order to compare MOKD and SSL-HSIC in a more __quantative__ way, we also provide __empirical results in Table 4 in Appendix D__ to compare the performance between SSL-HSIC and MOKD on cross-domain few-shot classification tasks. __Generally, MOKD performs better. An interesting phenomenon is that the performance of SSL-HSIC increases when TPM is applied. This strongly demonstrates the power of test power maximization__.

---

> > ### Comment · Reviewer_rkmN · 2023-11-20
> > **reviewer response for the authors' response**
> >
> > These explanations are irrelevant. I simply indicated that the method proposed here is an extension of a previously published paper, which limits the neovelty. That is it. You did not have to go through Hilbert-Schmidt Independence Criterion and all other stuff.

---

> ### Author Response · Authors · 2023-11-19
> **Rebuttal from authors to reviewer rkmN (Weakness 2 (1/2))**
>
> >__Weakness 2.__ There are some wrong statements in the paper. The authors define k(z,c_c) as the kernel function given at the likelihood function used in Equation (1). I checked the paper again. The correct term must be exp(-d(z,c_c)) and d(z,c_c) is actually distance between z and c_c. Furthermore, the authors (Li et al., 2021a) use Mahalanobis distance not the cosine distance. Please note that the whole term, exp((-d(z,c_c)) looks like a Gaussian kernel function without a width term. They must be corrected. If the authors used this terminology in their proofs, they must be corrected as well.
>
> Per your concern, we guess that there might be some misunderstandings here.
>
> __[About kernel definition.]__ __The mechanism of NCC loss is assigning a data sample to the most similar prototype via measuring the similarities between samples and prototypes.__ Such similarities are usually depicted as distance function $d(\boldsymbol{z}, \boldsymbol{c})$, such as Euclidean distances or cosine similarity, in recent cross-domian few-shot classification literatures[3][4][5]. __Both negative Euclidean distance and cosine similarity play the same role in measuring the similarities__. For Euclidean distance, smaller distance means higher similarity. Thus, negative sign is used to highlight the smaller distance. Differently, cosine similarity directly measures the similarities via angles between two vectors. __A special case is that $||\boldsymbol{a}-\boldsymbol{b}||^2=2(1-cos<\boldsymbol{a}, \boldsymbol{b}>)$ if the given two vectors $\boldsymbol{a}$ and $\boldsymbol{b}$ are normalized to unit length__.
>
> From our perspective, __it is normal and reasonable to view a question from different angles__. For example, __as mentioned in the original paper of NCC-based loss[9], NCC-based loss is treated as performing mixture density estimation with an exponential family distribution.__ Thus, it is reasonable that you treat $e^{-d(\boldsymbol{z}, \boldsymbol{c})}=$ as Gaussian kernels without bandwidth.
>
> However, __in our paper, we just simply define the distance function $k(\boldsymbol{z}, \boldsymbol{c})$ (or $d(\boldsymbol{z}, \boldsymbol{c})$) as kernel function that measures the similarities among samples, and treat $p(y=c|\boldsymbol{z}, \theta)=\frac{e^{k(\boldsymbol{z}, \boldsymbol{c}_{c})}}{\sum\_\boldsymbol{c}^{'}e^{k(\boldsymbol{z}, \boldsymbol{c}^{'})}}$ as a likehood in the format of softmax function (SimpleCNAPs also treats it as softmax function, see Eq. (1) of [10]).__ (Since we just define $k(\boldsymbol{z}, \boldsymbol{c})$ as cosine similarity instead of negative Euclidean distance[3] or negative cosine similarity[5], we drop the negative sign.) __Such definition of kernel function is also used in other works__. For example, SSL-HSIC[6] treats the $k(\boldsymbol{z}, \boldsymbol{z}^{'})$ in InfoNCE as the kernel function in a form of inner product[7] or cosine similarity[8]. This definition is also mentioned in other works regarding HSIC[1].
>
> __We hope that the above explanations helps address your concern. If you still have any question, please feel free to inform us. We will try our best to help address your concern.__
>
>
> __[About Mahalanobis distance.]__ __After we check the paper and source code of URL, we think the distance function used in URL is cosine similarity__.
> - According to Eq.(5) in [original URL paper[5]](https://arxiv.org/pdf/2103.13841.pdf), authors of URL claim that $d(\boldsymbol{z}, \boldsymbol{c}_l)$ as __negative cosine similarity__.
> - According to __discussion part in page 5 of [original URL paper[5]](https://arxiv.org/pdf/2103.13841.pdf)__, it just says that __"Our adaptation strategy can be seen as a generalization of Mahalanobias distance computation."__
> - According to [source code of loss (Line 134)](https://github.com/VICO-UoE/URL/blob/master/models/losses.py) and [source code of experimental settings (Line 80)](https://github.com/VICO-UoE/URL/blob/master/config.py), it is obvious that URL uses cosine similarity instead of Mahalanobias distance.

---

> > ### Author Response · Authors · 2023-11-19
> > **Rebuttal from authors to reviewer rkmN (Weakness 2 (2/2))**
> >
> > __[About Gaussian kernel with width.]__ __At the beginning of this part, we would like to highlight again that we define the distance function $k(\boldsymbol{z}, \boldsymbol{c})$ (or $d(\boldsymbol{z}, \boldsymbol{c})$) as a kernel function, and treat $p(y=c|\boldsymbol{z}, \theta)=\frac{e^{k(\boldsymbol{z}, \boldsymbol{c}\_{c})}}{\sum\_{c^{'}}e^{k(\boldsymbol{z}, \boldsymbol{c}^{'})}}$ as a likehood in the format of softmax function.__
> >
> > In this paper, __we propose a new interpretation of NCC-based loss from the angle of dependence measures__. Such interpretation reveals that NCC-based loss can be expressed in the format of dependence measure (Hilbert-Schmidt Independence Criterion), and such dependence measure plays an important role in learning a set of good class-specific representations from few labeled samples.
> >
> > However, according to the qualitive results in this paper, we find that conventional NCC-based loss sometimes fails to precisely measure the dependence between data samples, and in turn results in high similarities among samples. Such phenomenon may induce uncertainties and result in misclassification of samples.
> >
> > In order to solve this problem, we resort to maximizing the test power of the kernel dependence measures to increase their ability in precisely detecting dependence between data. __To this end, we select kernels with trainable parameters, such as Gaussian kernels and IMQ kernels, so that we can optimize the test power of kernel dependence measures before optimizing kernel dependence between data__. From our perspective, we think such strategy is reasonable __since we can just simply treated the optimized kernels as a much more powerful kernel__.

---

> > ### Comment · Reviewer_rkmN · 2023-11-20
> > **reviewer response**
> >
> > In that case, you are basically using cross-entropy loss without bias term. Trying to relate it to kernels is even more confusing. Also, in that case you confining the method to work in the outer shells of hyperspherical CNN feature spaces, This is very dangerous especially in open set recognition settings.
> > Furthemore, I  checked the paper again, it seems the authors using Eucliden distances, not the Mahalanobisdistances  and definitely not the cosine angles. See section 2.4 on page 4.

---

> ### Author Response · Authors · 2023-11-19
> **Rebuttal from authors to reviewer rkmN (Weakness 3)**
>
> > __Weakness 3.__  All the arguments given under Theorem 1 are not new, they are given in the main NCC paper. Similarly, the arguments given below Theorem also exist in (Li et al., 2021b).
>
> Thank you for your comments. We think there might be some misunderstandings here.
>
> __[Arguments of Theorem 1.]__ Per your first concern about the arguments of Theorem 1, we agree that the intuition of NCC-based loss is so concise and simple that it is easy for us to understand. However, as far as we know, few paper, including the original paper of NCC[7], has discussed the insights behind NCC-based loss __from the perspective of dependence measures__.
>
> In this paper, we __provide a new perspective from dependence measures to reveal that NCC-based loss can be expressed in the format of dependence measures (Hilbert-Schmdit Independence Criterion), and such dependence measure plays an important role in learning desirable class-specific representations, where similarities among samples within the same class are maximized while similarities between samples from different classes are minimized, from limited number of labeled data of a given task__. Such interpretation further motivates us to improve the capbility of dependence measures via maximizing the test power to increase the sensitivity of dependence measures to dependence.
>
> __[Arguments of Theorem 3.]__ Per your second concern regarding the arguments of Theorem 3, we agree that our arguments of Theorem 3 share __the same high-level insight__ with those of SSL-HSIC. In fact, __label information $Y$ is a kind of manually designed identity information of cluster structure__. Thus, __for any format of label information, maximizing ${\rm HSIC}(\boldsymbol{Z}, Y)$ is equivalent to exploring an optimal feature space that matches such cluster structure__.
>
> However, we would like to argue here that __different designs of label information determines completely different results of representation learning__.
>
> Let's take SSL-HSIC and MOKD as examples. SSL-HSIC is derived from unsupervised contrastive learning. In this learning paradigm, __each sample is treated individually as a 'class'__. Thus, maximizing ${\rm HSIC}(\boldsymbol{Z}, Y)$ is equivalent to learning __discriminative instance representations__ for each data sample so that all samples within a batch are distinct enough with each other.
>
> In contrast, as for MOKD, it is derived from supervised few-shot learning. In this learning paradigm, __the key is learning a set of representations, where the representations are general enough among samples within the same class yet discriminative enough from those of other classes, for each class__. Thus, based on this idea, identity information is set consistently for data samples within the same class. Thus, maximizing ${\rm HSIC}(\boldsymbol{Z}, Y)$ is equivalent to inferring __discriminative class-specific representations__ with only few samples for each class in a given task.
>
> Thus, although similar arguments are shared between MOKD and SSL-HSIC, due to the distinct label information designs under respective learning paradigm, they are different.

---

> ### Author Response · Authors · 2023-11-19
> **Rebuttal from authors to reviewer rkmN (Weakness 4 & Q2)**
>
> > __Weakness 4.__ The optimization problem given at (5) is a constrained optimization problem and I wonder how the authors enforce these two constraints. For the first one, they convert it to selection of a width term if I understand correctly. What about the second constraint? Please give more information on optimization of the problem. __Q2.__ Please explain how the optimization problem is solved in more details (e.g., how the contraints are enforced?)
>
> Per your concern, __we have stated how to perform optimization on the two constraints $\max\_{\sigma\_{\boldsymbol{Z}Y}}\frac{{\rm HSIC}(\boldsymbol{Z}, Y; \sigma\_{\boldsymbol{Z}Y}, \theta)}{\sqrt{v\_{\boldsymbol{Z}Y}+\epsilon}}, \max\_{\sigma\_{\boldsymbol{Z}\boldsymbol{Z}}}\frac{{\rm HSIC}(\boldsymbol{Z}, \boldsymbol{Z}; \sigma\_{\boldsymbol{Z}\boldsymbol{Z}}, \theta)}{\sqrt{v\_{\boldsymbol{Z}\boldsymbol{Z}}+\epsilon}}$ in Section 4.2 (Page 7) and Appedix F.3 (Page 20)__.
>
> As mentioned in our paper, MOKD is a bi-level optimization framework that include two optimization steps.
>
> __[Inner optimization: Test Power Maximization.]__ The inner optimization aims at maximizing the test power of kernel HSIC measures to enhance their ability in detecting dependence between two batches of data. The inner optimization is performed respectively for both $\frac{{\rm HSIC}(\boldsymbol{Z}, Y; \sigma_{\boldsymbol{Z}Y}, \theta)}{\sqrt{v_{\boldsymbol{Z}Y}+\epsilon}}$ and $\frac{{\rm HSIC}(\boldsymbol{Z}, \boldsymbol{Z}; \sigma_{\boldsymbol{Z}\boldsymbol{Z}}, \theta)}{\sqrt{v_{\boldsymbol{Z}\boldsymbol{Z}}+\epsilon}}$ via bandwidth selection. __Since the optimal bandwidths are independently selected from a discrete space respectively for $\frac{{\rm HSIC}(\boldsymbol{Z}, Y; \sigma\_{\boldsymbol{Z}Y}, \theta)}{\sqrt{v\_{\boldsymbol{Z}Y}+\epsilon}}$ and $\frac{{\rm HSIC}(\boldsymbol{Z}, \boldsymbol{Z}; \sigma\_{\boldsymbol{Z}\boldsymbol{Z}}, \theta)}{\sqrt{v\_{\boldsymbol{Z}\boldsymbol{Z}}+\epsilon}}$, it can be guaranteed that test power values of both $\frac{{\rm HSIC}(\boldsymbol{Z}, Y; \sigma\_{\boldsymbol{Z}Y}, \theta)}{\sqrt{v\_{\boldsymbol{Z}Y}+\epsilon}}$ and $\frac{{\rm HSIC}(\boldsymbol{Z}, \boldsymbol{Z}; \sigma\_{\boldsymbol{Z}\boldsymbol{Z}}, \theta)}{\sqrt{v\_{\boldsymbol{Z}\boldsymbol{Z}}+\epsilon}}$ are maximized in each adaptation episode__.
>
> __[Outer optimization: Dependence Optimization.]__ The outer optimization aims to __maximize the dependence between data features and labels to explore features that match the cluster structures in the given task__ while **minimize the dependence among all samples to penalize high variance kernelized features and alleviate high similarities resulted from trivial common features**. The outer optimization is performed via Adadelta optimizer.
>
> For more information, you can also check our code file `MOKD/models/hsic_estimation.py` in our zipped supplementary file.

---

> ### Author Response · Authors · 2023-11-19
> **Rebuttal from authors to reviewer rkmN (Weakness 5 & Q1)**
>
> >__Weakness 5.__ The authors compare their proposed method to URL, but I wonder why they do not compare against SSL-HSIC and NCC? __Q1.__ The authors compare their proposed method to URL, but I wonder why they do not compare against SSL-HSIC and NCC?
>
> Per your question, we think there might some misunderstandings here. In fact, we have compared MOKD respectively with NCC and SSL-HSIC in our paper.
>
> __[Comparison to NCC.]__ As mentioned in **preliminary section** of our paper, __NCC-based loss is the loss used in URL__. You can refer to `Pre-training Few-shot Classification Pipeline` of preliminary section for details. Thus, we have compared MOKD with NCC-based loss (URL) in our experiment section.
>
> __[Comparison to SSL-HSIC.]__ Per your concern, we have applied SSL-HSIC to solve CFC tasks in our paper and __the results are reported in Table 4 in Appendix D__. According to the reported results in both Table 2 and Table 4, we find that __MOKD performs better than SSL-HSIC__. __An interesting phenomenon we would like to note here is that the performance of SSL-HSIC increases when test power maximization is applied to SSL-HSIC. Such phenomenon significantly demonstrate the power of TPM__.
>
> #### Reference
> [1] Pogodin et al. Kernelized information bottleneck leads to biologically plausible 3-factor Hebbian learning in deep networks, NeurIPS 2020.
>
> [2] Ma et al., The HSIC Bottleneck: Deep Learning without Back-Propagation, AAAI 2020.
>
> [3] Triantafillou et al., Meta-dataset: a dataset of datasets for learning to learn from few-samples, ICLR 2020.
>
> [4] Liu et al., A universal representation transformer layer for few-shot image classification, ICLR 2021.
>
> [5] Li et al., Universal representation learning from multiple domains for few-shot classifiation, ICCV 2021.
>
> [6] Li et al., Self-supervised learning with kernel dependence maximization, NeurIPS 2021.
>
> [7] Oord et al., Representation learning with contrastive predictive coding, arXiv: 1807.03748, 2018.
>
> [8] Chen et al., A simple framework for contrastive learning of visual representation, ICML 2020.
>
> [9] Snell et al., Prototypical networks for few-shot learning, NeurIPS 2017.
>
> [10] Bateni et al., Improved few-shot visual classification, CVPR 2020.

---

> > ### Comment · Reviewer_rkmN · 2023-11-20
> > **Comparison to the related methods**
> >
> > It is better to report the results related to SSL-HSIC in Table 1 in the main text, not in the appendix section. It is the most related method to the proposed method given in the paper.

---

> > > ### Author Response · Authors · 2023-11-20
> > > **Thanks for your quick reply!**
> > >
> > > Dear Reviewer rkmN,
> > >
> > > Many thanks for your quick reply to our responses. We will try to address your further concerns more precisely and demonstrate our thoughts.
> > >
> > > Best regards,
> > >
> > > The authors

---

> ### Author Response · Authors · 2023-11-20
> **Would you mind checking our responses and confirming whether you have any further questions?**
>
> Dear Reviewer rkmN,
>
> Thank you again for your efforts in reviewing our work and valuable comments. We have provided detailed responses to all your concerns and questions regarding our work. Would you mind checking our responses and confirming whether you have any further questions?
>
> Since the deadline of discussion (22 Nov) is approaching, we hope that our responses helps address your concerns. If you still have any concern or question regarding our work, please let us know. We are glad to help address any concern and question.
>
> Best regards,
>
> Authors of Paper 2357

---

> ### Author Response · Authors · 2023-11-21
> **Responses from authors to reviewer rkmN for further concerns.**
>
> Dear Reviewer rkmN:
>
> Thank you again for your quick response. Now, we would like to address your concerns in the following.
>
> > These explanations are irrelevant. I simply indicated that the method proposed here is an extension of a previously published paper, which limits the neovelty. That is it. You did not have to go through Hilbert-Schmidt Independence Criterion and all other stuff.
>
> **Response:** If we directly look at our solution from the perspective of techniques and implementation, the main difference between our method and SSL-HSIC is that we consider learning a more powerful kernel via maximizing the test power of HSIC. By comparing to SSL-HSIC, our method performs better (see Table 4 in Appendix D).
>
> However, our contribution is **beyond** the above technical difference. In our paper, we **first provide a new perspective to interpret** NCC-based loss from kernel dependence measure, which is totally new to the field of few-shot classification. Based on our **new interpretation**, we are then motivated to consider a more powerful kernel dependence measure that is our final technical solution.
>
> What we want to emphsize is that **the new interpretation of NCC-based loss is also a solid contribution to the field**. Without this new interpretation, we cannot link kernel dependence measure to NCC-based loss and cannot be motivated to consider a powerful kernel dependence measure to learn better class-specific representations.
>
> > In that case, you are basically using cross-entropy loss without bias term. Trying to relate it to kernels is even more confusing. Also, in that case you confining the method to work in the outer shells of hyperspherical CNN feature spaces, This is very dangerous especially in open set recognition settings. Furthemore, I checked the paper again, it seems the authors using Eucliden distances, not the Mahalanobisdistances and definitely not the cosine angles. See section 2.4 on page 4.
>
> **Response:** Thanks for your new comments and suggestions. We split our responses into three parts based on your comments in the following.
>
> 1) Response to *"In that case, you are basically using cross-entropy loss without bias term. Trying to relate it to kernels is even more confusing."*: We are not sure what "cross-entropy loss without bias term" means and why this term is highly relevant to our method. In our method, we directly optimize **the dependence objective** mentioned in Eq. (5) (page 5 in our paper), instead of the cross-entropy loss. We hope to get more explanation regarding this term such that we can precisely address your concerns.
>
> 2) Response to *"This is very dangerous especially in open set recognition settings."*: Our paper mainly focuses on the cross-domain few-shot classification problem in the close-set scenario, which is an important research topic in the deep learning area [1-8]. The main aim is to obtain a good classifier based on few-shot data. Although considering open-set scenarios is interesting, it is out of scope of our paper. However we will include relevant discussion in our paper. Thanks for pointing out this!
>
> 3) Response to the remaining part: Per your question about the Euclidean distance, we also checked the URL paper and its codes again, and we are sure that the distance function used in URL is cosine similarity. To make it more clear, here are the original paper and the corresponding codes:
> - Original paper (URL): https://arxiv.org/pdf/2103.13841.pdf (see Eq. (5) in page 5 of URL paper)
> - Codes (config python file, line 80): https://github.com/VICO-UoE/URL/blob/56a57aaf5fbd9b8e287244a27daac89992e54d25/config.py#L80C1-L81C56
> - Codes (losses python file, line 133): https://github.com/VICO-UoE/URL/blob/56a57aaf5fbd9b8e287244a27daac89992e54d25/models/losses.py#L133C1-L134
> - Codes (test_extractor_pa python file, line 73): https://github.com/VICO-UoE/URL/blob/56a57aaf5fbd9b8e287244a27daac89992e54d25/test_extractor_pa.py#L73
>
> - Besides, As our previous response, cosine similarity plays the same role as Euclidean distance. This is also mentioned in BYOL in Eq. (2). See BYOL paper from this link: https://arxiv.org/pdf/2006.07733.pdf.
>
> > It is better to report the results related to SSL-HSIC in Table 1 in the main text, not in the appendix section. It is the most related method to the proposed method given in the paper.
>
> Per your concern, Table 1 mainly reported mainstream cross-domain few-shot classification works. Since our work mainly focuses on cross-domain few-shot classification instead of comparing MOKD and SSL-HSIC, we did not report Table 4 in Table 1 in our original submission. However, we agree that it is better to include the SSL-HSIC's results in Table 1, which can make readers know the benefits of our method more clearly. Thanks for the suggestions.

---

> > ### Author Response · Authors · 2023-11-21
> > **References of our new responses**
> >
> > #### Reference
> >
> > [1] Triantafillou et al., Meta-dataset: a dataset of datasets for learning to learn from few examples, ICLR 2020.
> >
> > [2] Requeima et al., Fast and flexible multi-task classification using conditional neural adaptive process, NeurIPS 2019.
> >
> > [3] Bateni et al., Improved few-shot visual classification, CVPR 2020.
> >
> > [4] Doersch et al., CrossTransformers: spatially-aware few-shot transfer, NeurIPS 2020.
> >
> > [5] Dvornik et al., Selecting relevant features from a multi-domain representation for few-shot classification, ECCV2020.
> >
> > [6] Liu et al., A universal representation transformer layer for few-shot image classification, ICLR 2021.
> >
> > [7] Triantafillou et al., Leanring a universal template for few-shot dataset generalization, ICML 2021.
> >
> > [8] Li et al., Universal representation learning from multiple domains for few-shot classification, ICCV 2021.

---

> > ### Comment · Reviewer_rkmN · 2023-11-21
> > **regarding cross entropy**
> >
> > If you use cosine distances in equation 1, i.e., $z^t c_c$, it basically becomes the well-known cross-entropy loss. The only difference a bias is not used in the sense that $z^t c_c + b_c$ and the features are normalized. Relating this formulation to kernels is more confusing. Cross-entropy is well-studied. In that case you are simply applying cross-entropy loss in hyperspehrical CNN feature spaces (outer shells of the hypersphere). This might work for high-dimensional spaces, but in genereal confining all feature space to the shells of a unit hypersphere is not a good idea.

---

> ### Author Response · Authors · 2023-11-22
> **Rebuttal for your concern about cross entropy**
>
> Dear Reviewer rkmN,
>
> Thank you for kindly providing detailed explanations. Now, we would like to address your concern in the following.
>
> > If you use cosine distances in equation 1, i.e., $\boldsymbol{z}^{t}\boldsymbol{c}\_c$, it basically becomes the well-known cross-entropy loss. The only difference a bias is not used in the sense that $\boldsymbol{z}^{t}\boldsymbol{c}\_c+\boldsymbol{b}\_c$ and the features are normalized. Relating this formulation to kernels is more confusing. Cross-entropy is well-studied. In that case you are simply applying cross-entropy loss in hyperspehrical CNN feature spaces (outer shells of the hypersphere). This might work for high-dimensional spaces, but in genereal confining all feature space to the shells of a unit hypersphere is not a good idea.
>
> Per your concern, we follow your suggestion and substitute $k(\boldsymbol{z}, \boldsymbol{c}\_c)$ with $\boldsymbol{z}^{t}\boldsymbol{c}\_c$, where both $\boldsymbol{z}$ and $\boldsymbol{c}\_c$ are normalized, in Eq. (1). Then, we can obtain $-log\frac{e^{\boldsymbol{z}^{t}\boldsymbol{c}\_c}}{\sum\_{c^{'}}e^{\boldsymbol{z}^{t}\boldsymbol{c}\_c}}$. In this case, due to the similar format, Eq. (1) can be viewed as a cross entropy loss.
>
> However, if we take a close look at cosine similarity, it is easy to observe that  __cosine similarity $\boldsymbol{z}^{t}\boldsymbol{c}\_c$ itself is actually a linear kernel__. Thus, it is reasonable to treat it as a kernel. Even in the most general case that we add the bias term and express it as $\boldsymbol{z}^{t}\boldsymbol{c}_c+\boldsymbol{b}_c$, it is still a specicial case of polynomial kernel. Thus, __we can express Eq. (1) in the format of $e^{k(\boldsymbol{z}, \boldsymbol{c}\_c)}$, where $k(\boldsymbol{z}, \boldsymbol{c}\_c)$ is a general expression of kernel function__. Such expression follows the general practice and is widely applied in previous works as we mentioned in our previous responses. In deep learning areas, a linear layer is usually adopted in model learning to function as a classifier. The output of such classifier is usually expressed in the format of inner product, such as $\boldsymbol{z}^{t}\boldsymbol{W}+\boldsymbol{b}$. Thus, we can also generally view it as a kernel $k(\boldsymbol{z}, \boldsymbol{W})$. Thus, defining cosine similarity as a kernel is just a way how we treat it. In this paper, treating cosine similarity as a kernel provides a unified and clear perspective to discuss insights behind NCC-based loss, and this also contributes to our further theoretical analysis that provides a new perspective from kernel HSIC dependence measure to interpret NCC-based loss.
>
> Moreover, per your concern about confining features to the shells of a unit hyperspere, we searched some literatures regarding the effect of hypersphere towards open set recognition. We found it is an interesting topic [1-4]. For example, Cevikalp et al. [4] proposes to approximate the class acceptance regions with compact hypersphere models for anomaly detection and open set recognition. From this perspective, if we would like to further extend our method to open set recognition, we have to reconsider the utilization of cosine similarity to avoid the condition you mentioned. Thank you for pointing out this, we will add some more discussion regarding this in our paper.
>
> #### Reference
>
> [1] Dhamija et al., Reducing network agnostophobia, NeurIPS 2018.
>
> [2] Yang et al., Convolutional prototype network for open set recognition, IEEE Pattern Analysis and Machine Intelligence, 2020.
>
> [3] Miller et al., Class anchor clustering: A loss for distance-based open set recognition, WACV 2021.
>
> [4] Cevikalp et al., From anomaly detection to open set recognition: Bridging the gap, Pattern Recognition, 2023.

---

> > ### Comment · Reviewer_rkmN · 2023-11-22
> > **last comment**
> >
> > I do not want to discuss this issue more, but Cevikalp et al. [4] did not use the outer shells of the hyeprspheres, they have done completely a different thing. They represented (approximated) each class with a hypersphere center and these hyperspheres were scatteted in all feature space, Moreover, the Eucliden distances are used for assignment. In contrast, you are not using entire feature space, in your case, the entire feature space is the outer shells of a hypersphere. This idea is used in ArcFace method and it cannot be used for open set recognition.

---

> ### Comment · Reviewer_rkmN · 2023-11-22
> **my final rating**
>
> In general, I believe the paper has some novelty, but it is limited. It is mor elike extension of an existing methodology. Furthermore, some of the theorems are already exist in similar studies.  Confining all feature space to outer of an hypersphere also limits the application areas of the method. Therefore, I am keeping my original score, the paper is marginally below the acceptance threshold.

---

### Comment · Area_Chair_yv8c · 2023-11-22
**Let's have more discussion with authors**

Dear reviewers,

Your interaction with the authors on this work is highly appreciated.

The author-reviewer discussion period is closing at the end of Wednesday Nov 22nd (AOE). Let's take this remaining time to have more discussions with the authors on their responses to your reviews. Should you have any further opinions, comments or questions, please let the authors know asap and this will allow the authors to address them.

Kind regards, AC

---

> ### Comment · Reviewer_rkmN · 2023-12-02
> **last evaluation**
>
> I am also keeping my initial score which is marginally below the acceptance threshold. I believe the employed loss function is not a good choice especially for few-shot classification. The authors employ the nearest centroid classifer, yet the distances are computed based on angles alone. This translates into restriction of entire feature space onto a boundary of a unit hypersphere as in ArcFace method. This is a great limitation in my opinion. I also want to clarify a point, no one asked the authors to conduct anything on open set recognition, but this setting is also similar (i.e, it is more like anomaly detection). Please note that we have very limited samples belonging to a particular class and we want to discriminate this particular class examples against to a diverse class categoreies. Using Euclidean distances makes more sense, and the cited papers by the authors state using the Euclidean or Mahalanobis distances. However, the authors claim they use cosine distances in the provided codes. I did not check it.

---

### Meta-Review · Area_Chair_yv8c · 2023-12-10

**Metareview:**

Based on the submission, reviews, and author feedback, the main points that have been raised are summarised as follows.

Strengths:
1. This work is built on theoretical foundations and derivations, demonstrating the interesting ties between NCC-based loss and HSIC measure.
2. This paper conducted extensive experiments to verify the proposed solution, which is empirically solid.
3. This work makes an interesting exploration by introducing the HSIC criterion into FSL.
4. The proposed method does not significantly increase the running time.

Issues:
1. This work is a revised version of SSL-HSIC in the literature, which limits the novelty.
2. There are some wrong statements; some theoretical results are not new.
3. Need to further clarify the proposed optimisation problem, the implicit assumption, and some arguments.
4. Experimental study (including more comparison, hyper-parameter sensitivity, marginal improvement) needs to be clarified.
5. (Raised by AC) The "uniform distribution" defined in Theorems does not seem to be sound.

The authors have done well in providing feedback to address each of the raised issues. After reading this submission, AC agrees that this work has its merits in demonstrating the ties between NCC-based loss and HSIC measure and developing a bi-level optimisation to optimise the kernel-based measure built upon HSIC. The experimental study shows the efficacy of the proposed method. Meanwhile, this work is developed upon existing work SSL-HSIC; the arguments at multiple places need to be strengthened; performance improvement sometime is marginal; more importantly, the rigorousness of theoretical derivation needs to be improved. AC discussed this work with SAC.

**Justification For Why Not Higher Score:**

Although this work has its merits, overall this work makes (good) incremental contribution with respect to existing work. Also, the arguments at multiple places need to be strengthened; and more importantly, the rigorousness of theoretical derivation needs to be improved.

**Justification For Why Not Lower Score:**

N/A

---

### Decision · Program_Chairs · 2024-01-16

Reject